# An Analytical Model for Overparameterized Learning Under Class Imbalance

**Eliav Mor**                                                       *moreliav@mail.tau.ac.il*
*Department of Computer Science*
*Tel Aviv Univeristy*

**Yair Carmon**                                                     *ycarmon@tauex.tau.ac.il*
*Department of Computer Science*
*Tel Aviv Univeristy*

**Reviewed on OpenReview:** *https://openreview.net/forum?id=69RntSRF5K*

## Abstract

We study class-imbalanced linear classification in a high-dimensional Gaussian mixture model. We develop a tight, closed form approximation for the test error of several practical learning methods, including logit adjustment and class dependent temperature. Our approximation allows us to analytically tune and compare these methods, highlighting how and when they overcome the pitfalls of standard cross-entropy minimization. We test our theoretical findings on simulated data and imbalanced CIFAR10, MNIST and FashionMNIST datasets.

## 1 Introduction

Learning from class-imbalanced data is a longstanding and central challenge in machine learning (He & Garcia, 2009; Van Horn & Perona, 2017; Buda et al., 2018; Huang et al., 2019). Its importance stems from the fact that in many applications some classes are under-represented in the training data, yet we still desire classifiers that perform well on these minority classes. Examples include face recognition (Merler et al., 2019; Wang et al., 2018; Deng et al., 2019), wildlife conservation (Tuia et al., 2022; Sullivan et al., 2009; Van Horn et al., 2018), medical diagnosis (Mac Namee et al., 2002), and fraud detection (Singh et al., 2022).

Standard machine learning methods are particularly sensitive to class imbalance in the *overparameterized* regime[1], in which the model is capable of perfectly fitting the entire training data. The predominant approach for multi-class classification, namely minimizing an empirical cross-entropy loss, often generalizes very poorly to minority classes. One may intuitively explain this fact by viewing empirical cross-entropy as a surrogate for the population classification error, which discounts minority classes. It is therefore natural to seek a loss that approximates a class-balanced population error, leading to classical techniques such as re-weighting the losses or re-sampling of the training data (He & Garcia, 2009). However, this intuition and the resulting techniques both fail in the overparameterized setting, where the training cross-entropy loss becomes zero (and hence does not approximate the population error) (Rosset et al., 2003; Soudry et al., 2018; Lyu & Li, 2020) and re-weighting or re-sampling the data has little effect on the resulting classifier (Byrd & Lipton, 2019; Fang et al., 2020; Xu et al., 2021; Zhai et al., 2023).

These observations have led to extensive work on methods that modify the cross-entropy objective to improve the performance of overparameterized models on minority classes (Cao et al., 2019; Menon et al., 2021; Ye et al., 2020; Lu et al., 2022; Nguyen et al., 2021; Kini et al., 2021; Lin et al., 2017; Cui et al., 2019; Hong et al., 2021; Ren et al., 2020; Samuel & Chechik, 2021; Li et al., 2021; Behnia et al., 2023). We focus on

---

[1]The "overparameterized regime" characterizes a setting where the number of parameters in a model significantly exceeds the size of the training set, allowing the model to interpolate the training set.

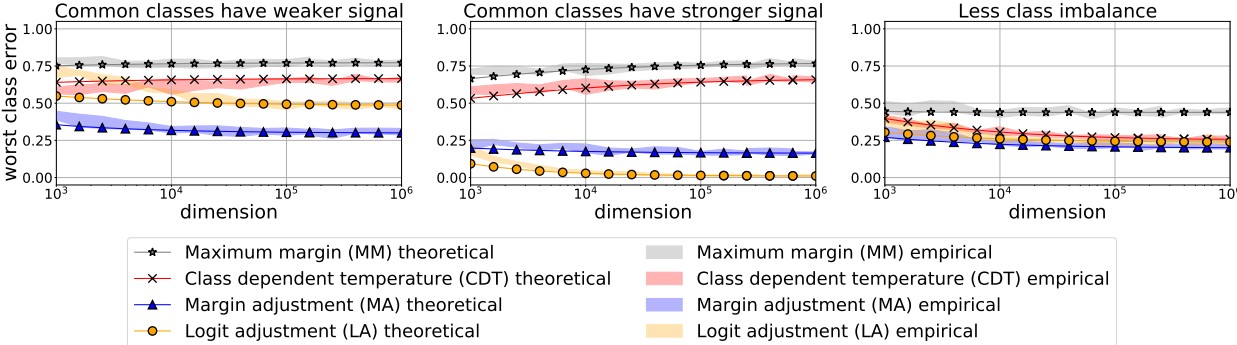

**Figure 1.** **Illustration of our main findings.** We plot the test error of the worst-performing class as a function of model dimension, for the different learning methods we consider. The shaded areas indicate empirical measurements and the solid lines show our analytical approximation prediction. Each panel shows a different set of model parameters; see Appendix G.1 for detailed description.

three leading approaches: logit adjustment (LA) (Menon et al., 2021), class-dependent temperature (CDT) Ye et al. (2020), and margin adjustment[2] (MA) (Behnia et al., 2023; Lu et al., 2022; Nguyen et al., 2021) (see Section 6 for discussion of additional pertinent methods). While LA, MA and CDT are effective in the overparameterized class-imbalanced setting, they each introduce additional hyperparameters, and it is unclear whether one method is fundamentally preferable to the other.

**Our contribution.** In this work, we shed light on the failure mechanism of class-imbalanced overparameterized cross-entropy minimization, and on the relative merits of LA, MA, and CDT. To do so, we consider high-dimensional (overparameterized) linear classification in a simple class-imbalanced Gaussian mixture data model with varying signal strengths between classes. We focus on learning by applying (stochastic) gradient descent on the empirical risk until reaching loss zero, which is equivalent to solving a (modified) margin maximization problem (Soudry et al., 2018; Behnia et al., 2023; Kini et al., 2021). In this setting, we develop a tractable analytical approximation for the test error as a function of the model parameters and balancing hyperparameters. Figure 1 illustrates the tightness of our approximation, as well as our key findings:

1. Our model clearly exhibits the failure of cross-entropy—i.e., the maximum margin (MM) classifier—on minority classes. Moreover, our approximation quantitatively shows how class imbalance affects test error, and reveals the failure mechanism of cross-entropy minimization: minority classifier vectors become dominated by noise from majority examples.

2. Our approximations allow us to derive closed form expressions for near-optimal hyperparameter tuning for MA and LA, and shows that both methods successfully mitigate class imbalance.

3. While CDT is sometimes helpful, there exist problem instances where MA and LA perform well and CDT performs arbitrarily badly, for any choice of its hyperparameters. Moreover, CDT sometimes fails to reach zero training error even for linearly separable training data.

4. Our analysis delineates the combined effect of class imbalance and class signal strength on test error across different losses. Notably, even in our simplified setting, no method universally dominates the other: carefully-tuned LA performs better when the signal strength is either identical across classes or positively correlated with per-class sample size, while MA may perform better in other settings.

Overall, we believe that our work provides a helpful perspective on the challenges of overparameterized learning of imbalanced data, as well as a useful theoretical proving ground for new techniques.
**Going beyond linear classification on Gaussian mixtures.** To test whether our conclusions hold beyond the stylized setting that we analyze, we experiment with non-linear classifiers and a standard class-imbalanced benchmark. More specifically, we study an imbalanced version of the CIFAR10 (Krizhevsky

---

[2]MA does not have a standard name in the literature: Behnia et al. (2023) call it "label dependent temperature," Lu et al. (2022) call it "importance tempering" and Nguyen et al. (2021) call it "margin scaling."

et al., 2009), MNIST (Deng, 2012) and FashionMNIST (Xiao et al., 2017) datasets using either a radial basis function (RBF) kernel or fine-tuning a pre-trained CLIP model (Radford et al., 2021). In the kernel experiments, our conclusions hold fairly well: cross-entropy generalizes poorly, CDT fails in certain settings, and our theoretically-derived tuning of MA and LA performs well. In the CLIP experiments, CDT performs about as well as MA but seems more sensitive to parameter-tuning, and our theoretically-derived tuning for LA performs comparably to standard tuning.

**Paper organization.** In Section 2 we present the problem setting, our data model, and the methods we consider. In Section 3 we describe our analytical approximation for the classifiers learned by the different methods and their test performance. In Section 4 we apply our approximation on MA, MM, LA and CDT, providing expressions for near-optimal hyperparameter tuning and comparing the different methods. In Section 5 we report on experiments within our model and beyond. Finally, in Section 6, we present an extended discussion covering additional related work, limitations of our study, and directions for future research.

## 2 Preliminaries

**Notation.** We denote matrices by capital letters, and vectors and scalars by lowercase letters. We denote the $i$'th column of $A$ by $a_i$, and we use brackets to refer to specific matrix or vectors entries, i.e., $A_{[i,j]} = a_{i[j]}$. We write $\|v\|$ for the Euclidean norm of $v$, while $\|A\|_{\mathrm{F}}$ and $\|A\|_{\mathrm{op}}$ denote the Frobenius and $\ell_2$ operator norm of $A$, respectively. We let $\mathbb{1}_{\{\mathfrak{E}\}}$ denote the indicator of event $\mathfrak{E}$.

### 2.1 Hypothesis class and error metrics

We consider linear predictors that classify $d$-dimensional features to one of $c$ classes. A predictor is parameterized by weights $W \in \mathbb{R}^{d \times c}$ and biases $b \in \mathbb{R}^c$. Given input $x \in \mathbb{R}^d$, it predicts $\hat{y} = \arg\max_{y \in [c]} \left\{ w_y^\top x + b_{[y]} \right\}$.

The prediction error for examples of class $y$ is

$$\mathrm{Err}_y(W, b) := \mathbb{E}_{x|y}[\ell_{0\text{-}1}(z; y)], \tag{1}$$

where $z = W^\top x + b$ and $\ell_{0\text{-}1}(z, y) = \mathbb{1}_{\left\{ y \notin \arg\max_i z_{[i]} \right\}}$ is the 0-1 loss. We also define the **worst class error** as

$$\mathrm{Err}_{\mathrm{wc}}(W, b) := \max_{y \in [c]} \mathrm{Err}_y(W, b). \tag{2}$$

Finally, we write the probability of class $k$ obtaining larger score than the true class $y$ as

$$\mathrm{Err}_{y \to k}(W, b) := \mathbb{P}_{x|y}\Big( \langle w_k, x \rangle + b_{[k]} > \langle w_y, x \rangle + b_{[y]} \Big), \tag{3}$$

and note that

$$\max_{k \neq y} \mathrm{Err}_{y \to k}(W, b) \leq \mathrm{Err}_{\mathrm{wc}}(W, b) \leq \max_y \sum_{k \neq y} \mathrm{Err}_{y \to k}(W, b) \leq c \cdot \max_{k \neq y} \mathrm{Err}_{y \to k}(W, b). \tag{4}$$

### 2.2 Learning methods

We now formally define the methods that our work analyzes and compares. Given a training set $\{x_i, y_i\}_{i \in [N]}$, we consider minimizing empirical risks of the form

$$\mathcal{L}(W, b) := \frac{1}{N} \sum_{i=1}^N \ell\left( W^\top x_i + b; y_i \right),$$

over $W, b$, where $\ell : \mathbb{R} \times [c] \to \mathbb{R}_+$ is a classification loss function.

**Cross-entropy, MA and CDT.** Perhaps the most common multiclass classification loss, cross-entropy is given by

$$\ell^{\text{ce}}(z; y) := \log\left(\sum_{i \in [c]} e^{z_{[i]} - z_{[y]}}\right).$$

We analyze two modified versions of the cross-entropy loss that aim to improve overparameterized learning over imbalanced data: margin adjustment (MA) (Behnia et al., 2023; Wang et al., 2022) and class-dependent temperature (CDT) (Ye et al., 2020; Kini et al., 2021), given by

$$\ell^{\text{ma}}(z; y) := \log\left(\sum_{i \in [c]} e^{\frac{z_{[i]} - z_{[y]}}{\delta_y}}\right) \quad \text{and} \quad \ell^{\text{cdt}}(z; y) := \log\left(\sum_{i \in [c]} e^{\frac{z_{[i]}}{\delta_i} - \frac{z_{[y]}}{\delta_y}}\right),$$

respectively. Both losses have per-class hyperparameters $\{\delta_i\}_{i \in [c]}$ for which increasing $\delta_i$ pushes the model to prefer class $i$. The standard approach for selecting these hyperparameters is setting $\delta_i = N_i^{-\gamma}$ and tuning $\gamma \geq 0$ (Lu et al., 2022; Kini et al., 2021; Ye et al., 2020; Behnia et al., 2023), where for $\gamma = 0$ we recover cross-entropy.

**Equivalent margin problems.** For linearly separable training data and for each of the empirical loss functions we consider, gradient descent converges to zero loss, and its implicit bias is given by the solution to a constrained optimization problem. More specifically, consider gradient descent steps of the form

$$W_{t+1} = W_t - \eta \nabla_W \mathcal{L}(W_t, b_t) \quad \text{and} \quad b_{t+1} = b_t - \eta' \nabla_b \mathcal{L}(W_t, b_t), \quad \text{where} \quad \left(\frac{\eta}{\eta'}\right)^2 =: \rho.$$

Then, for any $\rho$ and sufficiently small $\eta$, if $\ell = \ell^{\text{ma}}$ we have $\text{Err}_y(W_t, b_t) \xrightarrow{t \to \infty} \text{Err}_y(W^{\text{ma}}, b^{\text{ma}})$ for all $y \in [c]$, where

$$W^{\text{ma}}, b^{\text{ma}} := \underset{W \in \mathbb{R}^{d \times c}, b \in \mathbb{R}^c}{\arg\min} \|W\|_{\text{F}}^2 + \rho\|b\|^2$$

$$\text{subject to} \quad (w_{y_i} - w_k)^\top x_i + b_{[y_i]} - b_{[k]} \geq \delta_{y_i} \quad \text{for all} \quad i \leq N \text{ and } k \neq y_i. \tag{5}$$

When $\ell = \ell^{\text{cdt}}$ we instead have $\text{Err}_y(W_t, b_t) \xrightarrow{t \to \infty} \text{Err}_y(W^{\text{cdt}}, b^{\text{cdt}})$ for all $y \in [c]$, where

$$W^{\text{cdt}}, b^{\text{cdt}} := \underset{W \in \mathbb{R}^{d \times c}, b \in \mathbb{R}^c}{\arg\min} \|W\|_{\text{F}}^2 + \rho\|b\|^2$$

$$\text{subject to} \quad \frac{1}{\delta_{y_i}}\left(w_{y_i}^\top x_i + b_{[y_i]}\right) - \frac{1}{\delta_k}\left(w_k^\top x_i + b_{[k]}\right) \geq 1 \quad \text{for all} \quad i \leq N \text{ and } k \neq y_i. \tag{6}$$

Substituting $\delta_i = 1$ for all $i \in [c]$ into either eq. (5) or eq. (6) yields the standard maximum margin separator, which is the implicit bias of gradient descent for $\ell = \ell^{\text{ce}}$. See Appendix B for a more formal statement of these claims, and a proof based on the prior work (Soudry et al., 2018; Ji et al., 2020; Kini et al., 2021).

*Remark* 2.1 (Limitations of CDT). The MA constraint eq. (5) guarantees zero training error, i.e., $\arg\max_{y \in [c]}\{w_y^\top x_i + b_{[y]}\} = y_i$ for all training points $(x_1, y_1), \ldots, (x_n, y_n)$. By contrast, the CDT constraint eq. (6) generally does not offer such a guarantee. Consequently, for three or more classes, optimizing the CDT loss does not always fit all the training data. Furthermore, for two classes, the choice of $\delta_1$ and $\delta_2$ has no effect on the solution to eq. (6). See more details in Appendix F.3.

**Logit adjustment.** The LA method finds $W$ and $b$ by minimizing the empirical cross-entropy risk, and adjusts the bias vector by subtracting $\iota \in \mathbb{R}^c$, i.e., replacing $b$ with $b - \iota$. In the terminology of the original proposal (Menon et al., 2021), this is the "post-hoc" correction variant of logit adjustment. Menon et al. (2021) also propose an "LA loss" for empirical risk minimization. However, we note that the implicit bias of the LA loss is identical to that of normal cross-entropy; i.e., after long enough gradient descent training, it also converges in direction to the standard max margin separator. See Appendix B and Kini et al. (2021, Theorem 4) for a more formal statement of these claims. The standard approach for selecting $\iota$ is setting $\iota_{[i]} = \tau \log N_i$ and tuning $\tau \geq 0$.

### 2.3 Data model

Inspired by Sagawa et al. (2020b); Wald et al. (2022); Kini et al. (2021); Wang et al. (2021); Glasgow et al. (2023); Carmon et al. (2019) we model our data as a high-dimensional Gaussian mixture. Specifically, conditional on the label $y$, the input $x \in \mathbb{R}^d$ has the following Gaussian distribution

$$x|y \sim \mathcal{N}(\mu_y; \sigma^2 I_d) \tag{7}$$

for some $\mu_1, \ldots, \mu_c \in \mathbb{R}^d$ and $\sigma > 0$, where $I_d$ denotes the unit matrix.

**Scaling with dimension.** To ensure that the problem we study becomes neither trivial nor impossible to learn as the dimension $d$ grows, we must carefully set its parameters as a function of $d$. To that end, we choose

$$\sigma^2 = \frac{1}{d} \ \text{ and } \ \|\mu_i\|^2 = \frac{s_i}{\sqrt{d}} \ \text{ and } \ \langle \mu_i, \mu_j \rangle = 0 \ \forall \ i \neq j. \tag{8}$$

The scaling of $\|\mu_i\|^2/\sigma^2 \asymp \sqrt{d}$ is essential for obtaining non-trivial error as $d \to \infty$ with the training set size fixed, while the choice $\sigma^2 d = 1$ is without loss of generality. Assuming orthogonal $\mu_1, \ldots, \mu_c$ facilitates interpretable analytical expression, though potentially at some loss of generality.

## 3 Analytical approximation

This section describes the two key steps in deriving our analytical approximation for the error of different learning methods in our data model: replacing the empirical kernel with its expectation and replacing the score statistics (defined below) with their expectation.

### 3.1 Expected kernel approximation

The MA and CDT problems we consider (eqs. (5) and (6) respectively) are quadratic in $W, b$ and therefore their solutions are of the form $w_y = \sum_{i \in [N]} \beta_{y[i]} x_i$, where the optimal $\beta_1, \ldots, \beta_c \in \mathbb{R}^N$, and $b \in \mathbb{R}^c$ depend on the data only through the $N \times N$ kernel matrix (see substitution of $w_y$ in eq. (5) at eq. (10))

$$K_{[i,j]} \coloneqq \langle x_i, x_j \rangle, \tag{9}$$

$$\beta^{\mathrm{ma}}, b^{\mathrm{ma}} \coloneqq \underset{\beta_1, \ldots, \beta_1 \in \mathbb{R}^N, b \in \mathbb{R}^c}{\arg\min} \left( \sum_{i=1}^c \beta_i^\top K \beta_i + \rho b_{[i]}^2 \right)$$

$$\text{subject to} \quad (\beta_{y_i} - \beta_k)^\top k_i + b_{[y_i]} - b_{[k]} \geq \delta_{y_i} \ \text{ for all } \ i \leq N \text{ and } k \neq y_i. \tag{10}$$

The first step in our approximation is to *replace $K$ by its expectation* $\mathbb{E}K_{[i,j]} = \|\mu_{y_i}\|^2 \mathbb{1}_{\{y_i=y_j\}} + \sigma^2 d \mathbb{1}_{\{i=j\}}$. We justify this approximation by observing that, under our dimension scaling, eq. (8), the empirical kernel concentrates to its expectation as $\|K - \mathbb{E}K\|_{\mathrm{op}} = O\left( \frac{\sqrt{N}}{\sqrt{d}} + \frac{1}{\sqrt{d}} \left( 1 + N d^{-1/4} \max_i s_i \right) \sqrt{\log \frac{1}{\delta}} \right)$ with probability at least $1 - \delta$; see Appendix A.2 for proof.

Let $\{\tilde{\beta}_i, \tilde{b}_{[i]}\}$ be the (deterministic) values of $\{\beta_i, b_{[i]}\}$ under the expected kernel approximation. By symmetry of $\mathbb{E}K$, the entries of $\tilde{\beta}_i$ are identical for examples of the same class. That is, for every $j \in N$, we have $\tilde{\beta}_{i[j]} = \tilde{\alpha}_{i[y_j]}$ for some $\tilde{\alpha}_i \in \mathbb{R}^c$. The approximate predictor $\tilde{W}$ is of the form

$$\tilde{w}_y = \sum_{i=1}^c \tilde{\alpha}_{y[i]} \bar{x}_i \ \text{ where } \ \bar{x}_i \coloneqq \sum_{j \in S_i} x_j \tag{11}$$

is the sum of all feature vectors from the $i$'th class. That is, under the expected kernel approximation, all the data points become support vectors, whose weight depends only on their class identity. To find $\{\tilde{\alpha}_i, \tilde{b}_{[i]}\}$ in closed form (as we do in the next section) it is convenient to express them in terms of the reduced $c \times c$ expected kernel:

$$\bar{K}_{[i,j]} \coloneqq \mathbb{E}[\langle \bar{x}_i, \bar{x}_j \rangle] = N_i^2 \xi_i \mathbb{1}_{\{i=j\}} \ \text{ where } \ \xi_i \coloneqq \|\mu_i\|^2 + \frac{\sigma^2 d}{N_i}. \tag{12}$$

## 3.2 Expected score statistics approximation

For $x|y \sim \mathcal{N}(\mu_y; \sigma^2 I_d)$ and a classifier $(W, b)$, the prediction errors are

$$\text{Err}_y(W, b) = 1 - \mathbb{P}\Big(\mathcal{N}(\nu^{(y)}; \Sigma^{(y)}) \geq 0\Big) \quad \text{and} \quad \text{Err}_{y \to k}(W, b) = Q\left(\frac{\nu^{(y)}_{[k]}}{\sqrt{\Sigma^{(y)}_{[k,k]}}}\right), \tag{13}$$

where $\text{Err}_y$ and $\text{Err}_{y \to k}$ are defined in eqs. (1) and (3), respectively, $Q(t) = \mathbb{P}(\mathcal{N}(0,1) > t)$ is the Gaussian error function, and, for every $i, j \in [c]$

$$\nu^{(y)}_{[j]} = \frac{(w_y - w_j)^\top \mu_y + b_{[y]} - b_{[j]}}{\sigma} \quad \text{and} \quad \Sigma^{(y)}_{[i,j]} = (w_y - w_i)^\top (w_y - w_j). \tag{14}$$

We refer to $\{\nu^{(y)}, \Sigma^{(y)}\}$ as the *score statistics*. See derivation in Appendix A.4.

Applying the expected kernel approximation described above, we substitute $(W, b)$ with $(\tilde{W}, \tilde{b})$, where $\tilde{w}_y = \sum_{i \in [c]} \tilde{\alpha}_{y[i]} \bar{x}_i$ and $\{\tilde{\alpha}_i, \tilde{b}_{[i]}\}$ are deterministic. This yields the following approximate score statistics

$$\tilde{\nu}^{(y)} = \frac{\tilde{A}^{(y)} \bar{X}^\top \mu_y + \tilde{b}_{[y]} \mathbf{1} - \tilde{b}}{\sigma} \quad \text{and} \quad \tilde{\Sigma}^{(y)} = \tilde{A}^{(y)} \bar{X}^\top \bar{X} \left(\tilde{A}^{(y)}\right)^\top,$$

where $\bar{X} = [\bar{x}_1, \ldots, \bar{x}_c] \in \mathbb{R}^{d \times c}$ and $\tilde{A}^{(y)} \in \mathbb{R}^{c \times c}$ is a matrix whose $i$'th row is $\tilde{\alpha}_y - \tilde{\alpha}_i$.

The second and final step in our approximation is to replace the score statistics $\tilde{\nu}^{(y)}$ and $\tilde{\Sigma}^{(y)}$ by their expectation with respect to the training data $\bar{X}$, given by

$$\hat{\nu}^{(y)} := \mathbb{E}\tilde{\nu}^{(y)} = \frac{N_y \|\mu_y\|^2 \tilde{a}^{(y)}_y + \tilde{b}_{[y]} \mathbf{1} - \tilde{b}}{\sigma} \quad \text{and} \quad \hat{\Sigma}^{(y)} := \mathbb{E}\tilde{\Sigma}^{(y)} = \tilde{A}^{(y)} \bar{K} \left(\tilde{A}^{(y)}\right)^\top, \tag{15}$$

with $\bar{K}$ as in eq. (12) and $\tilde{a}^{(y)}_y$ denoting the $y$'th column of $\tilde{A}^{(y)}$. Again we justify this approximation using concentration of measure under parameter scaling eq. (8); see Appendix A.

The expected score statistics $\hat{\nu}^{(y)}$ and $\hat{\Sigma}^{(y)}$ are deterministic quantities which we obtain in closed form for MA, CDT and LA (see summary in Appendix E). By substituting them into the expressions eq. (13) we obtain tight analytical approximations for the generalization performance of these methods.

## 4 Analysis of class balancing interventions

We now apply our approximation strategy to MA (and MM as a special case), LA, and CDT. For MA and LA we obtain near-optimal parameter values with which these methods successfully mitigate class imbalance. In contrast, our analysis shows that MM and CDT may fail catastrophically.

### 4.1 Margin adjustment and maximum margin

We begin by applying the expected kernel approximation (Section 3.1) to MA, and finding the resulting coefficients in closed form (see Appendix C.1 for proof).

**Theorem 4.1.** *Let $\tilde{W}^{\text{ma}}, \tilde{b}^{\text{ma}}$ be the expected kernel approximation for the MA predictor eq. (5) with any $\delta$ that satisfies $\delta_i > 0$ for all $i \in [c]$. Then, for all $y \in [c]$ we have $\tilde{w}^{\text{ma}}_y = \sum_{i=1}^c \tilde{\alpha}^{\text{ma}}_{y[i]} \bar{x}_i$, where,*

$$\tilde{\alpha}^{\text{ma}}_{y[i]} = \frac{\delta_i}{N_i \xi_i} \left(\mathbb{1}_{\{i=y\}} - \frac{1}{c}\right) + \frac{\sum_{j=1}^c \left[\frac{\delta_j}{\xi_j} - \frac{\delta_y}{\xi_y}\right]}{c N_i \xi_i (M + \rho)} \quad \text{and} \quad \tilde{b}^{\text{ma}}_{[y]} = \frac{\sum_{j=y}^c \left[\frac{\delta_y}{\xi_y} - \frac{\delta_j}{\xi_j}\right]}{c (M + \rho)},$$

*using $\xi_i := \|\mu_i\|^2 + \frac{\sigma^2 d}{N_i}$ and $M := \sum_{i=1}^c \frac{1}{\xi_i}$*

Substituting $\{\tilde{\alpha}_i^{\mathrm{ma}}\}_{i=1}^c, \{\tilde{b}_{[i]}^{\mathrm{ma}}\}_{i=1}^c$ into eq. (15), we complete our approximation for the error of MA. We defer the full expressions to Appendix D.1 and here we highlight key consequences, focusing on the limit $d \to \infty$ with the scaling eq. (8). To that end, we define for every $\rho$ and $\delta$ (with the latter possibly depending on $d$)

$$\widetilde{\mathrm{Err}}_{\mathrm{wc}}^{\mathrm{ma}}(\delta, \rho) := \max_{k \neq y} \lim_{d \to \infty} \mathrm{Err}_{y \to k}\big(\tilde{W}^{\mathrm{ma}}(\delta, \rho), \tilde{b}^{\mathrm{ma}}(\delta, \rho)\big). \tag{16}$$

By eq. (4) we have that $\frac{\lim_{d \to \infty} \mathrm{Err}_{\mathrm{wc}}(\tilde{W}^{\mathrm{ma}}, \tilde{b}^{\mathrm{ma}})}{\widetilde{\mathrm{Err}}_{\mathrm{wc}}^{\mathrm{ma}}(\delta, \rho)}$ is between 1 and $c$.

**Performance of maximum margin (MM).** In the special case $\delta_i = 1$ for all $i \in [c]$, we obtain a simple approximation for the worse-class error of the implicit bias of gradient descent on the standard cross-entropy loss, i.e., the MM separator:[3]

$$\widetilde{\mathrm{Err}}_{\mathrm{wc}}^{\mathrm{mm}}(\rho) := \widetilde{\mathrm{Err}}_{\mathrm{wc}}^{\mathrm{ma}}(\mathbf{1}, \rho) = \begin{cases} \max_{k \neq y} Q\left(\frac{s_y \sqrt{N_y}}{\sqrt{1 + \frac{N_k}{N_y}}}\right) & \rho = \infty, \text{ i.e., } b \text{ fixed to } 0 \\ 1 & \rho < \infty \end{cases} \tag{17}$$

eq. (17) highlights a significant limitation of cross-entropy under class imbalance. Specifically, for any set of signal strengths $\{s_i\}$ we can bring the worst-class error arbitrarily close to $1/2$ simply by increasing the sample size of some class $k$ so that $\frac{N_k}{N_y} \gg s_y^2 N_y$ for some class $y$. Moreover, and perhaps surprisingly, learning a bias parameter (i.e., setting $\rho < \infty$) using cross-entropy results in catastrophic failure (worst class error of 1) in the limit $d \to \infty$; we validate this observation empirically (and explore behavior for finite $d$) in Appendix G.2.

**The failure mechanism of MM.** To gain further insight for why MM can generalize poorly in our overparameterized linear model, consider the expected kernel approximation coefficients $\tilde{\alpha}^{\mathrm{mm}}$ in the limit $d \to \infty$ and without bias learning ($\rho = \infty$). Theorem 4.1 shows that in this case we have $\lim_{d \to \infty} \tilde{\alpha}_{y[k]}^{\mathrm{mm}} = \mathbb{1}_{\{y=k\}} - \frac{1}{c}$. Therefore, the classifier for class $y$ tends to $\tilde{w}_y^{\mathrm{mm}} = \bar{x}_y - \frac{1}{c} \sum_{k \in [c]} \bar{x}_k$, where $\bar{x}_i = \sum_{j \in S_i} x_j$ is the sum of feature vectors with label $i$. When $y$ is a minority class, i.e., $N_y \ll N_k$ for some class $k$, we have $\|\bar{x}_y\| \ll \|\bar{x}_k\|$ and the classifier $\tilde{w}_y^{\mathrm{mm}}$ is dominated by training examples from majority classes, impairing its ability to generalize to new data. In contrast, with MA parameters $\{\delta_i\}$, we obtain (for $\rho = \infty$ and $d \to \infty$) that $\tilde{w}_y^{\mathrm{ma}} = \delta_y \bar{x}_y - \frac{1}{c} \sum_{k \in [c]} \delta_k \bar{x}_k$. Thus, increasing $\delta$ for minority classes lets us compensate for class imbalance.

**Near-optimal $\delta$ values.** In Appendix D.1 we derive a set of margin adjustment parameters that minimize $\widetilde{\mathrm{Err}}_{\mathrm{wc}}^{\mathrm{ma}}(\delta, \rho)$:

$$\delta_i^\star := \frac{\xi_i}{\|\mu_i\|^2 + 2\left(\sum_{j=1}^c \frac{1}{\xi_j} + \rho\right)^{-1}} \in \arg\min_\delta \widetilde{\mathrm{Err}}_{\mathrm{wc}}^{\mathrm{ma}}(\delta, \rho), \tag{18}$$

where we recall that $\xi_i = \|\mu_i\|^2 + \frac{\sigma^2 d}{N_i}$ and note that the minimization above is over $\delta$ as a function of $d$ given the scaling eq. (8). In the limit $d \to \infty$, the value of $\delta_i^\star$ tends to $\frac{1}{N_i}$ for $\rho < \infty$, and becomes proportional to $\frac{1}{s_i N_i}$ at $\rho = \infty$ (Multiplication of $\delta$ by a scalar has no effect on the resulting model). The optimal value of $\widetilde{\mathrm{Err}}_{\mathrm{wc}}^{\mathrm{ma}}$ is

$$\begin{cases} \max_{k \neq y} Q\left(\frac{1}{\sqrt{(s_y^2 N_y)^{-1} + (s_k^2 N_k)^{-1}}}\right) & \rho = \infty \\ \max_{k \neq y} Q\left(\frac{s_y + s_k}{2\sqrt{N_y^{-1} + N_k^{-1}}}\right) & \rho < \infty. \end{cases} \tag{19}$$

We observe a discontinuity in $\rho$ that is due to taking $d \to \infty$; in Appendix G.3 we illustrate how in finite $d$ the transition between the two error values occurs for $\rho$ around $\sqrt{d}$. However, in contrast to MM, learning a bias with carefully-tuned MA is sometimes helpful (see proof in Appendix F.1).

---

[3]In eq. (17), the case $\rho < \infty$ assumes that $N_j \neq N_k$ for some $j, k \in [c]$.

**Proposition 4.2.** *Let $N_1 \leq N_2 \leq \cdots \leq N_c$ and let $\rho < \infty$. If $s_1 \leq s_2 \leq \cdots \leq s_c$ then $\widetilde{\mathrm{Err}}_{\mathrm{wc}}^{\mathrm{ma}}(\delta^\star, \rho) \leq \widetilde{\mathrm{Err}}_{\mathrm{wc}}^{\mathrm{ma}}(\delta^\star, \infty)$, with equality when $s_1 = s_2 = \cdots = s_c$. When $s_1 > s_2 > \cdots > s_c$ then either $\widetilde{\mathrm{Err}}_{\mathrm{wc}}^{\mathrm{ma}}(\delta^\star, \rho) > \widetilde{\mathrm{Err}}_{\mathrm{wc}}^{\mathrm{ma}}(\delta^\star, \infty)$ or $\widetilde{\mathrm{Err}}_{\mathrm{wc}}^{\mathrm{ma}}(\delta^\star, \rho) < \widetilde{\mathrm{Err}}_{\mathrm{wc}}^{\mathrm{ma}}(\delta^\star, \infty)$ is possible.*

### 4.2 Logit adjustment

Recall that LA consists of performing standard cross-entropy training, followed by shifting the bias vector by $-\iota$ for some $\iota \in \mathbb{R}^c$. We analyze LA in our approximation framework by taking the expected score statistics eq. (15) for the MM expected kernel approximation $\tilde{W}^{\mathrm{mm}}, \tilde{b}^{\mathrm{mm}}$ and replacing $\tilde{b}^{\mathrm{mm}}$ with $\tilde{b}^{\mathrm{mm}} - \iota$. We defer the full expression for the resulting approximation to Appendix D.2 and, as in the previous section, focus on approximating the worst-class error in the limit $d \to \infty$ with the scaling eq. (8). To that end, we define

$$\widetilde{\mathrm{Err}}_{\mathrm{wc}}^{\mathrm{la}}(\iota, \rho) \coloneqq \max_{k \neq y} \lim_{d \to \infty} \mathrm{Err}_{y \to k}\big(\tilde{W}^{\mathrm{mm}}(\rho), \tilde{b}^{\mathrm{mm}}(\rho) - \iota\big). \tag{20}$$

**Near-optimal $\iota$ for equal signals.** Let $\xi_i = \|\mu_i\|^2 + \frac{\sigma^2 d}{N_i}$, let $M = \sum_{i \in [c]} \frac{1}{\xi_i}$, and let $M^{\backslash y} = M - \frac{1}{\xi_y}$. We show that taking

$$\iota_y^* = \frac{2 + \|\mu_y\|^2\big[M^{\backslash y} + \rho\big]}{2\xi_y[M + \rho]}, \tag{21}$$

minimizes $\widetilde{\mathrm{Err}}_{\mathrm{wc}}^{\mathrm{la}}(\iota, \rho)$ over $\iota$ for instances where $s_1 = s_2 = \cdots = s_c$ (we use $*$ instead of $\star$ as a superscript to denote this condition).

*Remark* 4.3. Prior work (Menon et al., 2021; Kini et al., 2021) recommend setting $\iota_i \propto \log N_i$. In contrast, for $\rho < \infty$ (i.e., a bias is learned) and at the limit $d \to \infty$, we have that $\iota^* \propto N_i$. We empirically compare these different recommendations in the next section.

Substituting back the $\iota_y^*$ above into our error approximation yields (for general signal strengths),

$$\widetilde{\mathrm{Err}}_{\mathrm{wc}}^{\mathrm{la}}(\iota^*, \rho) = \max_{k \neq y} Q\left( \frac{s_y N_y T_{(k \backslash y)} + s_k N_k T_{(y \backslash k)} - |s_y - s_k| N_y N_k}{2\sqrt{(N_k - N_y)^2 N^{\backslash y, k} + N_y T_{(k \backslash y)}^2 + N_k T_{(y \backslash k)}^2}} \right) \tag{22}$$

where $N^{\backslash y, k} = \sum_{i \neq y, k}^c N_i$ and $T_{(i \backslash j)} = 2N_i + N^{\backslash i, j} + \rho$.

For equal-strength signals, LA dominates MA (see proof in Appendix F.2).

**Proposition 4.4.** *If $s_1 = s_2 = \cdots = s_c$ then $\widetilde{\mathrm{Err}}_{\mathrm{wc}}^{\mathrm{la}}(\iota^*, \rho)$ is monotonic decreasing in $\rho$ and, for all values of $\rho$, we have $\widetilde{\mathrm{Err}}_{\mathrm{wc}}^{\mathrm{la}}(\iota^*, \rho) \leq \widetilde{\mathrm{Err}}_{\mathrm{wc}}^{\mathrm{ma}}(\delta^\star, \rho)$.*

**Discussion** Although Proposition 4.4 is limited to the case of equal strength due to technical reasons, we remain interested in understanding whether the LA predictor (using near-optimal parameters for equal strength signals) is superior to the MA predictor (using near-optimal parameters), when the signal strengths are reversed or aligned. We investigate this question in depth in Appendix G.9. Our findings in Figure 18 show that for aligned signals, it appears that LA (tuned for equal signals) is still uniformly better than MA without bias, but on rare occasions MA with bias outperforms it. For reversed signals, LA is mostly worse than MA with bias, but there are rare exceptions to this. These findings strongly suggest that LA is uniformly better than MA without bias for aligned signals, in line with our additional experiments in Section 5. Moreover, we see that the converse is not true. For reversed signals MA is not always superior to LA, even when the latter is sub-optimally tuned.

### 4.3 Class dependent temperature

Finally, we consider CDT, where for simplicity we focus on the bias-free setting ($\rho = \infty$). First, we apply the expected kernel approximation and find the CDT coefficients in closed form (See Appendix C.3 for proof).

**Theorem 4.5.** *Let $\tilde{W}^{\mathrm{cdt}}$ be the expected kernel approximation for the CDT predictor eq. (6) with any $\delta$ that satisfies $\delta_i > 0$ for all $i \in [c]$, using $\rho = \infty$. Then, for all $y \in [c]$ we have $\tilde{w}_y^{\mathrm{cdt}} = \sum_{i=1}^{c} \tilde{\alpha}_{y[i]}^{\mathrm{cdt}} \bar{x}_i$, where (for $\xi_i = \|\mu_i\|^2 + \frac{\sigma^2 d}{N_i}$),*

$$\tilde{\alpha}_{y[i]}^{\mathrm{cdt}} = \frac{\delta_i}{N_i \xi_i} \left( \mathbb{1}_{\{i=y\}} - \frac{\delta_y \delta_i}{\sum_{j=1}^{c} \delta_j^2} \right).$$

Comparing Theorem 4.5 to Theorem 4.1 (with $\rho = 0$), we see that the only difference between $\tilde{\alpha}_y^{\mathrm{ma}}$ and $\tilde{\alpha}_y^{\mathrm{cdt}}$ is a factor of $\frac{1}{c}\mathbf{1}$ that turns into $\frac{\delta_y}{\sum_{j=1}^{c} \delta_j^2}\delta$. Unfortunately, following through our approximation procedure reveals that CDT, with any $\delta_i > 0$, may fail to mitigate class imbalance, which we formalize in the following (see proof in Appendix F.4).

**Proposition 4.6.** *For any $\epsilon > 0$ and $c \geq 3$ classes, there exists an instance $\{s_i, N_i\}_{i=1}^{c}$ for which*

$$\lim_{d \to \infty} \mathrm{Err}_{\mathrm{wc}}\left(\tilde{W}^{\mathrm{cdt}}, 0\right) \geq \frac{1}{2} - \epsilon \quad while \quad \lim_{d \to \infty} \mathrm{Err}_{\mathrm{wc}}\left(\tilde{W}^{\mathrm{ma}}(\delta^\star, \infty), 0\right) \leq \epsilon.$$

The proof of Proposition 4.6 relies on instances with very strong class imbalance; in the next section we empirically show that CDT fails in such settings.

## 5    Empirical validation

To test our observations both within and beyond the scope of our model, we compare CDT, MA, and LA in three different setups. First, we test linear classification on data generated from our Gaussian model (Section 2.3). Second, we consider classification with an RBF kernel over artificially imbalanced CIFAR10 data (Krizhevsky et al., 2009; Cao et al., 2019; Cui et al., 2019) using features extracted from a CLIP model (Radford et al., 2021). Finally, using the same data, we compare MA, CDT and (post-hoc) LA when fine-tuning the CLIP model. We report full experimental details and provide results on additional problem instances in Appendix G.

**Loss hyperparameter tuning.** We tune each method by sweeping over a scalar hyperparameter. In the first experiment, we tune MA and LA using our theoretical expressions from Section 4, setting $\delta_i = (\delta_i^\star)^\gamma$ and $\iota_i = \tau \iota_i^*$, respectively, and sweeping over $\gamma$ and $\tau$ such that value 1 recovers the theoretical tuning and value 0 reduces to standard maximum margin. In the second and third experiments we assume equal signals and, following our theory, set $\delta_i = N_i^{-\gamma}$ for MA and $\iota_i = \tau N_i/N$ for LA; in the third experiment we also test the standard setting $\iota_i = \tau \log N_i$. For CDT (where our theory does not find good hyperparameters) we set $\delta_i = N_i^{-\gamma}$ as is standard.

**Linear classification with Gaussian model.** We begin by simulating three instances of our theoretical model (for $d = 10^5$), with signal strengths that are either (a) aligned with class sample size, (b) the same for all classes, or (c) inversely aligned with class sample size. Figure 2 shows excellent agreement between theory and practice, with our approximation tightly matching empirical performance and our key predictions upheld. Namely, our theoretically-derived tuning parameters ($\gamma, \tau = 1$) are optimal; for aligned signals, LA is better than MA with bias, which is in turn better than MA; for reversed signals, the opposite is true. Additionally, both CDT and MM (corresponding to $\gamma, \tau = 0$) can fail badly, with MM and $\rho < \infty$ obtaining worst class error 1 as predicted. In addition to the worst-class error studied so far, Figure 2 also displays our analytical approximation and empirical measurements for class-balanced error and the macro $F_1$ score. Here too we see excellent agreement between the approximation and measurements. Moreover, the best tuning parameters for worst-class perform well under balanced error and macro $F_1$ score as well. See Appendix G.6 for more details.

**Classification with RBF kernel.** To test our predictions beyond the setting of our model, we perform RBF kernel classification on imbalanced CIFAR10 data featurized using the ViT-B/32 CLIP model (Radford et al., 2021). We consider two class size profiles. The first is a standard "long-tailed" exponential profile (Cui et al., 2019; Cao et al., 2019) with 500 examples for the majority class and 5 examples for the minority class. The second has the same majority and minority sizes, but we increase the sample size of other classes to

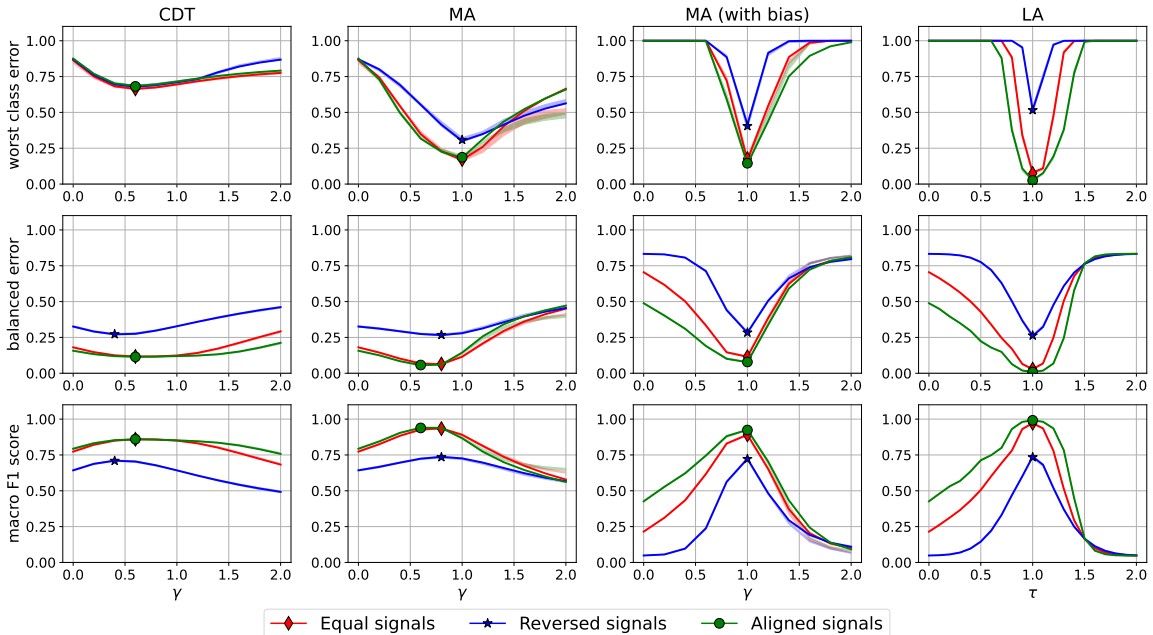

**Figure 2.** Worst class error, balanced error, and macro $F_1$ score vs. loss tuning hyperparameter for the different methods we consider and **linear classification** on synthetic data from our model. The shaded region shows empirical results (two standard deviations over 5 random seeds), and the solid lines show our analytical approximation.

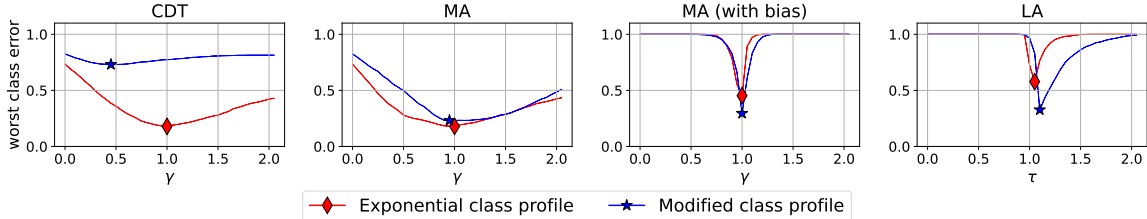

**Figure 3.** Empirical worst class test error vs. loss tuning hyperparameter for the different methods we consider and **kernel classification** on two imbalanced versions of CIFAR10. The markers show the minimum error for each instance.

make the theoretically-predicted CDT worse. Figure 3 shows that many of our predictions continue to hold for this setting, most notably the high errors of CDT and MM (particularly with bias) and optimal tuning around $\gamma, \tau = 1$. Similar experiments on MNIST and FashionMNIST datasets yield results that are mostly aligning with our observations on CIFAR10, see in Appendix G.4.

**CLIP fine-tuning.** Taking a step further, we experiment with end-to-end fine-tuning of the ViT-B/32 CLIP model (with a zero-shot initialization), on imbalanced subsets of CIFAR10 with different numbers of classes, where a smaller number of classes allows sharper class imbalance. Figure 4 shows only partial agreement with our theory, as CDT performs comparably to MA, and our theoretical setting of LA is slightly better at the 10 class instance but worse at the other two. Nevertheless we note that CDT is much more sensitive to the tuning of $\gamma$ than MA, which is opposite to the conclusions of Behnia et al. (2023). We also observe that CDT does not converge to zero training error (see Appendix G.5), corroborating Remark 2.1.

# 6 Discussion

In this section, we provide an extended discussion of prior work on the implicit bias of gradient methods, loss functions for class imbalance, and robustness to spurious correlations. We conclude by reviewing the limitations of our study and outlining directions for future work to extend our model.

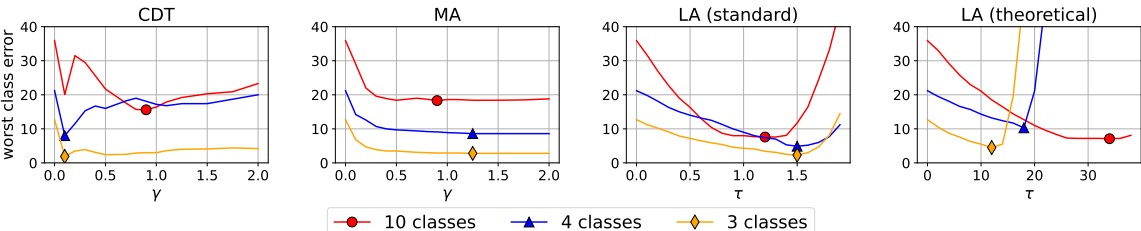

**Figure 4.** Empirical worst class test error vs. loss tuning hyperparameter for the different methods we consider and **neural network fine-tuning** on class-imbalanced subsets of CIFAR10.

**The implicit bias of gradient methods on separable training data.** The implicit bias of gradient descent when applied to certain loss functions and separable training data plays a central role in our study. Rosset et al. (2003) show that the regularization path of a family of loss functions (including cross-entropy) converges in direction to maximum margin separator in the limit of vanishing regularization. Later, Soudry et al. (2018) show that the same limit holds for gradient descent without any regularization, characterizing its implicit bias. Subsequent work generalizes these results to stochastic gradient methods (Nacson et al., 2019) and shows that the equivalence between regularization path limit and implicit bias holds to a high degree of generality (Ji et al., 2020). We build on these results, as well as an analysis of CDT by Kini et al. (2021), to tie the different loss functions we consider to margin maximization problems, as explained in Section 2.2. We note that the connection between gradient descent and margin regularization extends to neural network models (Lyu & Li, 2020), though there it is more involved (Vardi et al., 2022).

**Properties of margin maximizers in high dimension.** Our expected kernel approximation (Section 3.1) effectively assumes that all data-points are support vectors, with weight determined only by their labels. While this assumption may seem strong, the resulting approximate test error provides a tight fit to experiments even at moderate dimension (see Figure 1). Moreover, the realization that in high-dimensional Gaussian models all data points become support vectors is more or less apparent in a number of prior works. Sagawa et al. (2020b) and Wald et al. (2022) implicitly use this property in order to outline a robustness limitation of learned data separators. Glasgow et al. (2023) leverage a similar property to analyze generalization error of different data separators. Frei et al. (2023a;b) generalize this property to nonlinear classification of linearly separable training data. Finally, Wang et al. (2021) give explicit conditions under which all examples are support vectors, instantiating them for a Gaussian mixture model very similar to the one we study, except that the model they consider is class-balanced and has uniform signal strengths.

**Losses for imbalanced data.** There is extensive literature proposing and analyzing loss functions for class-imbalanced data, which we now briefly review. There are also many additional approaches to counter class-imbalance such as specialized model architectures, data augmentation, and ensembling, but they fall outside our scope; see (Zhang et al., 2023) for a recent survey.

As previously mentioned, techniques based on upweighting or oversampling minority classes (He & Garcia, 2009; Cui et al., 2019) often have limited success when combined with overparameterized models since the implicit bias of gradient descent on the cross-entropy loss is insensitive to these operations (Soudry et al., 2018; Byrd & Lipton, 2019; Lyu & Li, 2020; Zhai et al., 2023; Xu et al., 2021). The focal loss (Lin et al., 2017) attempts to down-weight easy examples and focus the model on challenging instances. However, it admits the same implicit bias as cross-entropy since it is not class-dependent and has an exponential tail. SMOTE by Chawla et al. (2002) can potentially get around this limitation by oversampling synthetic minority examples, though in separable linear classification the augmentation proposed in (Chawla et al., 2002) will not affect the implicit bias of gradient descent on cross-entropy.

In contrast to oversampling minority classes, undersampling majority classes (He & Garcia, 2009; Buda et al., 2018) is an effective class balancing intervention even in the overparameterized setting (Byrd & Lipton, 2019), but has the obvious disadvantage of ignoring part of the training data. Our analysis clearly demonstrates the benefits and costs of undersampling: for maximum margin classification (i.e., the implicit bias of cross-entropy minimized with gradient descent), the approximation equation 17 for the worst class error decreases

when decreasing the sample size of the majority class. In contrast, for well-tuned margin adjustment, our approximation equation 19 suggests that decreasing the sample size never helps.

Several prior works design losses to directly address classification margins. The label distribution aware margin (LDAM) loss (Cao et al., 2019) adds a class-dependent offset to the logit of the correct label, while Menon et al. (2021) propose the LA loss that offsets the logits of all labels and is *Fisher consistent*, i.e., given infinite data, minimizing it also minimizes the class-balanced error. However, when model weights grow to infinity (as they typically do in unconstrained overparameterized classification), either type of logit offset becomes negligible and the learned model again degenerates to a maximum margin separator. Therefore, in our work, we study a post-hoc variant of LA. In the face detection literature, losses that combine logit normalization and margin adjustment have proven successful (Wang et al., 2018; Deng et al., 2019); we believe exploring the connection between such losses and margin adjustment (MA) (Behnia et al., 2023; Wang et al., 2022; Nguyen et al., 2021) is an interesting topic for future work.

Ye et al. (2020) propose the class dependent temperature (CDT) loss to compensate for a minority feature deviation phenomenon, also observed in (Kang et al., 2020). Our analysis suggests that while CDT is sometimes effective at mitigating class imbalance, it also has significant limitations. Kini et al. (2021) propose combining the LA and CDT losses into a single loss called vector scaling (VS) loss and provides a theoretical analysis for the binary case. However, the binary classification loss they analyze is a two-class version of MA, whereas two-class CDT is always equivalent to maximum margin (Behnia et al., 2023). Li et al. (2021) suggest learning the parameters of VS-loss via bi-level optimization.

Finally, Kang et al. (2020) show that losses that yield good classifiers under class imbalance do not necessarily learn good representations, and (Samuel & Chechik, 2021; Jitkrittum et al., 2022) propose losses that directly target representation learning. Our analysis deals with linear classification and therefore has no direct implications for representation learning. Nevertheless, Kang et al. (2020) show that separately learning the representation and classifier is often effective, and our result can provide guidance for the latter part.

**Neural and minority collapse.** Building on the notion of neural collapse (Papyan et al., 2020), Fang et al. (2021) and Ji et al. (2021) introduce a "layer peeled model" for neural network training where they identify a "minority collapse" phenomenon, which potentially explains the tendency of neural networks to generalize poorly to minority classes. Lu et al. (2022); Behnia et al. (2023) show that MA and CDT can prevent minority collapse.

There are three differences between our results and (Fang et al., 2021; Lu et al., 2022; Behnia et al., 2023). First, our analysis focuses on a linear setting where the representation is fixed rather than part of the optimization objective. Second, we consider multi-class classification under general class size distribution whereas (Fang et al., 2021; Lu et al., 2022; Behnia et al., 2023) consider a special case where there are only two class sizes. Finally—and most importantly—we directly analyze the test performance of different balanced losses, while finding a direct link between neural collapse and generalization is an open problem.

**Robustness to spurious correlations.** The problem of learning under spurious correlations (i.e., signals that appear in part of the training data but not in the test data) has several parallels to class-imbalanced learning. In particular, cross-entropy minimization tends to rely on spurious features, and re-weighting techniques are not effective in the overparameterized setting (Sagawa et al., 2020a). Furthermore, loss functions proposed for addressing class imbalance are typically also relevant for balancing different sub-populations in the training data (Kini et al., 2021; Lu et al., 2022) and vice versa (Wang et al., 2022; Nguyen et al., 2021). Finally, a number of works use Gaussian models similar to the one we study in order to shed light on how spurious correlations affects generalization (Sagawa et al., 2020a; Nagarajan et al., 2021; Wald et al., 2022).

However, class imbalance is also fundamentally different from spurious correlation. The former is ubiquitous in practice while the latter is often challenging to define, identify and address (Hendrycks et al., 2021; Gulrajani & Lopez-Paz, 2021). Moreover, for overparameterized Gaussian mixture models there exists an information-theoretic barrier that prevents any interpolating method (such as MA) from generalizing well (Wald et al., 2022); our results indicate such a barrier does not exist in the class-imbalanced setting.

**Limitations and future work.** The main limitations of our work is the assumption of a very specific data distribution and (to a lesser extent) the focus on linear classification. Our preliminary experiments suggest that our theoretical predictions translate well to some non-linear, real-data kernel classification problems, but may be less directly applicable to neural network training.

We identify two potential reasons for this discrepancy. First, our neural network experiments do not involve very long training, while our analysis pertains to solutions obtain after infinitely many steps. Even in our simple Gaussian model, understanding the impact of shorter training, early stopping, and regularization is a fascinating topic for future work. Second, prior work observes that losses leading to good classifiers for imbalanced data do not necessarily induce good neural representations, and suggest approaching each task separately (Kang et al., 2020), while Behnia et al. (2023) argues that some losses can produce both good representations and good classifiers in the context of class imbalance. Consequently, future work can apply our approximation method for analyzing classifier-learning methods, or extend our model in order to also capture representation learning.

## 7 Acknowledgments

The authors acknowledge support from the Israeli Science Foundation (ISF) grant no. 2486/21 and the Alon Fellowship.

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

# Appendix

## A Approximation method

### A.1 Notation

**Training set distribution.** We let

$$\mathcal{P}_d\left(\{s_i, N_i\}_{i=1}^c\right)$$

denote the distribution of a training set of size $N = \sum_{i \in [c]} N_i$ containing (for each $y \in [c]$) $N_y$ examples from class $y$ drawn i.i.d. from eq. (7), with model parameters given by eq. (8). For dataset $\{(x_i, y_i)\}_{i \in [N]} \sim \mathcal{P}_d\left(\{s_i, N_i\}_{i=1}^c\right)$, we let $S_k := \{i \mid y_i = k\}$ be the set of indices for training examples belonging to class $k$.

### A.2 Expected kernel approximation

In this section we justify the use of $\mathbb{E}K$ instead of $K$ at the limit of $d \to \infty$. Recall the maximum margin predictor problem defined by:

$$\beta^{\mathrm{mm}}, b^{\mathrm{mm}} := \underset{\beta_1,\ldots,\beta_1 \in \mathbb{R}^N, b \in \mathbb{R}^c}{\arg\min} \left(\sum_{i=1}^c \beta_i^\top K \beta_i + \rho b_{[i]}^2\right)$$

$$\text{subject to} \quad (\beta_{y_i} - \beta_k)^\top k_i + b_{[y_i]} - b_{[k]} \geq 1 \quad \text{for all} \quad i \leq N \text{ and } k \neq y_i. \tag{23}$$

**Lemma A.1.** *Given a dataset $\mathcal{D} = \{(x_i, y_i)\}_{i=1}^N$, let $X \in \mathbb{R}^{N \times d}$ be the matrix of the features $X = [x_1, \ldots, x_N]^\top$, $M = [\mu_{y_i}, \ldots, \mu_{y_N}]^\top$ be the matrix of the corresponding signal vectors of each data point in $X$, and $G = X - M$. Then for every $t > 0$ with probability at least $1 - 4(c+1)\exp(-t^2/8)$ it holds that*

$$1 - \sqrt{\frac{N}{d}} - \frac{t}{\sqrt{d}} \leq s_{\min}\left(G^\top\right) \leq s_{\max}\left(G^\top\right) \leq 1 + \sqrt{\frac{N}{d}} + \frac{t}{\sqrt{d}} \tag{24}$$

$$\forall i \in [c] \; \|G\mu_i\| \leq t\sqrt{\frac{N}{d}}\|\mu_i\| \tag{25}$$

Lemma A.1 was largely derived from Wald et al. (2022), but in our case we obtain a different probability factor for the event to occur due to the union bound over $c+1$ events. For the reader's convenience, we have included the proof again here.

*Proof.* $G$ is a random Gaussian matrix with $G_{[i,j]} \sim \mathcal{N}\left(0, d^{-1}\right)$, hence from Roman (2012, Cor 5.35) it holds that with probability at least $1 - 2\exp(-t^2/2)$ eq. (24) holds. In addition, since $G$ is random Gaussian matrix, it holds that $G\mu_i \sim \mathcal{N}\left(0, d^{-1}\|\mu_i\|^2 I_N\right)$. We use a bound on a norm of Gaussian vectors (Ledoux & Talagrand, 2013, eq 3.5) and get that for any $t_i > 0$,

$$\mathbb{P}[\|G\mu_i\| > t_i] \leq 4\exp\left(-\frac{dt_i^2}{8N\|\mu_i\|^2}\right).$$

Specifically, for $t_i = t\sqrt{\frac{N_i}{d}}\|\mu_i\|$ it holds that with probability at least $1 - 4\exp(-t^2/8)$ eq. (25) holds. Taking the union bound over $c+1$ events and using the fact that $2\exp(\frac{-t^2}{2}) < 4\exp(\frac{-t^2}{8})$ for any $t > 0$ results in proving the desired lemma. $\square$

**Lemma A.2.** *(Concentration of features kernel)* *Given a dataset $\mathcal{D} \sim \mathcal{P}_d \left( \{s_i, N_i\}_{i=1}^c \right)$, let $X \in \mathbb{R}^{N \times d}$ be the matrix of the features $X = [x_1, \ldots, x_N]^\top$ and $K = XX^\top$ be the features kernel of $X$. Then for any $t > 0$, with probability at least $1 - 4(c+1)\exp(-t^2/8)$, it holds that*

$$\|K - \mathbb{E}[K]\|_{\mathrm{op}} \le 2 \left( \frac{\sqrt{N}+t}{\sqrt{d}} + \frac{(\sqrt{N}+t)^2}{d} + \frac{tN}{d^{3/4}} \max_{i \in [c]} s_i \right).$$

*Therefore for any dataset $\mathcal{D} \sim \mathcal{P}_d \left( \{s_i, N_i\}_{i=1}^c \right)$, let $K$ bet the kernel matrix of the feature vectors in $\mathcal{D}$ then, $\lim_{d \to \infty} \mathbb{P}[\|K - \mathbb{E}[K]\|_{\mathrm{op}} = 0] = 1$.*

*Proof.* Similarly to Lemma A.1 we can view $X = G + M$ where $G$ is a random Gaussian matrix with $G_{[i,j]} \sim \mathcal{N}\left(0, d^{-1}\right)$. Since $GG^\top \sim \mathcal{W}\left(d^{-1}I, d\right)$ it holds that $\mathbb{E}\left[GG^\top\right] = I$, then from eq. (24) it holds that

$$\|GG^\top - \mathbb{E}\left[GG^\top\right]\|_{\mathrm{op}} \le \left(1 + \sqrt{\frac{N}{d}} + \frac{t}{\sqrt{d}}\right)^2 - 1 \tag{26}$$

Combining this and eq. (25) we get

$$
\begin{aligned}
\|K - \mathbb{E}[K]\|_{\mathrm{op}} &\le \|GG^\top - \mathbb{E}\left[GG^\top\right]\|_{\mathrm{op}} + 2\|GM^\top\|_{\mathrm{op}} \\
&\overset{(a)}{\le} \left(1 + \sqrt{\frac{N}{d}} + \frac{t}{\sqrt{d}}\right)^2 - 1 + 2 \max_{x:\|x\|=1} \|GM^T x\| \\
&= \left(2\frac{\sqrt{N}+t}{\sqrt{d}} + \frac{(\sqrt{N}+t)^2}{d}\right) + 2 \max_{x:\|x\|=1} \|GM^T x\| \\
&= \left(2\frac{\sqrt{N}+t}{\sqrt{d}} + \frac{(\sqrt{N}+t)^2}{d}\right) + 2 \max_{x:\|x\|=1} \sqrt{x^\top M G^\top G M^\top x} \\
&= \left(2\frac{\sqrt{N}+t}{\sqrt{d}} + \frac{(\sqrt{N}+t)^2}{d}\right) + 2 \max_{x:\|x\|=1} \sqrt{x^\top \sum_{i,j} \left(G\mu_{y_i}^\top\right)^\top \left(G\mu_{y_j}^\top\right) x} \\
&\le \left(2\frac{\sqrt{N}+t}{\sqrt{d}} + \frac{(\sqrt{N}+t)^2}{d}\right) + 2 \max_{x:\|x\|=1} \sqrt{\max_{i \in [c]} \|G\mu_i^\top\|^2 \|x\|^2} \\
&= \left(2\frac{\sqrt{N}+t}{\sqrt{d}} + \frac{(\sqrt{N}+t)^2}{d}\right) + 2 \max_{i \in [c]} \|G\mu_i^\top\| \sqrt{N} \\
&\overset{(b)}{\le} 2 \left(\frac{\sqrt{N}+t}{\sqrt{d}} + \frac{(\sqrt{N}+t)^2}{d} + \frac{tN}{\sqrt{d}} \max_{i \in [c]} \|\mu_i\|\right)
\end{aligned}
$$

where transitions (a) and (b) are justified by Lemma A.1, which leads us to conclude the desired bound. $\qquad\square$

Essentially Lemma A.2 show that under our data assumption, Section 2.3, at the limit of $d \to \infty$ $K$, converge to $\mathbb{E}K$. This motivates us to substitute $K$ by $\mathbb{E}K$ and derive the margin problem in the overparameterized regime. Specifically the approximated maximum margin predictor problem is defined by

$$
\begin{aligned}
\beta^{\mathrm{mm}}, b^{\mathrm{mm}} &:= \underset{\beta_1, \ldots, \beta_c \in \mathbb{R}^N, b \in \mathbb{R}^c}{\arg\min} \left( \sum_{i=1}^c \beta_i^\top \mathbb{E}[K]\beta_i + \rho b_{[i]}^2 \right) \\
&\text{subject to} \quad (\beta_{y_i} - \beta_k)^\top \mathbb{E}[k_i] + b_{[y_i]} - b_{[k]} \ge 1 \quad \text{for all} \ \ i \le N \ \text{and} \ k \ne y_i.
\end{aligned} \tag{27}
$$

We continue by explaining why this substitution helps to simplify the optimization in eq. (23).

**Proposition A.3.** *For any pair of coefficients $\beta_{y[z]}^{\mathrm{mm}}, \beta_{y[k]}^{\mathrm{mm}}$ associated with points belonging to the same class, the optimal solution of Equation 27 guarantees that $\beta_{y[z]}^{\mathrm{mm}}$ is equal to $\beta_{y[k]}^{\mathrm{mm}}$.*

*Proof.* Let $\mathcal{D}_1 = \{(x_i, y_i)\}_{i=1}^N$ be a linearly separable dataset in $\mathbb{R}^d$ such that for any $i \in [N]$

$$x_i = \mu_{y_i} + n_i \quad \text{where} \quad n_{[i]} \sim \mathcal{N}\left(0, \sigma^2 I_d\right).$$

Let $\mathcal{D}_2$ be a dataset that is composed of the data points in $\mathcal{D}_1$ with a different permutation of the data points (within the same class). Additionaly, denote by $K^{(i)}$ be the kernel matrix of $\mathcal{D}_i$ and notice that:

$$\mathbb{E}\left[K^{(1)}\right] = \mathbb{E}\left[K^{(2)}\right].$$

Which means that the optimal solutions of eq. (27) under $\mathcal{D}_1$ and $\mathcal{D}_2$ are identical. Hence, for any pair of coefficients $k \neq z$ of the same class, it holds that $\beta_{y[k]}^{\mathrm{mm}} = \beta_{y[z]}^{\mathrm{mm}}$. □

Proposition A.3 shows that eq. (27) actually involve only $c \times (c+1)$ parameters (including the bias) under $c \times (c-1)$ constraints instead of $c \times N + c$ parameters with $N \times (c-1)$ constraints. This allows us to simplify the optimization in eq. (27) and conclude our final optimization problem:

$$\tilde{\alpha}^{\mathrm{mm}}, \tilde{b}^{\mathrm{mm}} := \underset{\tilde{\alpha}_1, \dots, \tilde{\alpha}_c \in \mathbb{R}^c, b \in \mathbb{R}^c}{\arg\min} \left( \sum_{i=1}^c \tilde{\alpha}_i^\top \bar{K} \tilde{\alpha}_i + \rho b_{[i]}^2 \right)$$

$$\text{subject to} \quad (\tilde{\alpha}_y - \tilde{\alpha}_k)^\top \frac{\bar{K}_y}{N_y} + b_{[y]} - b_{[k]} \geq 1 \quad \text{for all} \quad y \in [c] \text{ and } k \neq y \tag{28}$$

where $\bar{K}$ is the expectation of the features sum vectors kernel.

$$\bar{K}_{[i,j]} = \begin{cases} N_i^2 \xi_i & i = j \\ 0 & i \neq j \end{cases}$$

Specifically, denote $\bar{x}_i := \sum_{j \in S_i} x_j$ and $\xi_i = \|\mu_i\|^2 + \frac{\sigma^2 d}{N_i}$ then,

$$\bar{K}_{[i,i]} = \mathbb{E}[\langle \bar{x}_i, \bar{x}_i \rangle] = \mathbb{E}\left[\left\langle \sum_{j_1 \in S_i} x_{j_1}, \sum_{j_2 \in S_i} x_{j_2} \right\rangle\right]$$

$$= \mathbb{E}\left[\sum_{j_1 \in S_i} \sum_{j_2 \in S_i} \|\mu_i\|^2 + \sum_{j_1 \in S_i} N_i\langle n_{j_1}, \mu_y \rangle + \sum_{j_2 \in S_i} N_i\langle n_{j_2}, \mu_y \rangle + \sum_{j_1 \in S_i} \sum_{j_2 \in S_i} \langle n_{j_1}, n_{j_2} \rangle\right]$$

$$\overset{(1)}{=} N_i^2 \|\mu_i\|^2 + \mathbb{E}\left[\sum_{j_1 \in S_i} \sum_{j_2 \in S_i} \langle n_{j_1}, n_{j_2} \rangle\right] \overset{(2)}{=} N_i^2 \|\mu_i\|^2 + N_i \sigma^2 d = N_i^2 \xi_i$$

were (1) and (2) are justified by linearity of expectation and i.i.d random samples of gaussian noise. In addition $\bar{K}_{[i,j]} = 0$ where $i \neq j$ as the product of two i.i.d random variables with orthogonal signal component.

## A.3 Prediction error parameter approximation

**Lemma A.4.** *Let $x$ and $y$ be two independent vectors drawn from $\mathcal{N}(0, I_d)$. Then, for any $d \geq 2$, fixed $u \in \mathbb{R}^d$ and $\delta \in \left(0, \frac{1}{e}\right)$ we have*

$$\mathbb{P}\left(|\langle x, u \rangle| > \|u\| \sqrt{2 \log \frac{1}{\delta}}\right) \leq \delta,$$

$$\mathbb{P}\left(|\|x\|^2 - d| > 4\sqrt{d} \log \frac{2}{\delta}\right) \leq \delta, \text{ and}$$

$$\mathbb{P}\left(|\langle x, y \rangle| > \sqrt{d} \log \frac{4}{\delta}\right) \leq \delta.$$

Note that the last two inequalities sacrifice mid-tail dependence (i.e., depending on $\log \frac{1}{\delta}$ instead of $\sqrt{\log \frac{1}{\delta}}$ for values of $\delta$ above $e^{-\sqrt{d}}$) in favor of cleaner expressions.

*Proof.* For the first bound we note that $\langle x, u \rangle \sim \mathcal{N}\left(0, \|u\|^2\right)$ and therefore $\mathbb{P}\left(|\langle x, u \rangle| > t\right) = 2Q\left(\frac{t}{\|u\|}\right) \leq$ $e^{-\frac{t^2}{2\|u\|^2}}$. The second bound follows from substituting $a_i = 1$ and $x = \log \frac{2}{\delta}$ in Lemma 1 of Laurent & Massart (2000), which yields

$$\mathbb{P}\left(\left|\|x\|^2 - d > 2\right|\sqrt{d \log \frac{2}{\delta}} + 2\log \frac{2}{\delta}\right) \leq \delta,$$

and we note that, since $d$ and $\log \frac{1}{\delta}$ are both larger than 1, we have

$$2\sqrt{d \log \frac{1}{\delta}} + 2\log \frac{1}{\delta} \leq 2\sqrt{d}\log \frac{1}{\delta} + 2\sqrt{d}\log \frac{1}{\delta} = 4\sqrt{d}\log \frac{1}{\delta}.$$

Let us briefly derive the final bound from first principles. First, let $k$ and $z$ be independent standard Gaussians. Then, for all $|\lambda| < 1$,

$$\mathbb{E}e^{\lambda k z} = \mathbb{E}\left[\mathbb{E}\left(e^{\lambda k z}\right) \mid z\right] = \mathbb{E}e^{\frac{\lambda^2}{2} z} = \int_{-\infty}^{\infty} \frac{1}{\sqrt{2\pi}} e^{-\frac{z^2}{2}\left(1 - \lambda^2\right)} dz = \sqrt{\frac{1}{1 - \lambda^2}}.$$

Applying a Chernoff bound, we have that for all $\lambda \in (0, 1)$,

$$\mathbb{P}\left(\langle x, y \rangle > t\right) \leq e^{\lambda\left(\langle x, y \rangle - t\right)} = e^{-\lambda t}\left(\mathbb{E}e^{\lambda k z}\right)^d = e^{-\frac{d}{2}\log\left(1 - \lambda^2\right) - \lambda t}.$$

Now pick $\lambda = \frac{1}{\sqrt{d}}$, and note that $-d \log\left(1 - \frac{1}{d}\right)$ is decreasing in $d$, and therefore

$$-\frac{d}{2}\log\left(1 - \frac{1}{d}^2\right) \leq \log 2.$$

for every $d \geq 2$. Therefore, taking $t = \sqrt{d}\log \frac{4}{\delta} = \frac{1}{\lambda}\log \frac{4}{\delta}$ we obtain

$$\mathbb{P}\left(\langle x, y \rangle > \sqrt{d}\log \frac{4}{\delta}\right) \leq e^{\log 2 - \log \frac{4}{\delta}} = \frac{\delta}{2}.$$

Since $\langle x, y \rangle$ and $-\langle x, y \rangle$ are identically distributed, we also have $\mathbb{P}\left(\langle x, y \rangle < \sqrt{d}\log \frac{4}{\delta}\right) \leq \frac{\delta}{2}$, yielding the claimed bound on $|\langle x, y \rangle|$. $\qquad \square$

**Lemma A.5.** *(Concentration of $\tilde{\nu}^{(y)}$ and $\tilde{\Sigma}^{(y)}$) Let $\tilde{W}$, $\tilde{b}$ be an approximate predictor of the form eq. (11) under data assumptions (Section 2.3). Then for any $\delta \in (0, \frac{1}{e})$ w.p $1 - \delta$ it holds that*

$$\left|\tilde{\nu}_{[i]}^{(y)} - \hat{\nu}_{[i]}^{(y)}\right| < \frac{1}{\sqrt[4]{d}}\left|\sum_{j=1}^{c} u_{[i,j]}^{(y)}\right|\sqrt{2N_j s_y \log\left(\frac{1.5c(c+1)}{\delta}\right)},$$

*and*

$$\left|\tilde{\Sigma}_{[i,j]}^{(y)} - \hat{\Sigma}_{[i,j]}^{(y)}\right| \leq 4\frac{N_{\max}^{3/2}\left(\sqrt{s_{\max}} + 1\right)}{\sqrt{d}}\sum_{z=1}^{c}\sum_{k=1}^{c}\left|u_{[i,z]}^{(y)} u_{[j,k]}^{(y)}\right|\log\left(\frac{6c(c+1)}{\delta}\right)$$

*where*

$$\hat{\nu}_{[i]}^{(y)} = \frac{u_{[i,y]}^{(y)} N_y \|\mu_y\|^2 + b_{[y]} - b_{[i]}}{\sigma}, \quad \hat{\Sigma}_{[i,j]}^{(y)} = \sum_{z=1}^{c} u_{[i,z]}^{(y)} u_{[j,z]}^{(y)} N_z^2 \xi_z \quad and \quad u_{[i,j]}^{(y)} := \alpha_{y[j]} - \alpha_{i[j]} \qquad (29)$$

*Proof.* We start by showing that $\mathbb{E}\big[\tilde{\nu}^{(y)}\big] = \hat{\nu}^{(y)}$ and $\mathbb{E}\big[\tilde{\Sigma}^{(y)}\big] = \hat{\Sigma}^{(y)}$ then we continue by deriving the concentration bound for each parameter and use the union bound the get the desired result.

Recall in the define of $\tilde{\nu}^{(y)}_{[i]}$:

$$\frac{\sum_{j=1}^{c}\big[\alpha_{y[j]} - \alpha_{i[j]}\big]\langle \bar{x}_j, \mu_y\rangle + b_{[y]} - b_{[i]}}{\sigma} = \frac{\sum_{j=1}^{c} u^{(y)}_{[i,j]}\langle \bar{x}_j, \mu_y\rangle + b_{[y]} - b_{[i]}}{\sigma}.$$

Hence,

$$\mathbb{E}\Big[\tilde{\nu}^{(y)}_{[i]}\Big] = \mathbb{E}\left[\frac{\sum_{j=1}^{c} u^{(y)}_{[i,j]}\langle \bar{x}_j, \mu_y\rangle + b_{[y]} - b_{[i]}}{\sigma}\right]$$

$$\overset{(1)}{=} \frac{1}{\sigma}\sum_{j=1}^{c} u^{(y)}_{[i,j]}\mathbb{E}[\langle N_j\mu_j + \bar{n}_j, \mu_y\rangle] + \frac{b_{[y]} - b_{[i]}}{\sigma}$$

$$\overset{(2)}{=} \frac{u^{(y)}_{[i,y]}N_y\|\mu_y\|^2 + b_{[y]} - b_{[i]}}{\sigma} = \hat{\nu}^{(y)}_{[i]}$$

where (1) holds by definition of $\bar{x}_i$ and since the biases are fixed, (2) holds since $\langle \mu_i, \mu_j\rangle = \mathbb{1}_{\{i=j\}}$ and since the expectation of the noise vectors is zero.

Similarly we calculate the expectation of $\tilde{\Sigma}^{(y)}_{[i,j]}$. Recall in the definition of $\tilde{\Sigma}^{(y)}_{[i,j]}$

$$\tilde{\Sigma}^{([i,j])} = \sum_{z=1}^{c}\sum_{k=1}^{c}\big[\alpha_{y[z]} - \alpha_{i[z]}\big]\big[\alpha_{y[k]} - \alpha_{j[k]}\big]\langle \bar{x}_i, \bar{x}_j\rangle$$

$$= \sum_{z=1}^{c}\sum_{k=1}^{c} u^{(y)}_{[i,z]}u^{(y)}_{[j,k]}\langle \bar{x}_i, \bar{x}_j\rangle.$$

and the expectation is

$$\mathbb{E}\Big[\tilde{\Sigma}^{([i,j])}\Big] = \mathbb{E}\left[\sum_{z=1}^{c}\sum_{k=1}^{c} u^{(y)}_{[i,z]}u^{(y)}_{[j,k]}\langle \bar{x}_z, \bar{x}_k\rangle\right]$$

$$= \sum_{z=1}^{c}\sum_{k=1}^{c} u^{(y)}_{[i,z]}u^{(y)}_{[j,k]}\mathbb{E}[\langle \bar{x}_z, \bar{x}_k\rangle]$$

$$\overset{(1)}{=} \sum_{z=1}^{c} u^{(y)}_{[i,z]}u^{(y)}_{[j,z]}\mathbb{E}[\langle \bar{x}_z, \bar{x}_z\rangle]$$

$$= \sum_{z=1}^{c} u^{(y)}_{[i,z]}u^{(y)}_{[j,z]}N_z^2\xi_z = \hat{\Sigma}^{(y)}_{[i]}$$

where (1) is justified by that $\bar{x}_z$ and $\bar{x}_k$ are random variables with orthogonal mean vectors (when $k \neq z$). We are now ready to prove the concentration bounds.

$$\tilde{\nu}^{(y)}_{[i]} - \hat{\nu}^{(y)}_{[i]} = \frac{\sum_{j=1}^{c} u^{(y)}_{[i,j]}\langle \bar{x}_j, \mu_y\rangle + b_{[y]} - b_{[i]}}{\sigma} - \frac{u^{(y)}_{[i,y]}N_y\|\mu_y\|^2 + b_{[y]} - b_{[i]}}{\sigma}$$

$$\overset{(1)}{=} \frac{\sum_{j=1}^{c} u^{(y)}_{[i,j]}\langle N_y\mu_j + \bar{n}_j, \mu_y\rangle + b_{[y]} - b_{[i]}}{\sigma} - \frac{u^{(y)}_{[i,y]}N_y\|\mu_y\|^2 + b_{[y]} - b_{[i]}}{\sigma}$$

$$\overset{(2)}{=} \frac{\sum_{j=1}^{c} u^{(y)}_{[i,j]}\langle \bar{n}_j, \mu_y\rangle}{\sigma}$$

$$\overset{(3)}{=} \sum_{j=1}^{c} u^{(y)}_{[i,j]}\sqrt{N_j}\langle n, \mu_y\rangle$$

(1) holds by definition of $\bar{x}_i$, (2) by $\langle \mu_i, \mu_j, = \rangle \|\mu_{[i]}\| \mathbb{1}_{\{i=j\}}$ and (3) since $\bar{n}_j \sim \mathcal{N}\left(0, N_j \sigma^2 I_d\right)$ with $n \sim \mathcal{N}(0, I_d)$.

Hence we can use Lemma A.4 to bound $\left|\tilde{\nu}_{[i]}^{(y)} - \hat{\nu}_{[i]}^{(y)}\right|$ with probability $1 - \tilde{\delta}$ as follows:

$$\left|\tilde{\nu}_{[i]}^{(y)} - \hat{\nu}_{[i]}^{(y)}\right| \leq \frac{1}{\sqrt[4]{d}} \left|\sum_{j=1}^{c} u_{[i,j]}^{(y)}\right| \sqrt{2 N_j s_y \log\left(\frac{1}{\tilde{\tilde{\delta}}}\right)}. \tag{30}$$

Moving on to bound $\left|\tilde{\Sigma}_{[i]}^{(y)} - \hat{\Sigma}_{[i]}^{(y)}\right|$:

$$\begin{aligned}
\tilde{\Sigma}_{[i,j]}^{(y)} - \hat{\Sigma}_{[i,j]}^{(y)} &= \sum_{z=1}^{c}\sum_{k=1}^{c} u_{[i,z]}^{(y)} u_{[j,k]}^{(y)} \langle \bar{x}_z, \bar{x}_k \rangle - \sum_{z=1}^{c} u_{[i,z]}^{(y)} u_{[j,z]}^{(y)} N_z^2 \xi_z \\
&= \sum_{z=1}^{c}\sum_{k\neq z}^{c} u_{[i,z]}^{(y)} u_{[j,k]}^{(y)} \langle \bar{x}_z, \bar{x}_k \rangle + \sum_{z=1}^{c} u_{[i,z]}^{(y)} u_{[j,z]}^{(y)} \langle \bar{x}_z, \bar{x}_z \rangle - \sum_{z=1}^{c} u_{[i,z]}^{(y)} u_{[j,z]}^{(y)} N_z^2 \xi_z. 
\end{aligned} \tag{31}$$

Looking at left sum of $z \neq k$:

$$\begin{aligned}
\sum_{z=1}^{c}\sum_{k\neq z}^{c} u_{[i,z]}^{(y)} u_{[j,k]}^{(y)} \langle \bar{x}_z, \bar{x}_k \rangle &= \sum_{z=1}^{c}\sum_{k\neq z}^{c} u_{[i,z]}^{(y)} u_{[j,k]}^{(y)} \langle N_z \mu_z + \bar{n}_z, N_k \mu_k + \bar{n}_k \rangle \\
&\overset{(1)}{=} \sum_{z=1}^{c}\sum_{k\neq z}^{c} u_{[i,z]}^{(y)} u_{[j,k]}^{(y)} \left( \langle N_z \mu_z, \bar{n}_k \rangle + \langle N_k \mu_k, \bar{n}_z \rangle + \langle \bar{n}_z, \bar{n}_k \rangle \right).
\end{aligned} \tag{32}$$

where (1) holds since $\langle \mu_i, \mu_j \rangle = \|\mu_i\|^2 \mathbb{1}_{\{i=j\}}$.

Using *Lemma A.4* we can bound each term in eq. (32) such that w.p $1 - 3\tilde{\delta}$ the following holds:

$$\begin{aligned}
\langle N_z \mu_z, \bar{n}_k \rangle + \langle N_k \mu_k, \bar{n}_z \rangle + \langle \bar{n}_z, \bar{n}_k \rangle &\leq \frac{\sqrt{N_z N_k}}{\sqrt[3/4]{d}}\left(\sqrt{N_z s_z} + \sqrt{N_k s_k}\right)\sqrt{2\log\left(\frac{1}{\tilde{\tilde{\delta}}}\right)} + \frac{\sqrt{N_z N_k}}{\sqrt{d}}\log\left(\frac{4}{\tilde{\tilde{\delta}}}\right) \\
&\leq 2\frac{N_{\max}^{3/2}\sqrt{s_{\max}}}{\sqrt[3/4]{d}}\sqrt{2\log\left(\frac{1}{\tilde{\tilde{\delta}}}\right)} + \frac{N_{\max}}{\sqrt{d}}\log\left(\frac{4}{\tilde{\tilde{\delta}}}\right).
\end{aligned}$$

Moving on to bound the residual terms in eq. (31):

$$\sum_{z=1}^{c} u_{[i,z]}^{(y)} u_{[j,z]}^{(y)} \left( \langle \bar{x}_z, \bar{x}_z \rangle - N_z^2 \xi_z \right) = \sum_{z=1}^{c} u_{[i,z]}^{(y)} u_{[j,z]}^{(y)} \left( 2 N_z \langle \mu_z, \bar{n}_z \rangle + \|\bar{n}_z\|^2 - N_z \sigma^2 d \right). \tag{33}$$

Again using Lemma A.4 we can bound each term in eq. (33) such that w.p $1 - 2\tilde{\delta}$ the following holds:

$$\begin{aligned}
2 N_z \langle \mu_z, \bar{n}_z \rangle + \|\bar{n}_z\|^2 - N_z \sigma^2 d &\leq 2\frac{N_z^{3/2}\sqrt{s_z}}{\sqrt[3/4]{d}}\sqrt{2\log\left(\frac{1}{\tilde{\tilde{\delta}}}\right)} + 4\frac{N_z}{\sqrt{d}}\log\left(\frac{2}{\tilde{\tilde{\delta}}}\right) \\
&\leq 2\frac{N_{\max}^{3/2}\sqrt{s_{\max}}}{\sqrt[3/4]{d}}\sqrt{2\log\left(\frac{1}{\tilde{\tilde{\delta}}}\right)} + 4\frac{N_{\max}}{\sqrt{d}}\log\left(\frac{2}{\tilde{\tilde{\delta}}}\right)
\end{aligned}$$

Hence we get that w.p $1 - \left(2c + 3\frac{c^2-c}{2}\right)\tilde{\delta}$ for any $i, j \in [c]$ it holds that:

$$\begin{aligned}
\left|\tilde{\Sigma}_{[i,j]}^{(y)} - \hat{\Sigma}_{[i,j]}^{(y)}\right| &\leq \sum_{z=1}^{c}\sum_{k=1}^{c} \left|u_{[i,z]}^{(y)} u_{[j,k]}^{(y)}\right| \left( 2\frac{N_{\max}^{3/2}\sqrt{s_{\max}}}{\sqrt[3/4]{d}}\sqrt{2\log\left(\frac{1}{\tilde{\tilde{\delta}}}\right)} + 4\frac{N_{\max}}{\sqrt{d}}\log\left(\frac{2}{\tilde{\tilde{\delta}}}\right) \right) \\
&\leq 4\frac{N_{\max}^{3/2}\left(\sqrt{s_{\max}} + 1\right)}{\sqrt{d}}\sum_{z=1}^{c}\sum_{k=1}^{c}\left|u_{[i,z]}^{(y)} u_{[j,k]}^{(y)}\right|\log\left(\frac{2}{\tilde{\tilde{\delta}}}\right)
\end{aligned}$$

and

$$\left| \tilde{\nu}_{[i]}^{(y)} - \hat{\nu}_{[i]}^{(y)} \right| \leq \frac{1}{\sqrt[4]{d}} \left| \sum_{j=1}^{c} u_{[i,j]}^{(y)} \right| \sqrt{2 N_j s_y \log\left(\frac{1}{\tilde{\delta}}\right)}$$

where we used $3\frac{c^2-c}{2}$ concentration bounds for the case of $z \neq k$ and additional $2c$ concatenation bounds for the case of $k = z$ (notice that these bounds also hold for the concentration of $\hat{\nu}^{(y)}$). Finally we set $\tilde{\delta} = \frac{\delta}{1.5c(c+1)}$ since $2c + 3\frac{c^2-c}{2} = 1.5c(c+1)$ and get the desired result. $\qquad \square$

### A.4    The error of the approximate model

We now show the correctness of the expression for $\mathrm{Err}_{y\to k}(W,b)$ were $W$, $b$ is a predictor that is learned under our data assumption (Section 2.3). Then we continue by showing that given a predictor of the form of eq. (11) the error of the predictor at the limit of $d \to \infty$ can be expressed as a multivariate Gaussian distribution in $\mathbb{R}^c$ that is defined by the predictor's coefficients $\{\tilde{\alpha}_i\}_{i=1}^c$ and the problem parameters.

*Fact* 1. Let $x|y \sim \mathcal{N}(\mu_y; \sigma^2 I_d)$. Then, for any predictor $(W,b)$ that is learned under our data assumption (Section 2.3) and any $k \neq y$, we have

$$\mathrm{Err}_y(W,b) = 1 - \mathbb{P}\Big(\mathcal{N}(\nu^{(y)}; \Sigma^{(y)}) \geq 0\Big) \quad \text{and} \quad \mathrm{Err}_{y\to k}(W,b) = Q\left(\frac{\nu_{[k]}^{(y)}}{\sqrt{\Sigma_{[k,k]}^{(y)}}}\right),$$

where $\mathrm{Err}_y$ and $\mathrm{Err}_{y\to k}$ are defined in eqs. (1) and (3), respectively, $Q(t) = \mathbb{P}(\mathcal{N}(0,1) > t)$ is the Gaussian error function, and, for every $i, j \in [c]$

$$\nu_{[j]}^{(y)} = \frac{(w_y - w_j)^\top \mu_i + b_{[y]} - b_{[j]}}{\sigma} \quad \text{and} \quad \Sigma_{[i,j]}^{(y)} = (w_y - w_i)^\top (w_y - w_j).$$

where $\{\nu^{(y)}, \Sigma^{(y)}\}$ are defined as the **score statistics** of the predictor $W$, $b$.

*Proof.*

$$\begin{aligned}
\mathbb{P}\left[\langle w_k, x\rangle + b_k > \langle w_y, x\rangle + b_y\right] &= \mathbb{P}\left[\langle w_k, \mu_y + n\rangle + b_k > \langle w_y, \mu_y + n\rangle + b_y\right] \\
&= \mathbb{P}\left[\langle w_k - w_y, n\rangle > \langle w_y - w_k, \mu_y\rangle + b_y - b_k\right] \\
&= \mathbb{P}\left[\mathcal{N}\left(0, \|w_y - w_k\|^2 \sigma^2 I_d\right) > \langle w_y - w_k, \mu_y\rangle + b_y - b_k\right] \\
&= \mathbb{P}\left[\mathcal{N}\left(0, I_d\right) > \frac{\langle w_y - w_k, \mu_y\rangle + b_y - b_k}{\sigma \|w_y - w_k\|}\right] \\
&= Q\left(\frac{\langle w_y - w_k, \mu_y\rangle + b_y - b_k}{\sigma \|w_y - w_k\|}\right)
\end{aligned} \tag{34}$$

$\square$

**Theorem A.6.** *Let $\mathcal{D} \sim \mathcal{P}_d\left(\{s_i, N_i\}_{i=1}^c\right)$ be a dataset from $\mathcal{P}_d\left(\{s_i, N_i\}_{i=1}^c\right)$ as defined in Section 2.3 and $\tilde{W}$, $\tilde{b}$ be a predictor over $\mathcal{D}$ of the form of eq. (11). Then,*

$$\lim_{d\to\infty} \mathrm{Err}_y\left(\tilde{W}, \tilde{b}\right) = 1 - \mathbb{P}\left[\mathcal{N}\left(\hat{\nu}^{(y)}, \hat{\Sigma}^{(y)}\right) \geq 0\right],$$

*where,*

$$\hat{\nu}_{[i]}^{(y)} = \frac{\left[\tilde{\alpha}_{y[y]} - \tilde{\alpha}_{i[y]}\right] N_y \|\mu_y\|^2 + \tilde{b}_{[y]} - \tilde{b}_{[i]}}{\sigma} \quad \text{and} \quad \hat{\Sigma}_{[i,j]}^{(y)} = \sum_{z=1}^c \left[\tilde{\alpha}_{y[z]} - \tilde{\alpha}_{i[z]}\right]\left[\tilde{\alpha}_{y[z]} - \tilde{\alpha}_{j[z]}\right] N_z^2 \xi_z.$$

*In addition, the worst class error of $\tilde{W}$, $\tilde{b}$ at the limit of $d \to \infty$ is lower bounded by:*

$$\max_{k\neq y} \lim_{d\to\infty} Q\left(\frac{\left[\tilde{\alpha}_{y[y]} - \tilde{\alpha}_{k[y]}\right] N_y \|\mu_y\|^2 + \tilde{b}_{[y]} - \tilde{b}_{[k]}}{\sigma \sqrt{\sum_{i=1}^c \left(\tilde{\alpha}_{y[i]} - \tilde{\alpha}_{k[i]}\right)^2 N_i^2 \xi_i}}\right) = \max_{y, n\neq y} \lim_{d\to\infty} \mathrm{Err}_{y\to k}\left(\tilde{W}, \tilde{b}\right) \leq \lim_{d\to\infty} \mathrm{Err}_{\mathrm{wc}}\left(\tilde{W}, \tilde{b}\right). \tag{35}$$

*Proof.* Recall that the test error of a given predictor $\tilde{W}, \tilde{b}$ of the form of eq. (11) is defined by the probability of misclassifying a new test sample $(x, y)$.

$$\underbrace{\mathbb{E}_{x|y}\left[\ell_{0-1}\left(\tilde{W}, \tilde{b}; (x, y)\right)\right]}_{\text{error}} = \mathbb{P}\left[\exists k \neq y \; : \; \langle \tilde{w}_k, x \rangle + \tilde{b}_{[k]} > \langle \tilde{w}_y, x \rangle + \tilde{b}_{[y]}\right]. \tag{36}$$

eq. (36) can be rewritten by its complementary event,

$$\underbrace{\mathbb{E}_{x|y}\left[\ell_{0-1}\left(\tilde{W}, \tilde{b}; (x, y)\right)\right]}_{\text{error}} = 1 - \underbrace{\mathbb{P}\left[\forall k \in [c]: \; \langle \tilde{w}_y, x \rangle + \tilde{b}_{[y]} \geq \langle \tilde{w}_k, x \rangle + \tilde{b}_{[k]}\right]}_{\text{accuracy}}$$

$$= 1 - \mathbb{P}\left[\forall k \in [c]: \; \langle \tilde{w}_y - \tilde{w}_k, \mu_y \rangle + \langle \tilde{w}_y - \tilde{w}_k, n \rangle + \tilde{b}_{[y]} - \tilde{b}_{[k]} \geq 0\right]$$

$$= 1 - \mathbb{P}\left[\mathcal{N}\left(\tilde{\nu}^{(y)}, \tilde{\Sigma}^{(y)}\right) \geq 0\right],$$

where $\tilde{\nu}^{(y)}$ and $\tilde{\Sigma}^{(y)}$ are defined as follows:

$$\tilde{\nu}_{[k]}^{(y)} = \frac{\langle \tilde{w}_y - \tilde{w}_k, \mu_y \rangle + \tilde{b}_{[y]} - \tilde{b}_{[k]}}{\sigma} \quad and \quad \tilde{\Sigma}_{[i,j]}^{(y)} = \langle \tilde{w}_y - \tilde{w}_i, \tilde{y} - \tilde{w}_j \rangle.$$

Hence,

$$\lim_{d \to \infty} \mathbb{E}_{x|y}\left[\ell_{0-1}\left(W; (x, y)\right)\right] = 1 - \lim_{d \to \infty} \mathbb{P}\left[\mathcal{N}\left(\tilde{\nu}^{(y)}, \tilde{\Sigma}^{(y)}\right) \geq 0\right].$$

From Lemma A.5 it holds that at the limit of $d \to \infty$, $\nu^{(y)} \overset{p}{\to} \mathbb{E}\left[\nu^{(y)}\right]$ and $\Sigma^{(y)} \overset{p}{\to} \mathbb{E}\left[\Sigma^{(y)}\right]$. Additionally, since the multivariate-Gaussian distribution is continuous in its parameters we get that:

$$\lim_{d \to \infty} \mathbb{P}\left[\mathcal{N}\left(\hat{\nu}^{(y)}, \hat{\Sigma}^{(y)}\right) \geq 0\right] = \mathbb{P}\left[\mathcal{N}\left(\mathbb{E}\left[\tilde{\nu}^{(y)}\right], \mathbb{E}\left[\tilde{\Sigma}^{(y)}\right]\right) \geq 0\right].$$

Which leads us to the desired results,

$$\mathbb{E}\left[\tilde{\nu}_{[k]}^{(y)}\right] = \frac{\left[\tilde{\alpha}_{y[y]} - \tilde{\alpha}_{k[y]}\right]N_y\|\mu_y\|^2 + \tilde{b}_{[y]} - \tilde{b}_{[k]}}{\sigma} \quad and \quad \mathbb{E}\left[\tilde{\Sigma}_{[i,j]}^{(y)}\right] = \sum_{z=1}^{c}\left[\tilde{\alpha}_{y[z]} - \tilde{\alpha}_{i[z]}\right]\left[\tilde{\alpha}_{y[z]} - \tilde{\alpha}_{j[z]}\right]N_z^2\xi_z.$$

See explicit calculation for these expressions in Appendix A.3. Additionally, this means that the test error (at the limit of $d \to \infty$) of a given class $y$ can be lower bounded by the score $\lim_{d \to \infty} \text{Err}_{y \to k}$ of a single $k \neq y$ as follows:

$$\lim_{d \to \infty} \mathbb{E}_{x|y}\left[\ell_{0-1}\left(W; (x, y)\right)\right] = 1 - \lim_{d \to \infty} \mathbb{P}\left[\mathcal{N}\left(\tilde{\nu}^{(y)}, \tilde{\Sigma}^{(y)}\right) \geq 0\right]$$

$$= \lim_{d \to \infty} \mathbb{P}\left[\exists i \neq y : \mathcal{N}\left(\tilde{\nu}_{[i]}^{(y)}, \tilde{\Sigma}_{[i,i]}^{(y)}\right) \geq 2\tilde{\nu}_{[i]}^{(y)}\right]$$

$$\geq \max_{i \neq y} \lim_{d \to \infty} \mathbb{P}\left[\mathcal{N}\left(\tilde{\nu}_{[i]}^{(y)}, \tilde{\Sigma}_{[i,i]}^{(y)}\right) \geq 2\tilde{\nu}_{[i]}^{(y)}\right]$$

$$= \max_{i \neq y} \lim_{d \to \infty} \mathbb{P}\left[\mathcal{N}(0, 1) \geq \frac{\tilde{\nu}_{[i]}^{(y)}}{\sqrt{\tilde{\Sigma}_{[i,i]}^{(y)}}}\right].$$

Therefore we conclude that the worst class error at the limit of $d \to \infty$ is lower bounded as follows:

$$\max_{k \neq y} \lim_{d \to \infty} Q\left(\frac{\|\mu_y\|^2 N_y[\tilde{\alpha}_y - \tilde{\alpha}_k]_{[y]} + \tilde{b}_{[y]} - \tilde{b}_{[k]}}{\sigma\sqrt{\sum_{z=1}^{c}\left[\tilde{\alpha}_{y[z]} - \tilde{\alpha}_{k[z]}\right]^2 N_z^2\xi_z}}\right) \leq \lim_{d \to \infty} \text{Err}_{\text{wc}}\left(\tilde{W}, \tilde{b}\right)$$

$$\square$$

# B Implicit bias of gradient descent

In this section, we provide a straightforward extensions of prior work on the implicit bias of gradient descent (GD) to account training the bias parameter (with a separate learning rate), and the MA and CDT losses. We empirically validate these implicit biases in Appendix G.8.

## B.1 Implicit bias of cross entropy

Recall the standard definition for the maximum margin predictor.

$$W^{\mathrm{mm}} := \underset{W \in \mathbb{R}^{d \times c}}{\arg\min} \|W\|_{\mathrm{F}}^2$$
$$\text{subject to} \quad (w_{y_i} - w_k)^\top x_i \geq 1 \quad \text{for all} \ \ i \leq N \ \text{and} \ k \neq y_i. \tag{37}$$

In addition recall the definition of the log-loss objective for multiclass classification.

$$\mathcal{L}^{\mathrm{ce}}(W;b) = \frac{1}{N} \sum_{i=1}^N \ell^{\mathrm{ce}}\big(W^\top x_i + b_{y_i}; y_i\big) \ \ \text{where} \ \ \ell^{\mathrm{ce}}(z;y) := \log\left(\sum_{i \in [c]} e^{z_{[i]} - z_{[y]}}\right). \tag{38}$$

Soudry et al. (2018) show that minimizing the log-loss eq. (38) on a separable dataset using unregularized gradient descent with a small enough learning rate $\eta$ converges in direction to the maximum margin classifier eq. (37). Where gradient descent minimizes $\mathcal{L}(W)$ with iterates of the form:

$$W(t+1) = W(t) - \eta \nabla_W \mathcal{L}(W(t))$$

**Proposition B.1** (Soudry et al. (2018), Theorem 7)**.** *For almost all multiclass datasets which are linearly separable (i.e. the constraints in eq. (37) are feasible), any starting point $W_0$ and a small enough stepsize $\eta \leq \frac{1}{\max_{i \in [N]} \|x_i\|^2}$ we have*

$$\lim_{t \to \infty} \frac{W(t)}{\|W(t)\|_{\mathrm{F}}} = \frac{W^{\mathrm{mm}}}{\|W^{\mathrm{mm}}\|_{\mathrm{F}}},$$

*where $\lim_{t \to \infty} \|W(t)\|_F = \infty$.*

*Remark* B.2. To extend the result of Proposition B.1 to non-homogenous linear predictors, all input vectors $x_i$ need to be extended with an additional 1 component.

We extend the implicit bias of gradient descent for the case where the weights and biases are learned with different learning rates $\eta$ and $\eta'$. Specifically, we show that for any $\rho > 0$ such that $\rho = \left(\frac{\eta}{\eta'}\right)^2$ gradient decent converges in the direction of:

$$W^{\mathrm{mm}}, b^{\mathrm{mm}} := \underset{W \in \mathbb{R}^{d \times c}, b \in \mathbb{R}^c}{\arg\min} \|W\|_{\mathrm{F}}^2 + \rho \|b\|^2$$
$$\text{subject to} \quad (w_{y_i} - w_k)^\top x_i + b_{[y]} - b_{[k]} \geq 1 \quad \text{for all} \ \ i \leq N \ \text{and} \ k \neq y_i,$$

where gradient descent iterates are defined by:

$$W(t+1) = W(t) - \eta \nabla_W \mathcal{L}(W(t), b(t)) \ \ \text{and} \ \ b(t+1) = b(t) - \eta' \nabla_b \mathcal{L}(W(t), b(t)), \tag{39}$$

and $\mathcal{L}$ is defined as in eq. (38). In addition we denote the concatenation of the weights and biases i.e., adding the bias vector as the last row of $W$ by $[W|b]$.

**Lemma B.3.** *Let $\rho > 0$ and $\mathcal{D} = \{(x_i, y_i)\}_{i=1}^N$ be a linearly separable dataset ,$W(t)$ and $b(t)$ be the iterates of gradient descent 39 with $\mathcal{L}$ as defined in 38 with learning rate $\eta \leq \frac{1}{\max_i \|x_i\|^2 + \frac{1}{\rho}}$ and $\eta' = \frac{\eta}{\sqrt{\rho}}$ for the weights and biases respectively. Then,*

$$\lim_{t \to \infty} \frac{[W(t)|b(t)]}{\|[W(t)|b(t)]\|_F} = \frac{[W^{\mathrm{mm}}|b^{\mathrm{mm}}]}{\|[W^{\mathrm{mm}}|b^{\mathrm{mm}}]\|_F}$$

*Proof.* The proof of Lemma B.3 is a direct result of Proposition B.1. Specifically, we extend each point in the training set by $\frac{1}{\sqrt{\rho}}$ and run gradient descent with $\eta < \left(\max_i \|x_i\|^2 + \frac{1}{\rho}\right)^{-1}$. From Lemma B.9 it holds that $\mathcal{L}$ is $\max_{i\in[N]}\|x_i\|^2 + \frac{1}{\rho}$-smooth. Therefore, according to the implicit bias the iterates of GD converge in the direction of

$$W^{\mathrm{mm}}, b^{\mathrm{mm}} \coloneqq \underset{W\in\mathbb{R}^{d\times c}, b\in\mathbb{R}^c}{\arg\min} \quad \|W\|_{\mathrm{F}}^2 + \|b\|^2$$
$$\text{subject to} \quad (w_{y_i} - w_k)^\top x_i + \frac{1}{\sqrt{\rho}}b_{[y_i]} - \frac{1}{\sqrt{\rho}}b_{[k]} \geq 1 \quad \text{for all} \ \ i \leq N \text{ and } k \neq y_i.$$

Rearranging the bias terms as $\widetilde{b}_i = \frac{1}{\sqrt{\rho}}b_i$, we get

$$W^{\mathrm{mm}}, b^{\mathrm{mm}} \coloneqq \underset{W\in\mathbb{R}^{d\times c}, b\in\mathbb{R}^c}{\arg\min} \quad \|W\|_{\mathrm{F}}^2 + \rho\|\widetilde{b}\|^2$$
$$\text{subject to} \quad (w_{y_i} - w_k)^\top x_i + \widetilde{b}_{y_i} - \widetilde{b}_k \geq 1 \quad \text{for all} \ \ i \leq N \text{ and } k \neq y_i,$$

which results in the desired margin problem. Furthermore, notice that setting the bias feature to $\frac{1}{\sqrt{\rho}}$ (instead of 1) is equivalent to adjusting the learning rate of the biases to $\frac{\eta}{\sqrt{\rho}} = \eta'$. This equivalence arises due to the effect of applying a factor of $\frac{1}{\sqrt{\rho}}$ to the bias terms, resulting in the same factor in the gradient. Thus $\frac{\eta}{\sqrt{\rho}} = \eta' \implies \rho = \left(\frac{\eta}{\eta'}\right)^2$. $\qquad\square$

## B.2 Implicit bias of margin adjustment

Recall the definition of the MA predictor:

$$W^{\mathrm{ma}}, b^{\mathrm{ma}} \coloneqq \underset{W\in\mathbb{R}^{d\times c}, b\in\mathbb{R}^c}{\arg\min} \quad \|W\|_{\mathrm{F}}^2 + \rho\|b\|^2$$
$$\text{subject to} \quad (w_{y_i} - w_k)^\top x_i + b_{[y_i]} - b_{[k]} \geq \delta_{y_i} \quad \text{for all} \ \ i \leq N \text{ and } k \neq y_i. \tag{40}$$

In addition the margin adjustment loss is

$$\ell^{\mathrm{ma}}(z; y) \coloneqq \log\left(\sum_{i\in[c]} e^{\frac{z_{[i]}}{\delta_y} - \frac{z_{[y]}}{\delta_y}}\right). \tag{41}$$

To obtain the convergence of GD with the MA loss in the direction of the MA predictor eq. (40) we reduce the problem to standard cross-entropy training / margin maximization with rescaled features. Specifically notice that for any dataset $\{(x_i, y_i)\}_{i=1}^N$ it holds that

$$\sum_{i=1}^N \ell^{\mathrm{ma}}(W^\top x_i; y) = \sum_{i=1}^N \log\left(\sum_{j\in[c]} e^{\frac{w_j^\top x_i}{\delta_y} - \frac{w_y^\top x_i}{\delta_y}}\right) = \sum_{i=1}^N \ell^{\mathrm{ce}}(W^\top x_i'; y) \tag{42}$$

where $x_i' = \frac{x_i}{\delta_{y_i}}$.

**Lemma B.4.** *Let $\rho > 0$ and $\mathcal{D} = \{(x_i, y_i)\}_{i=1}^N$ be a linearly separable dataset, $W(t)$ and $b(t)$ be the iterates of gradient descent eq. (39) on $\mathcal{L}(W, b) = \frac{1}{N}\sum_{i=1}^N \ell^{\mathrm{ma}}(z_i; y_i)$ eq. (41), with $\{\delta_i\}_{i=1}^c > 0$, learning rates $\eta \leq \min_{i\in[N]} \frac{\delta_{y_i}^2}{\|x_i\|^2 + \frac{1}{\rho}}$ and $\eta' \leq \frac{\eta}{\sqrt{\rho}}$ for the weights and biases respectively. Then,*

$$\lim_{t\to\infty} \frac{[W(t)|b(t)]}{\|[W(t)|b(t)]\|_F} = \frac{[W^{\mathrm{ma}}|b^{\mathrm{ma}}]}{\|[W^{\mathrm{ma}}|b^{\mathrm{ma}}]\|_F}$$

*Proof.* We prove this by demonstrating that the iterates of gradient descent with the log-loss on the scaled features of $\mathcal{D}$ converge in the direction of the MA predictor. From eq. (42), we conclude the desired result.

First, we extend the data points by a factor of $\frac{1}{\sqrt{\rho}}$ and scale each point by a factor of $\frac{1}{\delta_{y_i}}$, i.e.,

$$x_i' = \frac{\left[ x_i | \frac{1}{\sqrt{\rho}} \right]}{\delta_{y_i}}$$

It should be noted that this augmentation alters the upper bound of smoothness for the log-loss to $\max_{i\in[N]} \frac{\|x_i\|^2 + \frac{1}{\rho}}{\delta_{y_i}^2}$. Consequently, based on eq. (42), Proposition B.10, and Lemma B.3 it is guaranteed that running GD with $\eta \leq \min_{i\in[N]} \frac{\delta_{y_i}^2}{\|x_i\|^2 + \frac{1}{\rho}}$ will converge in the direction of the max margin.

$$W^{\mathrm{mm}}, b^{\mathrm{mm}} := \underset{W\in\mathbb{R}^{d\times c}, b\in\mathbb{R}^c}{\arg\min} \quad \|W\|_{\mathrm{F}}^2 + \rho\|b\|^2$$

$$\text{subject to} \quad (w_{y_i} - w_k)^\top \frac{x_i}{\delta_{y_i}} + \frac{1}{\delta_{y_i}} b_{[y_i]} - \frac{1}{\delta_{y_i}} b_{[k]} \geq 1 \quad \text{for all} \quad i \leq N \text{ and } k \neq y_i. \tag{43}$$

Notice that eq. (43) is equivalent to the MA predictor margin problem, eq. (40) as for any $i \in [N]$ and $k \neq y$:

$$\frac{1}{\delta_{y_i}}(w_{y_i} - w_k)^\top x_i + \frac{1}{\delta_{y_i}} b_{[y_i]} - \frac{1}{\delta_{y_i}} b_{[k]} \geq 1 \iff (w_{y_i} - w_k)^\top x_i + b_{[y_i]} - b_{[k]} \geq \delta_{y_i}$$

Finally, the convergence of GD using different learning rates for the weights and biases is derived from Lemma B.3. In this case, scaling $\left[ x_i | \frac{1}{\sqrt{\rho}} \right]$ by $1/\delta_{y_i}$ does not affect the ratio of the weights and biases learning rates, yielding $\rho = \left( \frac{\eta}{\eta'} \right)^2$. $\qquad\qquad\square$

### B.3 Implicit bias of class dependent temperature

To show the implicit bias of gradient descent when using the CDT loss (eq. (45) below), we establish a connection between two findings from previous studies by Ji et al. (2020) and Kini et al. (2021). Ji et al. (2020) show that when running unregularized gradient descent on a convex, strictly decreasing loss function $f$ that is bounded below by zero and has an unattainable infimum, it converges in direction to the limit of the minimizer of $f$ in a ball of radius $R$, as $R \to \infty$. Additionally, Kini et al. (2021) show that the regularized CDT loss minimizer converges in the direction of the CDT predictor (Equation (6)). We proceed by presenting the relevant theorems from Ji et al. (2020) and Kini et al. (2021), followed by an explanation why the CDT loss satisfies the necessary conditions for the implicit bias of unregularized gradient descent.

$$W^{\mathrm{cdt}} := \underset{W\in\mathbb{R}^{d\times c}}{\arg\min} \|W\|_{\mathrm{F}}^2$$

$$\text{subject to} \quad \left( \frac{w_{y_i}}{\delta_{y_i}} - \frac{w_k}{\delta_k} \right)^\top x_i \geq 1 \quad \text{for all} \quad i \leq N \text{ and } k \neq y_i. \tag{44}$$

The CDT loss is defined by:

$$\ell^{\mathrm{cdt}}(z; y) := \log\left( \sum_{i\in[c]} e^{\frac{z_{[i]}}{\delta_i} - \frac{z_{[y]}}{\delta_y}} \right). \tag{45}$$

In addition,

$$\ell^{\mathrm{cdt}}(W^\top x; y) = \log\left( \sum_{i\in[c]} e^{\frac{w_i^\top x}{\delta_i} - \frac{w_y^\top x}{\delta_y}} \right) = \ell^{\mathrm{ce}}\left( \mathrm{diag}\left( \frac{1}{\delta_1}, \ldots, \delta_c \right) W^\top x; y \right). \tag{46}$$

**Proposition B.5** (Ji et al. (2020), Theorem 4). *Suppose $f$ is convex, $\beta$-smooth, bounded below by 0, and has an unattained infimum, step size $\eta$ satisfies $\eta \leq \min\left\{ \frac{1}{2f(w_0)}, \frac{1}{\beta} \right\}$. Consider the gradient descent iterates*

$(W(t))_{t\geq 0}$ *given by* $W(t+1) := W(t) - \eta\nabla f(W(t))$, *and the regularized solution given by*

$$\bar{w}(R) := \underset{\|w\|_F \leq R}{\arg\min} f(w).$$

*If* $\lim_{t\to\infty} \frac{W(t)}{\|W(t)\|} = \bar{u}$ *for some unit vector* $\bar{u}$*, then also* $\lim_{R\to\infty} \frac{\bar{w}(R)}{\|\bar{w}(R)\|} = \bar{u}$*.*

**Proposition B.6** (Kini et al. (2021), Theorem 4)**.** *Consider a c-classes classification problem as in eq.* (44) *and define the norm-constrained optimal classifier*

$$W(R) := \underset{\|W\|_F \leq R}{\arg\min} \sum_{i=1}^{N} \ell^{\mathrm{cdt}}(W^\top x_i; y_i) \tag{47}$$

*for positive parameters* $\{\delta_i\}_{i=1}^{c} > 0$*. Assume that the training dataset is linearly separable and that* $W^{\mathrm{cdt}}$ *is the solution of Equation* (44)*. Then it holds that,*

$$\lim_{R\to\infty} \frac{W(R)}{\|W(R)\|_F} = \frac{W^{\mathrm{cdt}}}{\|W^{\mathrm{cdt}}\|_F}$$

**Corollary B.7.** *Running gradient descent with the CDT loss, using step size* $\eta < \frac{\min_{j\in[c]} \delta_j^2}{\max_i \|x_i\|^2}$ *and* $\{\delta_i\}_{i=1}^{c} > 0$ *converges in the direction of the CDT predictor, eq.* (44)*, if the limit of the gradient descending iterates* $\lim_{t\to\infty} \frac{W(t)}{\|W(t)\|}$ *exists.*

*Proof.* The CDT loss, is convex, differentiable, $\frac{\max_i \|x_i\|^2}{\min_{j\in[c]} \delta_j^2}$-smooth (eq. (46) and Proposition B.11), bounded below by 0, as scaling the weights by $\delta$ does not affect these properties of the log-loss. Consequently, based on Proposition B.5, it holds that the unregularized gradient descent iterates converge in the direction of the regularized solution of the CDT loss. Furthermore, by applying Proposition B.6, we conclude that this solution converges to the direction of the CDT predictor. $\square$

To generalize the implicit bias of gradient descent with the CDT loss to eq. (6) (with $\rho$ and bias), we can use the argument previously described for the maximum margin classifier.

## B.4 Implicit bias of logit adjustment

We explain why applying gradient descent on the logit adjustment loss eventually results in the maximum margin predictor with a bias adjustment. The logit adjustment (LA) loss is defined by

$$\ell^{\mathrm{la}}(z; y) := \log\left(\sum_{i\in[c]} \exp\left(z_{[i]} + \iota_{[i]} - z_{[y]} - \iota_{[y]}\right)\right). \tag{48}$$

**Lemma B.8.** *Let* $\mathcal{D}$ *be a linearly separable dataset and denote by* $W^{\mathrm{mm}}$ *and* $b^{\mathrm{mm}}$ *the maximum margin predictor on* $\mathcal{D}$*. In addition let* $W^{\mathrm{la}}(t)$ *and* $b^{\mathrm{la}}(t)$ *be the weights and biases of the iterates of gradient descent with logit adjustment loss eq.* (39) *with* $\mathcal{L}(W, b) = \sum_{i=1}^{N} \ell^{\mathrm{la}}(z_i, y_i)$ *on* $\mathcal{D}$ *with hyperparameter* $\iota \in \mathbb{R}^c$ *and* $\eta < \frac{1}{\max_{i\in[N]} \|x_i\|^2}$*. Then,*

$$\lim_{t\to\infty} \frac{\left[W^{\mathrm{la}}(t)|b^{\mathrm{la}}(t)\right]}{\|[W^{\mathrm{la}}|b^{\mathrm{la}}](t)\|_F} = \frac{\left[W^{\mathrm{mm}}|b^{\mathrm{mm}}\right]}{\|[W^{\mathrm{mm}}|b^{\mathrm{mm}}]\|_F}$$

*Proof.* The key observation is that running GD on the LA loss is equivalent to running it on CE with offset biases, i.e.,

$$\ell^{\mathrm{la}}(W^\top x + b; y) = \log\left(\sum_{i\in[c]} \exp\left(w_i^\top x + b_{[i]} + \iota_{[i]} - w_y^\top x - b_{[y]} - \iota_{[y]}\right)\right) = \ell^{\mathrm{ce}}(W^\top x + b + \iota; y).$$

Hence from Proposition B.1 it holds that,

$$\lim_{t\to\infty} \frac{\left[W^{\mathrm{la}}(t)|b^{\mathrm{la}}(t)+\iota\right]}{\|[W^{\mathrm{la}}(t)|b^{\mathrm{la}}(t)+\iota]\|_F} = \lim_{t\to\infty} \frac{[W^{\mathrm{ce}}(t)|b^{\mathrm{ce}}(t)]}{\|[W^{\mathrm{ce}}(t)|b^{\mathrm{ce}}(t)]\|} = \frac{[W^{\mathrm{mm}}|b^{\mathrm{mm}}]}{\|[W^{\mathrm{mm}}|b^{\mathrm{mm}}]\|_F}. \tag{49}$$

where $W^{\mathrm{ce}}(t)$ and $b^{\mathrm{ce}}(t)$ are the iterates of gradient descent, eq. (39), with the log loss. Additional, notice that

$$\lim_{t\to\infty} \frac{\left[W^{\mathrm{la}}(t)|b^{\mathrm{la}}(t)+\iota\right]}{\|[W^{\mathrm{la}}(t)|b^{\mathrm{la}}(t)+\iota]\|_F} = \lim_{t\to\infty} \left( \frac{\left[W^{\mathrm{la}}(t)|b^{\mathrm{la}}(t)\right]}{\|[W^{\mathrm{la}}(t)|b^{\mathrm{la}}(t)+\iota]\|_F} + \frac{[0|\iota]}{\|[W^{\mathrm{la}}(t)|b^{\mathrm{la}}(t)+\iota]\|_F} \right)$$

$$\overset{(1)}{=} \lim_{t\to\infty} \frac{\left[W^{\mathrm{la}}(t)|b^{\mathrm{la}}(t)\right]}{\|[W^{\mathrm{la}}(t)|b^{\mathrm{la}}(t)+\iota]\|_F}$$

where (1) hold by Proposition B.1 as $\lim_{t\to\infty} = \|[W^{\mathrm{ce}}(t)|b^{\mathrm{ce}}(t)]\|_f = \infty$.

We continue by showing that $\lim_{t\to\infty} \frac{\|[W^{\mathrm{la}}(t)|b^{\mathrm{la}}(t)+\iota]\|_F^2}{\|[W^{\mathrm{la}}(t)|b^{\mathrm{la}}(t)]\|_F^2}$ exists. This allows us to conclude the desired result using limit arithmetics.

$$\lim_{t\to\infty} \frac{\|\left[W^{\mathrm{la}}(t)|b^{\mathrm{la}}(t)+\iota\right]\|_F^2}{\|[W^{\mathrm{la}}(t)|b^{\mathrm{la}}(t)]\|_F^2} = \lim_{t\to\infty} \frac{\sum_{i=1}^{c}\|w_i^{\mathrm{la}}(t)\|^2 + \left(b_i^{\mathrm{la}}(t)+\iota_i\right)^2}{\sum_{i=1}^{c}\|w_i^{\mathrm{la}}(t)\|^2 + \left(b_i^{\mathrm{la}}(t)\right)^2}$$

$$= \lim_{t\to\infty} \frac{\sum_{i=1}^{c}\|w_i^{\mathrm{la}}(t)\|^2 + \left(b_i^{\mathrm{la}}(t)+\iota_i\right)^2}{\sum_{i=1}^{c}\|w_i^{\mathrm{la}}(t)\|^2 + \left(b_i^{\mathrm{la}}(t)\right)^2}$$

$$= 1 + \lim_{t\to\infty} \frac{\sum_{i=1}^{c} 2b_i^{\mathrm{la}}(t)\iota_{[i]} + \iota_{[i]}^2}{\sum_{i=1}^{c}\|w_i^{\mathrm{la}}(t)\|^2 + \left(b_i^{\mathrm{la}}(t)\right)^2}$$

$$\overset{(2)}{=} 1 + \lim_{t\to\infty} \frac{\sum_{i=1}^{c} 2b_i^{\mathrm{la}}(t)\iota_{[i]}}{\sum_{i=1}^{c}\|w_i^{\mathrm{la}}(t)\|^2 + \left(b_i^{\mathrm{la}}(t)\right)^2}$$

$$\overset{(3)}{=} 1.$$

Where (2) holds since $\iota$ is fixed and the numerator goes to infinity. We show that (3) holds by bounding $\frac{\sum_{i=1}^{c} 2b_i^{\mathrm{la}}(t)\iota_{[i]}}{\sum_{i=1}^{c}\|w_i^{\mathrm{la}}(t)\|^2 + \left(b_i^{\mathrm{la}}(t)\right)^2}$:

**Upper bound.**

$$\frac{\sum_{i=1}^{c} 2b_i^{\mathrm{la}}(t)\iota_{[i]}}{\sum_{i=1}^{c}\|w_i^{\mathrm{la}}(t)\|^2 + \left(b_i^{\mathrm{la}}(t)\right)^2} \le \frac{\sum_{i=1}^{c} 2b_i^{\mathrm{la}}(t)\iota_{[i]}}{\sum_{i=1}^{c}\left(b_i^{\mathrm{la}}(t)\right)^2}$$

and at $t\to\infty$

$$\lim_{t\to\infty} \frac{\sum_{i=1}^{c} 2b_i^{\mathrm{la}}(t)\iota_{[i]}}{\sum_{i=1}^{c}\left(b_i^{\mathrm{la}}(t)\right)^2} = 0$$

**Lower bound.**

$$\frac{c \cdot \min_{i\in[c]} b_i^{\mathrm{la}}(t)\iota_{[i]}}{\sum_{i=1}^{c}\|w_i^{\mathrm{la}}(t)\|^2 + \left(b_i^{\mathrm{la}}(t)\right)^2} \le \frac{\sum_{i=1}^{c} 2b_i^{\mathrm{la}}(t)\iota_{[i]}}{\sum_{i=1}^{c}\|w_i^{\mathrm{la}}(t)\|^2 + \left(b_i^{\mathrm{la}}(t)\right)^2}$$

and at $t\to\infty$

$$\lim_{t\to\infty} \frac{c \cdot \min_{i\in[c]} b_i^{\mathrm{la}}(t)\iota_{[i]}}{\sum_{i=1}^{c}\|w_i^{\mathrm{la}}(t)\|^2 + \left(b_i^{\mathrm{la}}(t)\right)^2} = 0$$

Therefore, by limit arithmetics, if the limit of a two series $\lim_{t\to\infty} A(t)$ and $\lim_{t\to\infty} B(t)$ exist then,

$$\lim_{t\to\infty} A(t) \cdot B(t) = \lim_{t\to\infty} (A \cdot B)(t).$$

Thus,

$$\lim_{t\to\infty} \frac{\left[W^{\mathrm{la}}(t)|b^{\mathrm{la}}(t)\right]}{\|[W^{\mathrm{la}}(t)|b^{\mathrm{la}}(t)]\|_F} = \lim_{t\to\infty} \frac{\left[W^{\mathrm{la}}(t)|b^{\mathrm{la}}(t) + \iota\right]}{\|[W^{\mathrm{la}}(t)|b^{\mathrm{la}}(t) + \iota]\|_F} = \lim_{t\to\infty} \frac{\left[W^{\mathrm{ce}}(t)|b^{\mathrm{ce}}(t)\right]}{\|[W^{\mathrm{ce}}(t)|b^{\mathrm{ce}}(t)]\|} = \frac{\left[W^{\mathrm{mm}}|b^{\mathrm{mm}}\right]}{\|[W^{\mathrm{mm}}|b^{\mathrm{mm}}]\|_F}$$

as needed. $\qquad\square$

### B.5 Helper lemmas for smoothness of analyzed losses

In this subsection we present helper lemmas for bounding the smoothness of cross entropy, margin adjustment and cross dependent temperature losses.

**Lemma B.9.** *Let $\mathcal{D} = \{(x_i, y_i)\}_{i=1}^N$ in $\mathbb{R}^d$ and let $\mathcal{D}'$ be a dataset in $\mathbb{R}^{d+1}$ that contains all data points from $\mathcal{D}$ extended by a single coordinate with a fixed value $r \geq 0$. Then, $\mathcal{L}^{\mathrm{ce}}(W; b)$ over $\mathcal{D}'$ is $\max_{i\in[N]} \|x_i\|^2 + r^2$ smooth.*

*Proof.* We show this by bounding the Hessian of $\mathcal{L}^{\mathrm{ce}}(W; b)$

$$\mathcal{L}^{\mathrm{ce}}(W; b) = \frac{1}{N} \sum_{i=1}^N \ell^{\mathrm{ce}}(W^\top x_i + r b_{y_i}; y_i) = \frac{1}{N} \sum_{i=1}^N \log\left(\sum_{i\in[c]} e^{z_{[i]} - z_{[y]}}\right) = \frac{1}{N} \sum_{i=1}^N \left(\log\left(\sum_{i\in[c]} e^{z_{[i]}}\right) - z_{[y]}\right)$$

Hence,

$$\nabla^2 \mathcal{L}^{\mathrm{ce}}(W; b) = \nabla^2 \frac{1}{N} \sum_{i=1}^N \log\left(\sum_{i\in[c]} e^{z_{[i]}}\right)$$

as $z_{[y]}$ is linear in $W$ and $b$.

$$\frac{\partial \mathcal{L}^{\mathrm{ce}}}{\partial W_{k[j]}} = \frac{1}{N} \sum_{i=1}^N \frac{x_{i[j]} e^{z_{[k]}}}{\sum_{i\in[c]} e^{z_{[i]}}} \quad\text{and}\quad \frac{\partial \mathcal{L}^{\mathrm{ce}}}{\partial b_k} = \frac{1}{N} \sum_{i=1}^N \frac{r e^{z_{[k]}}}{\sum_{i\in[c]} e^{z_{[i]}}}$$

In addition,

$$\frac{\partial^2 \mathcal{L}^{\mathrm{ce}}}{\partial W_{l[m]} \partial W_{k[j]}} = \frac{1}{N} \sum_{i=1}^N \frac{x_{i[m]} x_{i[j]} e^{z_{[k]}} \sum_{i\in[c]} e^{z_{[i]}} \mathbb{1}_{\{l=k\}} - x_{i[m]} x_{i[j]} e^{z_{[k]}} e^{z_{[l]}}}{\left(\sum_{i\in[c]} e^{z_{[i]}}\right)^2}$$

$$\frac{\partial^2 \mathcal{L}^{\mathrm{ce}}}{\partial b_{[l]} \partial b_{[k]}} = \frac{1}{N} \sum_{i=1}^N \frac{r^2 e^{z_{[k]}} \sum_{i\in[c]} e^{z_{[i]}} \mathbb{1}_{\{l=k\}} - r^2 e^{z_{[k]}} e^{z_{[l]}}}{\left(\sum_{i\in[c]} e^{z_{[i]}}\right)^2}$$

$$\frac{\partial^2 \mathcal{L}^{\mathrm{ce}}}{\partial b_{[l]} \partial W_{k[j]}} = \frac{1}{N} \sum_{i=1}^N \frac{r x_{i[j]} e^{z_{[k]}} \sum_{i\in[c]} e^{z_{[i]}} \mathbb{1}_{\{l=k\}} - r x_{i[j]} e^{z_{[k]}} e^{z_{[l]}}}{\left(\sum_{i\in[c]} e^{z_{[i]}}\right)^2}$$

$$\frac{\partial^2 \mathcal{L}^{\mathrm{ce}}}{\partial W_{l[m]} \partial b_{[k]}} = \frac{1}{N} \sum_{i=1}^N \frac{r x_{i[m]} e^{z_{[k]}} \sum_{i\in[c]} e^{z_{[i]}} \mathbb{1}_{\{l=k\}} - r x_{i[m]} e^{z_{[k]}} e^{z_{[l]}}}{\left(\sum_{i\in[c]} e^{z_{[i]}}\right)^2}$$

Which means

$$\nabla^2 \mathcal{L}^{\mathrm{ce}}(W; b) = \frac{1}{N} \sum_{i=1}^N \left(\sigma(y_i) \operatorname{diag}\left(x_1^2, \ldots, x_d^2, r^2\right) - P^\top P\right)$$

where $P^{(i)} \in \mathbb{R}^{d \times 1}$ and $\sigma(x_i) \in R^c$ defined as

$$P^{(i)}_{[k]} = \begin{cases} x_{i[k]} \frac{e^{z[k]}}{\sum_{i \in [c]} e^{z[i]}} & k < d \\ r \frac{e^{z[k]}}{\sum_{i \in [c]} e^{z[i]}} & k = d \end{cases} \quad \text{and} \quad \sigma(x_i, y_i) = \frac{e^{w_{y_i}^{\top} x_i + b_{[y_i]}}}{\sum_{j \in [c]} e^{w_j^{\top} x_i + b_{[j]}}}. \tag{50}$$

Therefore,

$$\|\nabla^2 \mathcal{L}^{\text{ce}}(W; b)\|_{\text{op}} = \|\frac{1}{N} \sum_{i=1}^{N} \big( \sigma(y_i) \operatorname{diag}\big(x_1^2, \ldots, x_d^2, r^2\big) - P^{\top} P \big) \|_{\text{op}} \le \max_{i \in [N]} \|x_i\|^2 + r^2$$

as needed. $\qquad \square$

**Proposition B.10.** *Let $\mathcal{D} = \{(x_i, y_i)\}_{i=1}^{N}$ in $\mathbb{R}^d$ and let $\mathcal{D}'$ be a dataset in $\mathbb{R}^{d+1}$ that contains all data points from $\mathcal{D}$ extended by a single coordinate with a fixed value $r \ge 0$. Further, each data point $x_i \in \mathcal{D}'$ is scaled by a factor $q_i > 0$. Then, $\mathcal{L}^{\text{ce}}(W; b)$ over $\mathcal{D}'$ is $\max_{i \in [N]} q_i \big(\|x_i\|^2 + r^2\big)$ smooth.*

*Proof.* Following the same calculation as in Lemma B.9 we get that each entree of the Hessian of $\mathcal{L}^{\text{ce}}(W)$ has additional factor of $q_i^2$ as follows:

$$\nabla^2 \mathcal{L}^{\text{ce}}(W; b) = \frac{1}{N} \sum_{i=1}^{N} q_i^2 \big( \big( \sigma(y_i) \operatorname{diag}\big(x_1^2, \ldots, x_d^2, r^2\big) - P^{\top} P \big) \big)$$

where $P^{(i)} \in \mathbb{R}^{d \times 1}$ and $\sigma(x_i) \in R^c$ are defined as in eq. (50) Therefore,

$$\|\nabla^2 \mathcal{L}^{\text{ce}}(W; b)\|_{\text{op}} = \|\frac{1}{N} \sum_{i=1}^{N} q_i^2 \big( \sigma(y_i) \operatorname{diag}\big(x_1^2, \ldots, x_d^2, r^2\big) - P^{\top} P \big) \|_{\text{op}} \le \max_{i \in [N]} q_i^2 \big( \|x_i\|^2 + r^2 \big)$$

$\qquad \square$

**Proposition B.11.** *Let $\mathcal{D} = \{(x_i, y_i)\}_{i=1}^{N}$ in $\mathbb{R}^d$ and let $\{\delta_i\}_{i=1}^{c} > 0$ then, $\mathcal{L}^{\text{ce}}\Big(\operatorname{diag}\Big(\frac{1}{\delta_1}, \ldots, \frac{1}{\delta_c}\Big)W; 0\Big)$ over $\mathcal{D}$ is $\frac{\max_{i \in [N]} \|x_i\|^2}{\min_{i \in [c]} \delta_i^2}$ smooth.*

*Proof.* By the chain rule it holds that

$$\nabla^2 \mathcal{L}^{\text{ce}}\Big(\operatorname{diag}\Big(\frac{1}{\delta_1}, \ldots, \frac{1}{\delta_c}\Big)W\Big) = \operatorname{diag}\Big(\frac{1}{\delta_1}, \ldots, \frac{1}{\delta_c}\Big)^{\top} \nabla^2 \mathcal{L}^{\text{ce}}\Big(\operatorname{diag}\Big(\frac{1}{\delta_1}, \ldots, \frac{1}{\delta_c}\Big)W\Big) \operatorname{diag}\Big(\frac{1}{\delta_1}, \ldots, \frac{1}{\delta_c}\Big)$$

Therefore,

$$\|\nabla^2 \mathcal{L}^{\text{ce}}\Big(\operatorname{diag}\Big(\frac{1}{\delta_1}, \ldots, \frac{1}{\delta_c}\Big)W\Big)\|_{\text{op}} \le \frac{\max_{i \in [N]} \|x_i\|^2}{\min_{i \in [c]} \delta_i^2}$$

$\qquad \square$

# C Methods analysis

In this section we show detailed derivations for the $\tilde{\alpha}$ coefficients of each predictor we mention in Section 4. Then in Appendix D we use these expressions to lower bound the test-error of each predictor and derive near optimal parameters for the worst class error. The order of predictors we analyze in this section is: MA (Appendix C.1), LA (Appendix C.2) and CDT (Appendix C.3). Notice that MM coefficients can be easily recovered from MA's by setting $\delta = \mathbf{1}$.

## C.1 Margin adjustment

**Theorem 4.1.** *Let $\tilde{W}^{\mathrm{ma}}, \tilde{b}^{\mathrm{ma}}$ be the expected kernel approximation for the MA predictor eq. (5) with any $\delta$ that satisfies $\delta_i > 0$ for all $i \in [c]$. Then, for all $y \in [c]$ we have $\tilde{w}_y^{\mathrm{ma}} = \sum_{i=1}^c \tilde{\alpha}_{y[i]}^{\mathrm{ma}} \bar{x}_i$, where,*

$$\tilde{\alpha}_{y[i]}^{\mathrm{ma}} = \frac{\delta_i}{N_i \xi_i} \left( \mathbb{1}_{\{i=y\}} - \frac{1}{c} \right) + \frac{\sum_{j=1}^c \left[ \frac{\delta_j}{\xi_j} - \frac{\delta_y}{\xi_y} \right]}{c N_i \xi_i (M + \rho)} \quad \text{and} \quad \tilde{b}_{[y]}^{\mathrm{ma}} = \frac{\sum_{j=y}^c \left[ \frac{\delta_y}{\xi_y} - \frac{\delta_j}{\xi_j} \right]}{c (M + \rho)},$$

*using $\xi_i := \|\mu_i\|^2 + \frac{\sigma^2 d}{N_i}$ and $M := \sum_{i=1}^c \frac{1}{\xi_i}$*

*Proof.* The approximated margin adjustment predictor is

$$\{\tilde{\alpha}_i^{\mathrm{ma}}\}_{i=1}^c, \{\tilde{b}_i^{\mathrm{ma}}\}_{i=1}^c := \operatorname*{arg\,min}_{\alpha \in \mathbb{R}^c, b \in \mathbb{R}^c} \left( \sum_{y=1}^c \alpha_y^\top \bar{K} \alpha_y + \rho b_{[y]}^2 \right)$$

$$\text{subject to} \quad (\alpha_y - \alpha_k)^\top \frac{\bar{K}_{[y]}}{N_y} + b_{[y]} - b_{[k]} \geq \delta_y \quad \text{for all } y \in [c] \text{ and } k \neq y. \quad (51)$$

where $b_{[y]}$ represents the bias parameter of the $y$'th class predictor. Recall that:

$$\bar{K}_{[i,j]} = \mathbb{1}_{\{i=j\}} N_i^2 \xi_i.$$

Hence we can rewrite the optimization in eq. (51) in its explicit form:

$$\{\tilde{\alpha}_i^{\mathrm{ma}}\}_{i=1}^c, \{\tilde{b}_i^{\mathrm{ma}}\}_{i=1}^c := \operatorname*{arg\,min}_{\alpha \in \mathbb{R}^c, b \in \mathbb{R}^c} \left( \sum_{y=1}^c \sum_{i=1}^c \alpha_{y[i]}^2 N_i^2 \xi_i + \rho b_{[y]}^2 \right)$$

$$\text{subject to} \quad (\alpha_{y[y]} - \alpha_{k[y]}) N_y \xi_y + b_{[y]} - b_{[k]} \geq \delta_y \quad \text{for all } y \in [c] \text{ and } k \neq y. \quad (52)$$

Next we isolate $\alpha_{k[y]}$ of each constraint and assume equality holds:

$$-\alpha_{k[y]} = \frac{\delta_y - \alpha_{y[y]} N_y \xi_y - [b_{[y]} - b_{[k]}]}{N_y \xi_y}. \quad (53)$$

Taking the power of $\alpha_{k[y]}$:

$$\alpha_{y[i]}^2 = \frac{(\delta_i - [b_{[i]} - b_{[y]}])^2 - 2\alpha_{i[i]} N_i \xi_i (\delta_i - [b_{[y]} - b_{[i]}]) \alpha_{i[i]}^2 N_i^2 \xi_i^2}{N_i^2 \xi_i^2}, \quad (54)$$

for any pair $y \neq i$.

Let's plug in eq. (54) in the objective of eq. (52):

$$\sum_{y=1}^c \sum_{i=1}^c \alpha_{y[i]}^2 N_i^2 \xi_i + \rho b_{[y]}^2 = \sum_{y=1}^c \alpha_{y[y]}^2 N_y^2 \xi_y + \rho b_{[y]}^2 + \sum_{y=1}^c \sum_{i \neq y}^c \alpha_{y[i]}^2 N_i^2 \xi_i \quad (55)$$

$$= \sum_{y=1}^c \alpha_{y[y]}^2 N_y^2 \xi_y + \rho b_{[y]}^2$$

$$+ \sum_{y=1}^c \sum_{i \neq y}^c \frac{(\delta_i - [b_{[y]} - b_{[i]}])^2 - 2\alpha_{i[i]} N_i \xi_i (\delta_i - [b_{[y]} - b_{[i]}]) + \alpha_{i[i]}^2 N_i^2 \xi_i^2}{\xi_i} = F.$$

We continue by calculating the optimality condition for $\alpha_{y[y]}$, i.e. $\frac{\partial F}{\partial \alpha_{y[y]}} = 0$:

$$\frac{\partial F}{\partial \alpha_{y[y]}} = 2\alpha_{y[y]} N_y^2 \xi_y - 2N_y \left( [c-1]\delta_y - \left[ [c-1]b_{[y]} - \sum_{k \neq y} b_{[k]} \right] \right) + 2[c-1]\alpha_{y[y]} N_y^2 \xi_y = 0.$$

Which means that,

$$\alpha_{y[y]} = \frac{[c-1]\delta_y - \left[[c-1]b_{[y]} - \sum_{k\neq y} b_{[k]}\right]}{cN_y\xi_y}.$$

**Lemma C.1.** *Let $\tilde{\alpha}^{\mathrm{ma}}, \tilde{b}^{\mathrm{ma}}$ be optimal for eq. (51) with $\rho > 0$ then the sum of biases is zero i.e.,*

$$\sum_{i=1}^{c} \tilde{b}_i^{\mathrm{ma}} = 0.$$

*Proof.* Assume by contradiction that Lemma C.1 does not hold. Then we can use $\tilde{\alpha}^{\mathrm{ma}}, \tilde{b}^{\mathrm{ma}}$ to find a better solution (with lower objective) than $\tilde{\alpha}^{\mathrm{ma}}, \tilde{b}^{\mathrm{ma}}$ as follows:

$$\hat{\alpha}_i = \tilde{\alpha}_i^{\mathrm{ma}} \quad \text{and} \quad \hat{b}_i = \tilde{b}_i^{\mathrm{ma}} - \frac{1}{c}\sum_{j=1}^{c} \tilde{b}_j^{\mathrm{ma}},$$

clearly,

$$\sum_i^c \hat{b}_i = \sum_{i=1}^c \tilde{b}_i^{\mathrm{ma}} - \sum_{j=1}^c \tilde{b}_j^{\mathrm{ma}} = 0.$$

We'll now show that $\hat{\alpha}, \hat{b}$ is feasible and optimal in contradiction to optimality of $\tilde{\alpha}^{\mathrm{ma}}, \tilde{b}^{\mathrm{ma}}$.

**Feasibility** For any $y$ and $k \neq y$ it holds that

$$[\hat{\alpha}_y - \hat{\alpha}_k]^\top \frac{\bar{K}_y}{N_y} + \hat{b}_y - \hat{b}_k = \left[\tilde{\alpha}_y^{\mathrm{ma}} - \alpha_k^{\mathrm{ma}}\right]^\top \frac{\bar{K}_y}{N_y} + \tilde{b}_y^{\mathrm{ma}} - b_{[k]}^{\mathrm{ma}} \geq \delta_y.$$

**Optimality**

$$\begin{aligned}
\sum_{y=1}^{c} \hat{\alpha}_y^\top \bar{K} \hat{\alpha}_y + \rho\hat{b}_y^2 &= \sum_{y=1}^{c} \left(\tilde{\alpha}_y^{\mathrm{ma}}\right)^\top \bar{K}\tilde{\alpha}_y^{\mathrm{ma}} + \sum_{y=1}^{c} \rho\left(\tilde{b}_y^{\mathrm{ma}} - \frac{1}{c}\sum_{j=1}^{c}\tilde{b}_j^{\mathrm{ma}}\right)^2 \\
&= \sum_{y=1}^{c} \left(\tilde{\alpha}_y^{\mathrm{ma}}\right)^\top \bar{K}\tilde{\alpha}_y^{\mathrm{ma}} + \rho\sum_{y=1}^{c} (\tilde{b}_y^{\mathrm{ma}})^2 - 2\tilde{b}_y^{\mathrm{ma}}\frac{1}{c}\sum_{j=1}^{c}\tilde{b}_j^{\mathrm{ma}} + \frac{1}{c^2}\left(\sum_{j=1}^{c}\tilde{b}_j^{\mathrm{ma}}\right)\left(\sum_{j=1}^{c}\tilde{b}_j^{\mathrm{ma}}\right) \\
&= \sum_{y=1}^{c} \left(\tilde{\alpha}_y^{\mathrm{ma}}\right)^\top \bar{K}\tilde{\alpha}_y^{\mathrm{ma}} + \rho\left(\sum_{y=1}^{c}(\tilde{b}_y^{\mathrm{ma}})^2 - \frac{2}{c}\sum_{y=1}^{c}\tilde{b}_y^{\mathrm{ma}}\sum_{j=1}^{c}\tilde{b}_j^{\mathrm{ma}} + \frac{1}{c}\left(\sum_{j=1}^{c}\tilde{b}_j^{\mathrm{ma}}\right)\left(\sum_{j=1}^{c}\tilde{b}_j^{\mathrm{ma}}\right)\right) \\
&= \sum_{y=1}^{c} \left(\tilde{\alpha}_y^{\mathrm{ma}}\right)^\top \bar{K}\tilde{\alpha}_y^{\mathrm{ma}} + \rho\sum_{y=1}^{c}(\tilde{b}_y^{\mathrm{ma}})^2 - \frac{\rho}{c}\left(\sum_{j=1}^{c}\tilde{b}_j^{\mathrm{ma}}\right)^2 \\
&\stackrel{\text{assumption}}{<} \sum_{y=1}^{c} \left(\tilde{\alpha}_y^{\mathrm{ma}}\right)^\top \bar{K}\tilde{\alpha}_y^{\mathrm{ma}} + \rho\sum_{y=1}^{c}(\tilde{b}_y^{\mathrm{ma}})^2
\end{aligned}$$

In contradiction to optimality of $\tilde{\alpha}^{\mathrm{ma}}, \tilde{b}^{\mathrm{ma}}$. $\qquad\square$

From Lemma C.1 it holds that in case $\rho > 0$ then $\tilde{b}^{\mathrm{ma}}$ must satisfy $\tilde{b}_y^{\mathrm{ma}} = -\sum_{k\neq y} b_{[k]}^{\mathrm{ma}}$. In case $\rho = 0$ we can always take the optimal biases $\tilde{b}^{\mathrm{ma}}$ and normalize them (by subtracting a constant from all the biases)

similarly to Lemma C.1 and get that $\tilde{b}_y^{\mathrm{ma}} = -\sum_{k\neq y} b_{[k]}^{\mathrm{ma}}$. Hence, we add this as "implicit" constraint to the optimization problem and get that eq. (55) can be simplified as follows:

$$\alpha_{y[y]} = \frac{[c-1]\delta_y - cb_{[y]}}{cN_y\xi_y}. \tag{56}$$

Continue by plugging $\alpha_{y_{[y]}}$ in eq. (53) to conclude $\alpha_{k[y]}$

$$
\begin{aligned}
-\alpha_{k[y]} &= \frac{\delta_y - \alpha_{y[y]}N_y\xi_y - \left[b_{[y]} - b_{[k]}\right]}{N_y\xi_y} \\
&= \frac{\delta_y - \frac{[c-1]\delta_y - cb_{[y]}}{cN_y\xi_y}N_y\xi_y - \left[b_{[y]} - b_{[k]}\right]}{N_y\xi_y} \\
&= \frac{\delta_y + cb_{[k]}}{cN_y\xi_y}.
\end{aligned} \tag{57}
$$

Now all its left is to find a set of biases that satisfy the optimality conditions ($\frac{\partial F}{b_{[y]}} = 0$).

Let's plug in $\alpha_{k[y]}$ and $\alpha_{y[y]}$ in $F$:

$$
\sum_{y=1}^{c}\alpha_{y[y]}^2 N_y^2\xi_y + \rho b_{[y]}^2 + \sum_{y=1}^{c}\sum_{i\neq y}^{c}\alpha_{y[i]}^2 N_i^2\xi_i =
$$
$$
\sum_{y=1}^{c}\left(\frac{[c-1]^2\delta_y^2 - 2cb_{[y]}[c-1]\delta_y + c^2b_{[y]}^2}{c^2N_y^2\xi_y^2}\right)N_y^2\xi_y + \rho b_{[y]}^2 + \sum_{y=1}^{c}\sum_{i\neq y}^{c}\left(\frac{\delta_i^2 + 2cb_{[y]}\delta_i + c^2b_{[y]}^2}{c^2N_i^2\xi_i^2}\right)N_i^2\xi_i =
$$
$$
\sum_{y=1}^{c}\left(\frac{[c-1]^2\delta_y^2 - 2cb_{[y]}[c-1]\delta_y + c^2b_{[y]}^2}{c^2\xi_y}\right) + \rho b_{[y]}^2 + \sum_{y=1}^{c}\sum_{i\neq y}^{c}\left(\frac{\delta_i^2 + 2cb_{[y]}\delta_i + c^2b_{[y]}^2}{c^2\xi_i}\right)
$$

Moving on to calculate $\frac{\partial F}{b_{[y]}}$ and find $\tilde{b}_y^{\mathrm{ma}}$:

$$\frac{\partial F}{\partial b_{[y]}} = \frac{-2[c-1]\delta_y + 2cb_{[y]}}{c\xi_y} + \frac{2\rho c\xi_y b_{[y]}}{c\xi_y} + \sum_{i\neq y}\frac{2\delta_i}{c\xi_i} + 2b_y\sum_{i\neq y}\frac{c}{c\xi_i} = 0$$

Denote $\bar{\xi}_i = \prod_{j\neq i}\xi_j$ then,

$$\frac{\partial F}{\partial b_{[y]}} = -[c-1]\delta_y\bar{\xi}_y + cb_{[y]}\left[1 + \rho\xi_y\right]\bar{\xi}_y + \sum_{i\neq y}\delta_i\bar{\xi}_i + b_{[y]}\sum_{i\neq y}c\bar{\xi}_i = 0$$

which means

$$\tilde{b}_y^{\mathrm{ma}} = \frac{[c-1]\frac{\delta_y}{\xi_y} - \sum_{i\neq y}\frac{\delta_i}{\xi_i}}{c\left(\sum_{i=1}^{c}\frac{1}{\xi_i} + \rho\right)} = \frac{\sum_{i=1}^{c}\frac{\delta_y}{\xi_y} - \frac{\delta_i}{\xi_i}}{c(\sum_{i=1}^{c}M + \rho)}. \tag{58}$$

Finally, let us plug in eq. (58) in eq. (56) and eq. (57) and conclude

$$\alpha_{y[y]}^{\mathrm{ma}} = \frac{\rho[c-1]\delta_y + \sum_{i\neq y}^{c}\frac{1}{\xi_i}\left[[c-1]\delta_y + \delta_i\right]}{cN_y\xi_y(M + \rho)} \tag{59}$$

and

$$\alpha_{k[y]}^{\mathrm{ma}} = -\frac{\rho\delta_y + \frac{\delta_y + [c-1]\delta_k}{\xi_k} + \sum_{i=1}^{c}\frac{\delta_y - \delta_i}{\xi_i}}{cN_y\xi_y(M + \rho)}. \tag{60}$$

Which yields:

$$\alpha_{y[i]}^{\mathrm{ma}} = \frac{\delta_i}{N_i\xi_i}\left(\mathbb{1}_{\{i=y\}} - \frac{1}{c}\right) + \frac{\sum_{j=1}^{c}\left(\frac{\delta_j}{\xi_j} - \frac{\delta_y}{\xi_y}\right)}{cN_i\xi_i(M + \rho)}$$

where $M := \sum_{i=1}^{c}\frac{1}{\xi_i}$. $\qquad\square$

## C.2 Logit adjustment

As explained in Appendix B at convergence of gradient descent LA converges to the maximum margin with bias adjustment of $-\iota$. Therefore its $\tilde{\alpha}$ and $\tilde{b}$ are the same as MM with bias adjustment of $-\iota$. From Appendix C.1

$$\alpha_{y[i]}^{\text{mm}} = \frac{1}{N_i \xi_i}\left(\mathbb{1}_{\{i=y\}} - \frac{1}{c}\right) + \frac{\sum_{j=1}^{c}\left(\frac{1}{\xi_j} - \frac{1}{\xi_y}\right)}{cN_i \xi_i (M + \rho)} \quad\text{and}\quad \tilde{b}_y^{\text{la}} = \frac{\sum_{i=1}^{c}\frac{1}{\xi_y} - \frac{1}{\xi_i}}{c(\sum_{i=1}^{c} M + \rho)} - \iota_y \tag{61}$$

where $M \coloneqq \sum_{i=1}^{c}\frac{1}{\xi_i}$.

## C.3 CDT

**Theorem 4.5.** *Let $\tilde{W}^{\text{cdt}}$ be the expected kernel approximation for the CDT predictor eq. (6) with any $\delta$ that satisfies $\delta_i > 0$ for all $i \in [c]$, using $\rho = \infty$. Then, for all $y \in [c]$ we have $\tilde{w}_y^{\text{cdt}} = \sum_{i=1}^{c}\tilde{\alpha}_{y[i]}^{\text{cdt}}\bar{x}_i$, where (for $\xi_i = \|\mu_i\|^2 + \frac{\sigma^2 d}{N_i}$),*

$$\tilde{\alpha}_{y[i]}^{\text{cdt}} = \frac{\delta_i}{N_i \xi_i}\left(\mathbb{1}_{\{i=y\}} - \frac{\delta_y \delta_i}{\sum_{j=1}^{c}\delta_j^2}\right).$$

*Proof.* Recall the expected kernel approximation of the CDT predictor at $\rho = \infty$ (learning without bias):

$$\{\tilde{\alpha}_i^{\text{cdt}}\}_{i=1}^{c} \coloneqq \underset{\alpha \in \mathbb{R}^c}{\arg\min}\left(\sum_{y=1}^{c}\alpha_y^{\top}\bar{K}\alpha_y\right)$$

$$\text{subject to}\quad \left(\frac{\alpha_y}{\delta_y} - \frac{\alpha_k}{\delta_k}\right)^{\top}\frac{\bar{K}_{[y]}}{N_y} \geq 1 \quad\text{for all}\quad y \in [c] \text{ and } k \neq y. \tag{62}$$

We repeat the same analysis as in (C.1).

First we explicitly write the optimization problem:

$$\{\tilde{\alpha}_i^{\text{cdt}}\}_{i=1}^{c} \coloneqq \underset{\alpha \in \mathbb{R}^c}{\arg\min}\left(\sum_{y=1}^{c}\sum_{i=1}^{c}\alpha_{y[i]}^2 N_i^2 \xi_i\right) \tag{63}$$

$$\text{subject to}\quad \left(\frac{\alpha_{y[y]}}{\delta_y} - \frac{\alpha_{k[y]}}{\delta_k}\right)N_y \xi_y \geq 1 \quad\text{for all}\quad y \in [c] \text{ and } k \neq y.$$

Next we assume the constraints are tight and get:

$$-\alpha_{k[y]} = \frac{\delta_y \delta_k - \alpha_{y[y]} N_y \xi_y \delta_k}{\delta_y N_y \xi_y}. \tag{64}$$

Plug in eq. (64) in the objective of eq. (63):

$$\min_{\alpha}\sum_{y=1}^{c}\sum_{i=1}^{c}\alpha_{y[i]}^2 N_i^2 \xi_i = \min_{\alpha}\sum_{i=1}^{c}\alpha_{i[i]}^2 N_i^2 \xi_i + \sum_{i=1}^{c}\sum_{y\neq i}^{c}\alpha_{i[y]}^2 N_y^2 \xi_y$$

$$= \min_{\alpha}\sum_{i=1}^{c}\alpha_{i[i]}^2 N_i^2 \xi_i + \sum_{i=1}^{c}\sum_{y\neq i}^{c}\left(\frac{\delta_i^2 \delta_y^2 - 2\delta_i^2 \delta_y \alpha_{y[y]} N_y \xi_y + \alpha_{y[y]}^2 \delta_i^2 N_y^2 \xi_y^2}{\delta_y^2 \xi_y}\right)$$

$$= \min_{\alpha} F(\alpha).$$

Moving on to calculate the optimality condition of $F$ for $\alpha_{y[y]}$:

$$\frac{\partial F}{\partial \alpha_{y[y]}} = 2\alpha_{y[y]}N_y^2\xi_y\delta_y^2 - 2\sum_{i\neq y}^{c}\delta_i^2\delta_y N_y + 2\sum_{i\neq y}^{c}\alpha_{y[y]}\delta_i^2 N_y^2\xi_y = 0$$

$$\Longleftrightarrow$$

$$\alpha_{y[y]} = \frac{\sum_{i\neq y}\delta_i^2\delta_y}{N_y\xi_y\sum_{i=1}^{c}\delta_i^2}.$$

Denote

$$\Delta^{\backslash y} := \sum_{i\neq y}\delta_i^2 \quad\text{and}\quad \Delta := \sum_{i=1}^{c}\delta_i^2$$

Then

$$\tilde{\alpha}_{y[y]}^{\text{cdt}} = \frac{\Delta^{\backslash y}\delta_y}{\Delta N_y\xi_y} \quad\text{and}\quad \tilde{\alpha}_{y[i]}^{\text{cdt}} = -\frac{\delta_y\delta_i^2}{\Delta N_y\xi_y} \tag{65}$$

where $\tilde{\alpha}_{i[y]}^{\text{cdt}}$ was obtained by plugging $\tilde{\alpha}_{y[y]}^{\text{cdt}}$ in eq. (64). Hence

$$\tilde{\alpha}_{y[i]}^{\text{cdt}} = \frac{\delta_i}{N_i\xi_i}\left(\mathbb{1}_{\{i=y\}} - \frac{\delta_y\delta_i}{\Delta}\right)$$

Which means we find a solution that satisfy the constraints and the optimality condition and due to convexity this is sufficient to know we find the optimum. □

# D  Error approximation

In this section we show how to use the $\tilde{\alpha}$ and $\tilde{b}$ expressions of each predictor (from Appendix C) to derive the error score parameters. Then for MA and LA we use the error expressions to derive near optimal hyper-parameters for the worst class error. The order of predictors in this section is: MA, MM, LA and CDT.

## D.1  Margin adjustment

We show the derivation for $\hat{\nu}^{(y)}$ and $\hat{\Sigma}^{(y)}$ using the optimal solution of the approximated margin problem, eq. (51).

$$\alpha_{y[i]}^{\text{ma}} = \frac{\delta_i}{N_i\xi_i}\left(\mathbb{1}_{\{i=y\}} - \frac{1}{c}\right) + \frac{\sum_{j=1}^{c}\left[\frac{\delta_j}{\xi_j} - \frac{\delta_y}{\xi_y}\right]}{cN_i\xi_i(M+\rho)} \quad\text{and}\quad \tilde{b}_{[y]}^{\text{ma}} = \frac{\sum_{i=1}^{c}\left[\frac{\delta_y}{\xi_y} - \frac{\delta_i}{\xi_i}\right]}{c(M+\rho)}.$$

Therefore,

$$\tilde{\alpha}_{y[i]}^{\text{ma}} - \tilde{\alpha}_{k[i]}^{\text{ma}} = \frac{\delta_i}{N_i\xi_i}\left(\mathbb{1}_{\{i=y\}} - \mathbb{1}_{\{i=k\}}\right) + \frac{\Delta_k}{N_i\xi_i(M+\rho)},$$

and

$$\tilde{b}_y^{\text{ma}} - \tilde{b}_k^{\text{ma}} = -\frac{\Delta_k}{(M+\rho)},$$

where

$$\Delta_k := \frac{\delta_k}{\xi_k} - \frac{\delta_y}{\xi_y} \quad\text{and}\quad M := \sum_{i=1}^{c}\frac{1}{\xi_i}.$$

Consequently eq. (29)

$$\hat{\Sigma}_{[k,k']} = \sum_i \left( \alpha_{y[i]}^{\mathrm{ma}} - \alpha_{k[i]}^{\mathrm{ma}} \right) \left( \alpha_{y[i]}^{\mathrm{ma}} - \alpha_{k'[i]}^{\mathrm{ma}} \right) N_i^2 \xi_i$$

$$= A_{[k,k']} + \frac{1}{M+\rho} \left( B_{[k,k']} + B_{[k',k]} \right) + \frac{1}{(M+\rho)^2} C_{[k,k']},$$

where

$$A_{[k,k']} = \sum_i \frac{\delta_i^2}{\xi_i} \left( \mathbb{1}_{\{i=y\}} - \mathbb{1}_{\{i=k\}} \right) \left( \mathbb{1}_{\{i=y\}} - \mathbb{1}_{\{i=k'\}} \right) = \frac{\delta_y^2}{\xi_y} + \frac{\delta_k^2}{\xi_k} \mathbb{1}_{\{k=k'\}}$$

$$B_{[k,k']} = \sum_i \frac{\delta_i}{\xi_i} \left( \mathbb{1}_{\{i=y\}} - \mathbb{1}_{\{i=k\}} \right) \Delta_{k'} = \left( \frac{\delta_y}{\xi_y} - \frac{\delta_k}{\xi_k} \right) \Delta_{k'} = -\Delta_k \Delta_{k'} = B_{[k',k]}$$

$$C_{[k,k']} = \sum_i \frac{1}{\xi_i} \Delta_k \Delta_{k'} = M \Delta_k \Delta_{k'}.$$

Substituting back, we have

$$\hat{\Sigma}_{[k,k']} = \frac{\delta_y^2}{\xi_y} + \frac{\delta_k^2}{\xi_k} \mathbb{1}_{\{k=k'\}} - \Delta_k \Delta_{k'} \left( \frac{2}{M+\rho} - \frac{M}{(M+\rho)^2} \right)$$

$$= \frac{\delta_y^2}{\xi_y} + \frac{\delta_k^2}{\xi_k} \mathbb{1}_{\{k=k'\}} - \frac{M+2\rho}{(M+\rho)^2} \left( \frac{\delta_k}{\xi_k} - \frac{\delta_y}{\xi_y} \right) \left( \frac{\delta_{k'}}{\xi_{k'}} - \frac{\delta_y}{\xi_y} \right)$$

$$= \frac{\delta_y^2}{\xi_y} + \frac{\delta_k^2}{\xi_k} \mathbb{1}_{\{k=k'\}} + \psi_{k,k'}(\rho) \tag{66}$$

Now look at $\hat{\nu}^{(y)}$, from eq. (29):

$$\hat{\nu}_{[i]}^{(y)} = \frac{1}{\sigma} \left( \left( \tilde{\alpha}_{y[y]}^{\mathrm{ma}} - \tilde{\alpha}_{i[y]}^{\mathrm{ma}} \right) N_y \|\mu_y\|^2 + \tilde{b}_y^{\mathrm{ma}} - \tilde{b}_i^{\mathrm{ma}} \right) \overset{(1)}{=} s_y \frac{\delta_y}{\xi_y} + \phi_i^{\mathrm{ma}}(\rho) \tag{67}$$

where,

$$\phi_i^{\mathrm{ma}}(\rho) = \left( \frac{s_y}{\xi_y} - \sqrt{d} \right) \frac{\Delta_i}{(M+\rho)} \text{ and } \psi_{k,k'}^{\mathrm{ma}}(\rho) = -\frac{M+2\rho}{(M+\rho)^2} \left( \frac{\delta_k}{\xi_k} - \frac{\delta_y}{\xi_y} \right) \left( \frac{\delta_{k'}}{\xi_{k'}} - \frac{\delta_y}{\xi_y} \right)$$

(1) holds since $\frac{1}{\sigma} = \sqrt{d}$ and $\|\mu_y\|^2 = \frac{s_y}{\sqrt{d}}$. Therefore,

$$\lim_{d \to \infty} \mathrm{Err}_{y \to k}(\tilde{W}^{\mathrm{ma}}, \tilde{b}^{\mathrm{ma}}) = Q \left( \frac{s_y N_y \tilde{\delta}_y + \tilde{\phi}_k^{\mathrm{ma}}(\rho)}{\sqrt{\tilde{\delta}_y^2 N_y + \tilde{\delta}_k^2 N_k + \tilde{\psi}_{k,k}^{\mathrm{ma}}(\rho)}} \right)$$

where $\tilde{\delta}_i := \lim_{d \to \infty} \delta_i$.

*Remark* D.1. The expressions for the maximum margin predictor are obtained by setting $\delta_i = 1$.

Recall that the lower bound for the worst class error of MA is defined by:

$$\widetilde{\mathrm{Err}}_{\mathrm{wc}}^{\mathrm{ma}}(\delta, \rho) := \max_{k \neq y} \lim_{d \to \infty} \mathrm{Err}_{y \to k}\left( \tilde{W}^{\mathrm{ma}}(\delta, \rho), \tilde{b}^{\mathrm{ma}}(\delta, \rho) \right).$$

We now continue by showing how to find near-optimal margins for promoting fairness by the MA predictor.

**Near optimal margins.**

**Theorem D.2.** *For any dataset $\mathcal{D} \sim \mathcal{P}\left( \{s_i, N_i\}_{i=1}^c \right)$ it holds that*

$$\delta_i^\star := \frac{\xi_i}{\|\mu_i\|^2 + 2(M+\rho)^{-1}} \in \arg\min_\delta \widetilde{\mathrm{Err}}_{\mathrm{wc}}^{\mathrm{ma}}(\delta, \rho),$$

*Proof.* We would like to find a set of margins that minimizes the worst class error, eq. (2). To do so, we are looking for a set of margins, $\{\delta_i\}_{i=1}^c$ that minimizes the lower bound of the worst class error at the limit of $d \to \infty$. To find such margins we would have to analyze $\widetilde{\mathrm{Err}}_{\mathrm{wc}}^{\mathrm{ma}}(\delta, \rho)$ at a finite dimension, $d$, since the order of limits of $\rho \to \infty$ and $d \to \infty$ matters.

$$
\min_{\delta} \max_{k \neq y} Q \left( \frac{\frac{s_y}{\xi_y}\left[\frac{\delta_k}{\xi_k} + \delta_y\left(M^{\backslash y} + \rho\right)\right] + \sqrt{d}\left[\frac{\delta_y}{\xi_y} - \frac{\delta_k}{\xi_k}\right]}{\sqrt{\frac{\left(\frac{\delta_k}{\xi_k} + \delta_y\left[M^{\backslash y} + \rho\right]\right)^2}{\xi_y} + \frac{\left(\frac{\delta_y}{\xi_y} + \delta_k\left[M^{\backslash k} + \rho\right]\right)^2}{\xi_k} + \left(\frac{\delta_k}{\xi_k} - \frac{\delta_y}{\xi_y}\right)^2 M^{\backslash y,k}}} \right).
$$

where $M^{\backslash y} := \sum_{i \neq y}^c \frac{1}{\xi_i}$.

By the monotonicity of the Q-function this is equivalent to find a set of margins that maximizes:

$$
\max_{\delta} \min_{y} \min_{y \neq k} \frac{\frac{s_y}{\xi_y}\left[\frac{\delta_k}{\xi_k} + \delta_y\left(M^{\backslash y} + \rho\right)\right] + \sqrt{d}\left[\frac{\delta_y}{\xi_y} - \frac{\delta_k}{\xi_k}\right]}{\sqrt{\frac{\left(\frac{\delta_k}{\xi_k} + \delta_y\left[M^{\backslash y} + \rho\right]\right)^2}{\xi_y} + \frac{\left(\frac{\delta_y}{\xi_y} + \delta_k\left[M^{\backslash k} + \rho\right]\right)^2}{\xi_k} + \left(\frac{\delta_k}{\xi_k} - \frac{\delta_y}{\xi_y}\right)^2 M^{\backslash y,k}}} \tag{68}
$$

We start by explaining why $\delta^\star$ (that satisfy eq. (68)) equalizes each $\delta_y, \delta_k$ dependent pair:

$$
\frac{s_y}{\xi_y}\left[\frac{\delta_k}{\xi_k} + \delta_y\left(M^{\backslash y} + \rho\right)\right] + \sqrt{d}\left[\frac{\delta_y}{\xi_y} - \frac{\delta_k}{\xi_k}\right] = \frac{s_k}{\xi_k}\left[\frac{\delta_y}{\xi_y} + \delta_k\left(M^{\backslash k} + \rho\right)\right] + \sqrt{d}\left[\frac{\delta_k}{\xi_k} - \frac{\delta_y}{\xi_y}\right]. \tag{69}
$$

*Remark* D.3. Notice that denominator of eq. (68) is symmetric for $y$ and $k$, hence we can analyze the numerators.

First we note that there is always a set of positive margins that satisfy eq. (69). Let $\delta_i' = \frac{\delta_i}{\xi_i}$ and plug in $\delta_y', \delta_k'$ in eq. (69).

$$
\frac{s_y \delta_k'}{\xi_y} + s_y \delta_y'\left(M^{\backslash y} + \rho\right) + \sqrt{d}\left[\delta_y' - \delta_k'\right] = \frac{s_k \delta_y'}{\xi_k} + s_k \delta_k'\left(M^{\backslash k} + \rho\right) + \sqrt{d}\left[\delta_k' - \delta_y'\right]. \tag{70}
$$

Let $\delta_k = \xi_k \implies \delta_k' = 1$ and substitute in eq. (70).

$$
\frac{s_y}{\xi_y} + s_y \delta_y'\left(M^{\backslash y} + \rho\right) + \sqrt{d}\left[\delta_y' - 1\right] = \frac{s_k \delta_y'}{\xi_k} + s_k\left(M^{\backslash k} + \rho\right) + \sqrt{d}\left[1 - \delta_y'\right].
$$

Which yields:

$$
\delta_y' = \frac{s_k(M + \rho) + 2\sqrt{d} - \frac{s_y}{\xi_y} - \frac{s_k}{\xi_k}}{s_y(M + \rho) + 2\sqrt{d} - \frac{s_y}{\xi_y} - \frac{s_k}{\xi_k}}.
$$

In addition, $\delta_y' > 0$ since:

$$
2\sqrt{d} - \frac{s_y}{\xi_y} - \frac{s_k}{\xi_k} \overset{(1)}{=} 2\sqrt{d} - \frac{\sqrt{d}}{1 + \frac{\sqrt{d}}{s_y N_y}} - \frac{\sqrt{d}}{1 + \frac{\sqrt{d}}{s_k N_k}} > 0.
$$

where (1) holds since $\xi_i := \|\mu_i\|^2 + \frac{\sigma^2 d}{N_i} = \frac{s_i}{\sqrt{d}} + \frac{1}{N_i}$. Note that this set of margins is not necessarily feasible for the case of $c > 2$ (multiclass). However, we later show that we can make $\delta_i$ depends only on its parameters which is asymptotically optimal. We continue by showing that the error term of the $y$'th class is monotonically increasing in $\delta_y'$ and that the error term of the $k$'th class is monotonically decreasing in $\delta_y'$ which means that the optimum of eq. (68) is obtained at the intersection of the error term functions. We start by rearranging the term inside the Q-function of eq. (68) as follows (for $\delta_y'$ and $\delta_k'$ as described above):

$$
\frac{s_y \delta_y'(M + \rho) + \left[\sqrt{d} - \frac{s_y}{\xi_y}\right]\left[\delta' - 1\right]}{\sqrt{\frac{\left(1 + \delta_y' \xi_y\left[M^{\backslash y} + \rho\right]\right)^2}{\xi_y} + \frac{\left(\delta_y' + \xi_k\left[M^{\backslash k} + \rho\right]\right)^2}{\xi_k} + \left(1 - \delta_y'\right)^2 M^{\backslash y,k}}}
$$

Let $f(\delta'_y)$ and $g(\delta'_y)$ be defined as the functions of the $\delta_y, \delta_k$ dependent terms as follows:

$$f(\delta'_y) := \frac{s_y\delta'_y(M+\rho) + \left[\sqrt{d} - \frac{s_y}{\xi_y}\right][\delta'_y - 1]}{\sqrt{\frac{\left(1+\delta'_y\xi_y\left[M^{\setminus y}+\rho\right]\right)^2}{\xi_y} + \frac{\left(\delta'_y + \xi_k\left[M^{\setminus k}+\rho\right]\right)^2}{\xi_k} + \left(1-\delta'_y\right)^2 M^{\setminus y,k}}} = \frac{f_1(\delta'_y)}{h(\delta'_y)},$$

and,

$$g(\delta'_y) := \frac{s_k(M+\rho) + \left[\sqrt{d} - \frac{s_k}{\xi_k}\right][1 - \delta'_y]}{\sqrt{\frac{\left(1+\delta'_y\xi_y\left[M^{\setminus y}+\rho\right]\right)^2}{\xi_y} + \frac{\left(\delta'_y + \xi_k\left[M^{\setminus k}+\rho\right]\right)^2}{\xi_k} + \left(1-\delta'_y\right)^2 M^{\setminus y,k}}} = \frac{g_1(\delta'_y)}{h(\delta'_y)}.$$

Starting by looking at the derivative of $h(\delta'_y)$:

$$h'(\delta'_y) = \left(\frac{\left(1+\delta'_y\xi_y\left[M^{\setminus y}+\rho\right]\right)^2}{\xi_y} + \frac{\left(\delta'_y + \xi_k\left[M^{\setminus k}+\rho\right]\right)^2}{\xi_k} + \left(1-\delta'_y\right)^2 M^{\setminus y,k}\right)^{-1/2} \times$$

$$\left(\left(1+\delta'_y\xi_y\left[M^{\setminus y}+\rho\right]\right)\left[M^{\setminus y}+\rho\right] + \frac{\left(\delta'_y + \xi_k\left[M^{\setminus k}+\rho\right]\right)}{\xi_k} - (1-\delta'_y)M^{\setminus y,k}\right)$$

$$= \underbrace{\left(\frac{\left(1+\delta'_y\xi_y\left[M^{\setminus y}+\rho\right]\right)^2}{\xi_y} + \frac{\left(\delta'_y + \xi_k\left[M^{\setminus k}+\rho\right]\right)^2}{\xi_k} + \left(1-\delta'_y\right)^2 M^{\setminus y,k}\right)^{-1/2}}_{\geq 0} \times$$

$$\underbrace{\left(\left(1+\delta'_y\xi_y\left[M^{\setminus y}+\rho\right]\right)\left[M^{\setminus y}+\rho\right] + \frac{\left(\delta'_y + \xi_k\rho\right)}{\xi_k} + \delta'_y M^{\setminus y,k} + \frac{1}{\xi_y}\right)}_{\geq 0} \geq 0$$

Therefore $h$ is monotonically increasing in $\delta'_y$.

We continue by showing that $f_1$ and $g_2$ are monotonically increasing and decreasing in $\delta'_y$ respectively.

$$f'(\delta'_y) = s_y(M+\rho) + \left[\sqrt{d} - \frac{s_y}{\xi_y}\right] \overset{(*)}{>} 0$$

and

$$g'(\delta'_y) = -\left[\sqrt{d} - \frac{s_k}{\xi_k}\right] \overset{(*)}{<} 0$$

(*) for large enough $d$, which means that $f$ and $g$ are monotonically increasing and decreasing in $\delta'_y$ respectively.

Therefore, the optimal $\delta$ must satisfy eq. (69) which is obtained by

$$\delta'_y = \frac{s_k(M+\rho) + 2\sqrt{d} - \frac{s_y}{\xi_y} - \frac{s_k}{\xi_k}}{s_y(M+\rho) + 2\sqrt{d} - \frac{s_y}{\xi_y} - \frac{s_k}{\xi_k}} \quad \text{and} \quad \delta'_k = 1.$$

In order to make it symmetric we can multiply both deltas by $\frac{1}{s_k(M+\rho)+2\sqrt{d}-\frac{s_y}{\xi_y}-\frac{s_k}{\xi_k}}$ and obtain

$$\delta'_y = \frac{1}{s_y(M+\rho) + 2\sqrt{d} - \frac{s_y}{\xi_y} - \frac{s_k}{\xi_k}} \quad \text{and} \quad \delta'_k = \frac{1}{s_k(M+\rho) + 2\sqrt{d} - \frac{s_y}{\xi_y} - \frac{s_k}{\xi_k}}.$$

Which means that:

$$\delta_y^\star = \frac{\xi_y}{\frac{s_y(M+\rho)}{2\sqrt{d}} + 1 - \frac{s_y}{2\sqrt{d}\xi_y} - \frac{s_k}{2\sqrt{d}\xi_k}} \quad \text{and} \quad \delta_k^\star = \frac{\xi_k}{\frac{s_k(M+\rho)}{2\sqrt{d}} + 1 - \frac{s_y}{2\sqrt{d}\xi_y} - \frac{s_k}{2\sqrt{d}\xi_k}}.$$

Since $\delta_y^\star$ can not be dependent on $\xi_k$ and $s_k$ for all the classes $k \neq y$ we neglect the $-\frac{s_y}{2\sqrt{d}\xi_y} - \frac{s_k}{2\sqrt{d}\xi_k}$ factor. This is valid since when looking at eq. (68) at the limit of $d \to \infty$ the only term that is affected by this factor is $\sqrt{d}\left[\frac{\delta_y^\star}{\xi_y} - \frac{\delta_k^\star}{\xi_k}\right]$ (in any other case this factor vanishes). Additionally notice that when we plug in $\delta^\star$ in $\sqrt{d}\left[\frac{\delta_y^\star}{\xi_y} - \frac{\delta_k^\star}{\xi_k}\right]$ we get:

$$\sqrt{d}\left[\frac{\delta_y^\star}{\xi_y} - \frac{\delta_k^\star}{\xi_k}\right] = \sqrt{d}\left[\frac{1}{\frac{s_y(M+\rho)}{2\sqrt{d}} + 1 - \frac{s_y}{2\sqrt{d}\xi_y} - \frac{s_k}{2\sqrt{d}\xi_k}} - \frac{1}{\frac{s_k(M+\rho)}{2\sqrt{d}} + 1 - \frac{s_y}{2\sqrt{d}\xi_y} - \frac{s_k}{2\sqrt{d}\xi_k}}\right]$$

$$= \sqrt{d}\left[\frac{s_k(M+\rho)}{2\sqrt{d}} + 1 - \frac{s_y}{2\sqrt{d}\xi_y} - \frac{s_k}{2\sqrt{d}\xi_k} - \frac{s_y(M+\rho)}{2\sqrt{d}} - 1 + \frac{s_y}{2\sqrt{d}\xi_y} + \frac{s_k}{2\sqrt{d}\xi_k}\right]$$

$$= \sqrt{d}\left[\frac{s_k(M+\rho)}{2\sqrt{d}} - \frac{s_y(M+\rho)}{2\sqrt{d}}\right]$$

Which is essentially the same expression that is obtained for

$$\hat{\delta}_y = \frac{\xi_y}{\|\mu_y\|^2 + 2(M+\rho)^{-1}}$$

Therefore, at $d \to \infty$ the error terms of $\delta^\star$ and $\hat{\delta}$ converge to the same value and

$$\frac{\xi_y}{\|\mu_y\|^2 + 2(M+\rho)^{-1}} \in \arg\min_\delta \widetilde{\mathrm{Err}}_{\mathrm{wc}}^{\mathrm{ma}}(\delta, \rho).$$

$\square$

Following Theorem D.2 we evaluate the lower bound of the worst class error of MA at $\delta^\star$ when $\rho = \infty$ and $\rho < \infty$ i.e., a predictor without and with bias respectively. Recall that

$$\lim_{d\to\infty} \mathrm{Err}_{y\to k}(\tilde{W}^{\mathrm{ma}}, \tilde{b}^{\mathrm{ma}}) = Q\left(\frac{s_y N_y \tilde{\delta}_y + \tilde{\phi}_k^{\mathrm{ma}}(\rho)}{\sqrt{\tilde{\delta}_y^2 N_y + \tilde{\delta}_k^2 N_k + \tilde{\psi}_{k,k}^{\mathrm{ma}}(\rho)}}\right),$$

where

$$\phi_i^{\mathrm{ma}}(\rho) = \left(\frac{s_y}{\xi_y} - \sqrt{d}\right)\frac{\frac{\delta_i}{\xi_k} - \frac{\delta_y}{\xi_y}}{M+\rho} \quad \text{and} \quad \psi_{k,k'}^{\mathrm{ma}}(\rho) = -\frac{M+2\rho}{(M+\rho)^2}\left(\frac{\delta_k}{\xi_k} - \frac{\delta_y}{\xi_y}\right)\left(\frac{\delta_{k'}}{\xi_{k'}} - \frac{\delta_y}{\xi_y}\right).$$

Notice that for $\rho = \infty$ it holds that:

$$\phi_i^{\mathrm{ma}}(\infty) = 0 \quad, \quad \psi_{k,k'}^{\mathrm{ma}}(\infty) = 0 \quad \text{and} \quad \lim_{d\to\infty} \mathrm{Err}_{y\to k}(\tilde{W}^{\mathrm{ma}}, \tilde{b}^{\mathrm{ma}}) = Q\left(\frac{1}{\sqrt{\frac{1}{s_y^2 N_y} + \frac{1}{s_k^2 N_k}}}\right).$$

For the case of $\rho < \infty$ we get when looking at the error term at a finite $d$ it hold that:

$$\frac{s_y N_y \frac{\xi_y}{\|\mu_y\|^2 + 2(M+\rho)^{-1}} + \phi_k(\rho)}{\sqrt{\left(\frac{\xi_y}{\|\mu_y\|^2 + 2(M+\rho)^{-1}}\right)^2 N_y + \left(\frac{\xi_k}{\|\mu_k\|^2 + 2(M+\rho)^{-1}}\right)^2 N_k + \psi_{y,k}(\rho)}} =$$

$$\frac{s_y N_y \frac{\frac{s_y}{\sqrt{d}} + \frac{1}{N_y}}{\frac{s_y}{\sqrt{d}} + 2(M+\rho)^{-1}} - \sqrt{d}\left[\frac{1}{\frac{s_k}{\sqrt{d}} + 2(M+\rho)^{-1}} - \frac{1}{\frac{s_y}{\sqrt{d}} + 2(M+\rho)^{-1}}\right](M+\rho)^{-1}}{\sqrt{\left(\frac{\frac{s_y}{\sqrt{d}} + \frac{1}{N_y}}{\frac{s_y}{\sqrt{d}} + 2(M+\rho)^{-1}}\right)^2 N_y + \left(\frac{\frac{s_k}{\sqrt{d}} + \frac{1}{N_k}}{\frac{s_k}{\sqrt{d}} + 2(M+\rho)^{-1}}\right)^2 N_k - \frac{M+2\rho}{(M+\rho)^2}\left(\frac{1}{\frac{s_k}{\sqrt{d}} + 2(M+\rho)^{-1}} - \frac{1}{\frac{s_y}{\sqrt{d}} + 2(M+\rho)^{-1}}\right)^2}}$$

which results at the limit of $d \to \infty$ in

$$\frac{s_y + s_k}{2\sqrt{N_y^{-1} + N_k^{-1}}}.$$

**Corollary D.4.** *We have*

$$\widetilde{\mathrm{Err}}_{\mathrm{wc}}^{\mathrm{ma}}(\delta^\star, \rho) = \begin{cases} \max_{k \neq y} Q\left(\frac{1}{\sqrt{\left(s_y^2 N_y\right)^{-1} + \left(s_k^2 N_k\right)^{-1}}}\right) & \rho = \infty \\ \max_{k \neq y} Q\left(\frac{s_y + s_k}{2\sqrt{N_y^{-1} + N_k^{-1}}}\right) & \rho < \infty. \end{cases} \tag{71}$$

As we see in eq. (71) we get a discontinuity in $\rho$ at the limit of $d \to \infty$ we empirically investigate this in Appendix G.3 by testing the effect of $\rho$ when $d$ is finite.

## D.2 Logit adjustment

As we explain in Appendix B running gradient descent with logit adjustment is equivalent to running gradient descent with cross-entropy and post-hoc update its biases. From Appendix D.1 it holds that the approximated maximum margin $\hat{\Sigma}^{(y)}$ is:

$$\hat{\Sigma}_{[k,k']} = \frac{1}{\xi_y} + \frac{1}{\xi_k} \mathbb{1}_{\{k=k'\}} - \frac{M + 2\rho}{(M+\rho)^2}\left(\frac{1}{\xi_k} - \frac{1}{\xi_y}\right)\left(\frac{1}{\xi_{k'}} - \frac{1}{\xi_y}\right),$$

On the other hand, when adjusting the biases by a factor of $-\iota$ we get:

$$\hat{\nu}_{[k]}^{(y)} = \frac{1}{\sigma}\left(\left(\tilde{\alpha}_{y[y]}^{\mathrm{mm}} - \tilde{\alpha}_{i[y]}^{\mathrm{mm}}\right) N_y \|\mu_y\|^2 + \tilde{b}_y^{\mathrm{mm}} - \tilde{b}_i^{\mathrm{mm}} - \iota_{[y]} + \iota_{[k]}\right)$$

$$= s_y N_y \left(\frac{1}{N_y \xi_y} + \frac{\left(\frac{1}{\xi_k} - \frac{1}{\xi_y}\right)}{N_y \xi_y (M+\rho)}\right) - \sqrt{d}\frac{\left(\frac{1}{\xi_k} - \frac{1}{\xi_y}\right) + \left(\iota_{[y]} - \iota_{[k]}\right)(M+\rho)}{(M+\rho)}.$$

**Corollary D.5.** *Let $\tilde{W}^{\mathrm{mm}}, \tilde{b}^{\mathrm{mm}}$ be the maximum margin expected kernel approximation and let $\iota_1, \dots, \iota_c$ be additive factors for the predictor's biases:*

$$\tilde{b}_{[i]}^{\mathrm{la}} = \tilde{b}_{[i]}^{\mathrm{mm}} - \iota_i.$$

*Then for every $y \neq k \in [c]$,*

$$\lim_{d \to \infty} \mathrm{Err}_{y \to k}(\tilde{W}^{\mathrm{mm}}, \tilde{b}^{\mathrm{la}}) = Q\left(\frac{s_y N_y + \tilde{\phi}_k^{\mathrm{la}}(\rho)}{\sqrt{N_y + N_k + \tilde{\psi}_{k,k}^{\mathrm{mm}}(\rho)}}\right)$$

*where,*

$$\phi^{\mathrm{la}}(\rho) = \left(\frac{\frac{s_y}{\xi_y} - \sqrt{d}}{M + \rho}\right)\left(\frac{1}{\xi_k} - \frac{1}{\xi_y}\right) - \sqrt{d}\left(\iota_{[y]} - \iota_{[k]}\right).$$

$$\psi_{k,k'}^{\mathrm{mm}}(\rho) = -\frac{M + 2\rho}{(M+\rho)^2}\left(\frac{1}{\xi_k} - \frac{1}{\xi_y}\right)\left(\frac{1}{\xi_{k'}} - \frac{1}{\xi_y}\right).$$

Recall that the lower bound for the worst class error of the logit adjustment predictor is defined by

$$\widetilde{\mathrm{Err}}_{\mathrm{wc}}^{\mathrm{la}}(\iota, \rho) := \max_{k \neq y} \lim_{d \to \infty} \mathrm{Err}_{y \to k}\left(\tilde{W}^{\mathrm{mm}}(\rho), \tilde{b}^{\mathrm{mm}}(\rho) - \iota\right).$$

**Near optimal hyperparameters.**

**Theorem D.6.** *For any dataset $\mathcal{D} \sim \mathcal{P}\left(\{s_i, N_i\}_{i=1}^c\right)$ such that the signals are within equals strength $\|\mu_i\|^2 = \frac{s}{\sqrt{d}}$ for some positive $s \in \mathbb{R}$, then it holds that*

$$\iota_y^* = \frac{2 + \|\mu_y\|^2\left[M^{\backslash y} + \rho\right]}{2\xi_y[M + \rho]} \in \arg\min_\iota \widetilde{\mathrm{Err}}_{\mathrm{wc}}^{\mathrm{la}}(\iota, \rho)$$

*Proof.* Similarly to analysis of MA, since the order of limits of $\rho \to \infty$ and $d \to \infty$ matters we investigate the lower bound of the worst class error at a finite dimension. Recall that:

$$\mathrm{Err}_{y \to k}\left(\tilde{W}_{\mathrm{MM}}, \tilde{b}_{\mathrm{LA}}\right) \stackrel{*}{=} Q\left(\frac{\frac{s}{\xi_y}\left[\frac{1}{\xi_k} + M^{\backslash y} + \rho\right] + \sqrt{d}\left[\frac{1}{\xi_y} - \frac{1}{\xi_k}\right] + \sqrt{d}[\iota_k - \iota_y][M + \rho]}{\sqrt{\frac{\left(\frac{1}{\xi_k} + [M^{\backslash y} + \rho]\right)^2}{\xi_y} + \frac{\left(\frac{1}{\xi_y} + [M^{\backslash k} + \rho]\right)^2}{\xi_k} + \left(\frac{1}{\xi_k} - \frac{1}{\xi_y}\right)^2 M^{\backslash y,k}}}\right)$$

(\*) in case that the signals are within equal strengths.

$$\iota^* \in \arg\min_\iota \max_{k \neq y} Q\left(\frac{\frac{s}{\xi_y}\left[\frac{1}{\xi_k} + M^{\backslash y} + \rho\right] + \sqrt{d}\left[\frac{1}{\xi_y} - \frac{1}{\xi_k}\right] + \sqrt{d}[\iota_k - \iota_y][M + \rho]}{\sqrt{\frac{\left(\frac{1}{\xi_k} + [M^{\backslash y} + \rho]\right)^2}{\xi_y} + \frac{\left(\frac{1}{\xi_y} + [M^{\backslash k} + \rho]\right)^2}{\xi_k} + \left(\frac{1}{\xi_k} - \frac{1}{\xi_y}\right)^2 M^{\backslash y,k}}}\right). \tag{72}$$

By the monotonicity of the Q-function it holds that finding an $\iota$ that minimizes eq. (72) is equivalent to find $\iota$ that maximizes:

$$\iota^* \in \arg\max_\iota \min_{y, k \neq y} Q\left(\frac{\frac{s}{\xi_y}\left[\frac{1}{\xi_k} + M^{\backslash y} + \rho\right] + \sqrt{d}\left[\frac{1}{\xi_y} - \frac{1}{\xi_k}\right] + \sqrt{d}[\iota_k - \iota_y][M + \rho]}{\sqrt{\frac{\left(\frac{1}{\xi_k} + [M^{\backslash y} + \rho]\right)^2}{\xi_y} + \frac{\left(\frac{1}{\xi_y} + [M^{\backslash k} + \rho]\right)^2}{\xi_k} + \left(\frac{1}{\xi_k} - \frac{1}{\xi_y}\right)^2 M^{\backslash y,k}}}\right) \tag{73}$$

Similarly to MA analysis we show that the iota that maximizes the minimal term in eq. (73) is the one that makes each $\iota_y$, $\iota_k$ dependent terms equals.

$$\frac{s}{\xi_y}\left[\frac{1}{\xi_k} + M^{\backslash y} + \rho\right] + \sqrt{d}\left[\frac{1}{\xi_y} - \frac{1}{\xi_k}\right] + \sqrt{d}[\iota_k - \iota_y][M + \rho] =$$
$$\frac{s}{\xi_k}\left[\frac{1}{\xi_y} + M^{\backslash k} + \rho\right] + \sqrt{d}\left[\frac{1}{\xi_k} - \frac{1}{\xi_y}\right] + \sqrt{d}[\iota_y - \iota_k][M + \rho] \tag{74}$$

*Remark* D.7. By the symmetry of the expression inside the Q-function of eq. (73) it is sufficient to compare the numerators of it.

First notice that:

$$\iota_y = \frac{2 + \|\mu_y\|^2\left[M^{\backslash y} + \rho\right]}{2\xi_y[M + \rho]}$$

satisfy eq. (74). In addition we define the functions that control the error score of the class $y \to k$ and $k \to y$ as $f$ and $g$ as follows:

$$f(z_{yk}) = \frac{s}{\xi_y}\left[\frac{1}{\xi_k} + M^{\backslash y} + \rho\right] + \sqrt{d}\left[\frac{1}{\xi_y} - \frac{1}{\xi_k}\right] + \sqrt{d}z_{yk}[M + \rho]$$

and

$$g(z_{yk}) = \frac{s}{\xi_k}\left[\frac{1}{\xi_y} + M^{\backslash k} + \rho\right] + \sqrt{d}\left[\frac{1}{\xi_k} - \frac{1}{\xi_y}\right] - \sqrt{d}z_{yk}[M + \rho]$$

where $z_{yk} = \iota_k - \iota_y$. Then it is clear that $f$ and $g$ are monotonically increasing and decreasing in $z_{yk}$. Therefore the minimal element is maximized in the intersection of $f$ and $g$. Which is obtained at

$$\iota_y^* = \frac{2 + \|\mu_y\|^2 \left[M^{\setminus y} + \rho\right]}{2\xi_y[M + \rho]}$$

$\square$

**Discussion.** Our analysis for near-optimal $\iota$ for the case of equal strength signals show that adjusting the logits of the maximum margin predictor can help mitigating its catastrophically failure on minorities. Specifically by removing the negative effect of $-\sqrt{d}\left[\frac{1}{\xi_k} - \frac{1}{\xi_y}\right]$.

**Corollary D.8.** *In case the signals are withing equal strengths it holds that*

$$\widetilde{\mathrm{Err}}_{\mathrm{wc}}^{\mathrm{la}}(\iota^*, \rho) = \max_{k \neq y} Q\left(\frac{s_y N_y T_{(k\setminus y)} + s_k N_k T_{(y\setminus k)} - |s_y - s_k| N_y N_k}{2\sqrt{(N_k - N_y)^2 N^{\setminus y,k} + N_y T_{(k\setminus y)}^2 + N_k T_{(y\setminus k)}^2}}\right) \tag{75}$$

*where $N^{\setminus y,k} = \sum_{i \neq y,k}^c N_i$ and $T_{(i\setminus j)} = 2N_i + N^{\setminus i,j} + \rho$.*

## D.3 CDT

From Appendix C.3 it holds that for any pair $y, i \in [c]$:

$$\tilde{\alpha}_{y[i]}^{\mathrm{cdt}} = \frac{\delta_i}{N_i \xi_i}\left(\mathbb{1}_{\{i=y\}} - \frac{\delta_y \delta_i}{\Delta}\right) \quad \text{where} \quad \Delta = \sum_{i=1}^c \delta_i^2$$

Substituting into eq. (29), we obtain expressions for $\hat{\nu}^{(y)}$ and $\hat{\Sigma}^{(y)}$ for every $y \in [c]$:

$$\hat{\nu}_{[i]}^{(y)} = \frac{\delta_y s_y}{\Delta \xi_y}\left(\Delta + \delta_{i-y}^{(y)}\right)$$

Notice that by definition of $\tilde{\alpha}_{y[i]}^{\mathrm{cdt}}$ it holds that:

$$\tilde{\alpha}_{y[z]}^{\mathrm{cdt}} - \tilde{\alpha}_{k[z]}^{\mathrm{cdt}} = \frac{\delta_z}{N_z \xi_z \Delta}\left(\mathbb{1}_{\{z=y\}}\Delta - \mathbb{1}_{\{z=k\}}\Delta + \delta_z(\delta_k - \delta_y)\right).$$

Substituting this in $\hat{\Sigma}_{[k,k']}^{(y)}$:

$$\hat{\Sigma}_{[k,k']}^{(y)} = \sum_{z=1}^c \left(\tilde{\alpha}_{y[z]}^{\mathrm{cdt}} - \tilde{\alpha}_{k[z]}^{\mathrm{cdt}}\right)\left(\tilde{\alpha}_{y[z]}^{\mathrm{cdt}} - \tilde{\alpha}_{k'[z]}^{\mathrm{cdt}}\right)N_z^2 \xi_z$$

$$= \sum_{z=1}^c \frac{\delta_z^2}{\xi_z \Delta^2}\left(\mathbb{1}_{\{z=y\}}\Delta - \mathbb{1}_{\{z=k\}}\Delta + \delta_z(\delta_k - \delta_y)\right)\left(\mathbb{1}_{\{z=y\}}\Delta - \mathbb{1}_{\{z=k'\}}\Delta + \delta_z(\delta_{k'} - \delta_y)\right)$$

$$= \sum_{z=1}^c \frac{\delta_z^2}{\xi_z \Delta^2} A_{[k,k']}^{(y)}(z).$$

Where

$$A_{[k,k']}^{(y)}(y) = \left(\Delta + \delta_{k-y}^{(y)}\right)\left(\Delta + \delta_{k'-y}^{(y)}\right),$$

$$A_{[k,k']}^{(y)}(k) = \left(\Delta + \delta_{y-k}^{(k)}\right)\left(\mathbb{1}_{\{k=k'\}}\Delta + \delta_{y-k'}^{(k)}\right),$$

$$A_{[k,k']}^{(y)}(k') = \left(\Delta + \delta_{y-k'}^{(k')}\right)\left(\mathbb{1}_{\{k=k'\}}\Delta + \delta_{y-k}^{(k')}\right),$$

and for all $i \neq k, k', y$

$$A^{(y)}_{[k,k']}(i) = \delta^{(i)}_{k-y}\delta^{(i)}_{k'-y}.$$

Which means

$$\hat{\Sigma}^{(y)}_{[k,k']} = \frac{\delta_y^2}{\Delta^2 \xi_y}\left(\Delta + \delta^{(y)}_{k-y}\right)\left(\Delta + \delta^{(y)}_{k'-y}\right) + \frac{\delta_k^2}{\Delta^2 \xi_k}\left(\Delta + \delta^{(k)}_{y-k}\right)\left(\mathbb{1}_{\{k=k'\}}\Delta + \delta^{(k)}_{y-k'}\right)$$
$$+ \mathbb{1}_{\{k \neq k'\}}\left(\frac{\delta_{k'}^2 \delta^{(k')}_{y-k}}{\Delta^2 \xi_{k'}}\right)\left(\Delta + \delta^{(k')}_{y-k'}\right) + \frac{\zeta^{(y)}_{k,k'}}{\Delta^2}$$

where

$$\delta^{(k)}_{i-y} = \delta_k(\delta_i - \delta_y) \quad \text{and} \quad \zeta^{(y)}_{i,j} = \sum_{k \neq y,i,j}\frac{\delta_k^2 \delta^{(k)}_{i-y}\delta^{(k)}_{j-y}}{\xi_k}$$

**Corollary D.9.** *The CDT expected kernel approximation predictor satisfies, for every $y \neq k \in [c]$,*

$$\lim_{d \to \infty}\text{Err}_{y \to k}(\tilde{W}^{\text{cdt}}) = Q\left(\frac{s_y\sqrt{N_y}}{\sqrt{1 + \left(\frac{\tilde{\delta}_k}{\tilde{\delta}_y}\right)^2\left(\frac{\tilde{\Delta}+\tilde{\delta}^{(k)}_{y-k}}{\tilde{\Delta}+\tilde{\delta}^{(y)}_{k-y}}\right)^2\frac{N_k}{N_y} + \left(\frac{1}{\tilde{\Delta}+\tilde{\delta}^{(y)}_{k-y}}\right)^2\frac{\tilde{\zeta}^{(y)}_{k,k}}{\tilde{\delta}_y^2 N_y}}}\right),$$

*where*

$$\Delta = \sum_{i \in [c]}\delta_i^2 \quad, \quad \delta^{(k)}_{i-y} = \delta_k(\delta_i - \delta_y) \quad \text{and} \quad \zeta^{(y)}_{i,j} = \sum_{k \neq y,i,j}\frac{\delta_k^2 \delta^{(k)}_{i-y}\delta^{(k)}_{j-y}}{\xi_k},$$

*and tilde denotes the limit at infinity, e.g.,*

$$\tilde{\Delta} := \lim_{d \to \infty}\Delta.$$

# E   Analytical approximation summary

In this section we summarize the parameterized that define the error of each predictor we analyze in this work. Denote by $\xi_i := \|\mu_i\|^2 + \frac{\sigma^2 d}{N_i}$, $M := \sum_{i=1}^c \frac{1}{\xi_i}$ and $M^{\backslash y} := \sum_{i \neq y}^c \frac{1}{\xi_i}$.

**Max margin.** The expected kernel approximation gives

$$\tilde{\alpha}^{\text{mm}}_{y[i]} = \frac{1}{N_i\xi_i}\left(\mathbb{1}_{\{i=y\}} - \frac{1}{c}\right) + \frac{\sum_{j=1}^c\left(\frac{1}{\xi_j} - \frac{1}{\xi_y}\right)}{cN_i\xi_i(M+\rho)} \quad \text{and} \quad \tilde{b}^{\text{mm}}_{[y]} = \frac{\sum_{i=1}^c\left(\frac{1}{\xi_y} - \frac{1}{\xi_i}\right)}{c(M+\rho)}$$

and the expected score statistics approximation gives

$$\hat{\nu}^{(y)}_{[k]} = s_yN_y\left(\frac{1}{N_y\xi_y} + \frac{\left(\frac{1}{\xi_k} - \frac{1}{\xi_y}\right)}{N_y\xi_y(M+\rho)}\right) - \sqrt{d}\frac{\left(\frac{1}{\xi_k} - \frac{1}{\xi_y}\right)}{(M+\rho)} \quad \text{and}$$

$$\hat{\Sigma}^{(y)}_{[k,k']} = \frac{1}{\xi_y} + \frac{1}{\xi_k}\mathbb{1}_{\{k=k'\}} - \frac{M+2\rho}{(M+\rho)^2}\left(\frac{1}{\xi_k} - \frac{1}{\xi_y}\right)\left(\frac{1}{\xi_{k'}} - \frac{1}{\xi_y}\right).$$

**Margin adjustment.** The expected kernel approximation gives

$$\tilde{\alpha}^{\mathrm{ma}}_{y[i]} = \frac{\delta_i}{N_i \xi_i}\left(\mathbb{1}_{\{i=y\}} - \frac{1}{c}\right) + \frac{\sum_{j=1}^{c}\left(\frac{\delta_j}{\xi_j} - \frac{\delta_y}{\xi_y}\right)}{cN_i \xi_i (M+\rho)} \quad\text{and}\quad \tilde{b}^{\mathrm{ma}}_{[y]} = \frac{\sum_{i=1}^{c}\left(\frac{\delta_y}{\xi_y} - \frac{\delta_i}{\xi_i}\right)}{c(M+\rho)}$$

and the expected score statistics approximation gives

$$\hat{\nu}^{(y)}_{[k]} = s_y N_y \left(\frac{\delta_y}{N_y \xi_y} + \frac{\frac{\delta_k}{\xi_k} - \frac{\delta_y}{\xi_y}}{N_y \xi_y (M+\rho)}\right) - \sqrt{d}\left(\frac{\frac{\delta_k}{\xi_k} - \frac{\delta_y}{\xi_y}}{M+\rho}\right) \quad\text{and}$$

$$\hat{\Sigma}^{(y)}_{[k,k']} = \frac{\delta_y^2}{\xi_y} + \frac{\delta_k^2}{\xi_k}\mathbb{1}_{\{k=k'\}} - \frac{M+2\rho}{(M+\rho)^2}\left(\frac{\delta_k}{\xi_k} - \frac{\delta_y}{\xi_y}\right)\left(\frac{\delta_{k'}}{\xi_{k'}} - \frac{\delta_y}{\xi_y}\right).$$

Additionally, the expected score statistics at the near-optimal margin, eq. (18), gives

$$\hat{\nu}^{(y)}_{[k]} = s_y \left(\frac{\frac{\xi_y}{s_y q+1}(M+\rho) + q\left(\frac{s_k - s_y}{(s_k q+1)(s_y q+1)}\right)}{\xi_y (M+\rho)}\right) + \frac{1}{2}\left(\frac{s_k - s_y}{(s_k q+1)(s_y q+1)}\right) \quad\text{and}$$

$$\hat{\Sigma}^{(y)}_{[k,k']} = \frac{\xi_y}{(s_y q+1)^2} + \frac{\xi_k}{(s_k q+1)^2}\mathbb{1}_{\{k=k'\}} - \frac{M+2\rho}{4d}\left(\frac{s_k - s_y}{(s_k q+1)(s_y q+1)}\right)\left(\frac{s_{k'} - s_y}{(s_{k'} q+1)(s_y q+1)}\right),$$

$$\text{where}\qquad q := (M+\rho)\left(2\sqrt{d}\right)^{-1} \quad\text{and}\quad \delta_i^{\star} = \frac{\xi_i}{\|\mu_i\|^2(M+\rho)2^{-1}+1}.$$

Notice that we scale $\delta^{\star}$ in eq. (18) by $\frac{1}{(M+\rho)2^{-1}}$ (as scaling all the parameters by a scalar does not change the solution).

**Logit adjustment.** The expected kernel approximation is the same as maximum margin, except we offset the bias by iota. This results in obtaining the same parameters as the maximum margin with bias adjustment of $\iota$:

$$\tilde{\alpha}^{\mathrm{la}}_{y[i]} = \tilde{\alpha}^{\mathrm{mm}}_{y[i]} \quad\text{and}\quad \tilde{b}^{\mathrm{la}}_{[y]} = \tilde{b}^{\mathrm{mm}}_{[y]} - \iota_{[y]}.$$

The expected score statistics are

$$\hat{\nu}^{(y)}_{[k]} = s_y N_y \left(\frac{1}{N_y \xi_y} + \frac{\left(\frac{1}{\xi_k} - \frac{1}{\xi_y}\right)}{N_y \xi_y (M+\rho)}\right) - \sqrt{d}\frac{\left(\frac{1}{\xi_k} - \frac{1}{\xi_y}\right) + \left(\iota_{[y]} - \iota_{[k]}\right)(M+\rho)}{(M+\rho)} \quad\text{and}$$

$$\hat{\Sigma}^{(y)}_{[k,k']} = \frac{1}{\xi_y} + \frac{1}{\xi_k}\mathbb{1}_{\{k=k'\}} - \frac{M+2\rho}{(M+\rho)^2}\left(\frac{1}{\xi_k} - \frac{1}{\xi_y}\right)\left(\frac{1}{\xi_{k'}} - \frac{1}{\xi_y}\right).$$

**Class dependent temperature.** The expected kernel approximation gives

$$\tilde{\alpha}^{\mathrm{cdt}}_{y[i]} = \frac{\delta_i}{N_i \xi_i}\left(\mathbb{1}_{\{i=y\}} - \frac{\delta_y \delta_i}{\Delta}\right) \quad\text{where}\quad \Delta = \sum_{i=1}^{c}\delta_i^2,$$

The expected score statistics are

$$\hat{\nu}^{(y)}_{[k]} = \frac{\delta_y s_y}{\Delta \xi_y}\left(\Delta + \delta^{(y)}_{k-y}\right) \quad\text{where}\quad \delta^{(y)}_{k-y} = \delta_y(\delta_k - \delta_y)$$

and

$$\hat{\Sigma}^{(y)}_{[k,k']} = \frac{\delta_y^2}{\Delta^2 \xi_y}\left(\Delta + \delta^{(y)}_{k-y}\right)\left(\Delta + \delta^{(y)}_{k'-y}\right) + \frac{\delta_k^2}{\Delta^2 \xi_k}\left(\Delta + \delta^{(k)}_{y-k}\right)\left(\mathbb{1}_{\{k=k'\}}\Delta + \delta^{(k)}_{y-k'}\right)$$

$$+ \mathbb{1}_{\{k \neq k'\}}\left(\frac{\delta_{k'}^2 \delta^{(k')}_{y-k}}{\Delta^2 \xi_{k'}}\right)\left(\Delta + \delta^{(k')}_{y-k'}\right) + \frac{\zeta^{(y)}_{k,k'}}{\Delta^2} \quad\text{where}\quad \zeta^{(y)}_{k,k'} = \sum_{z \neq y,i,j}\frac{\delta_z^2}{\xi_z}\delta^{(z)}_{k-y}\delta^{(z)}_{k'-y}$$

# F    Methods comparison

In this section we present missing proofs from Section 4.

## F.1    Margin adjustment

**Proposition 4.2.** *Let $N_1 \leq N_2 \leq \cdots \leq N_c$ and let $\rho < \infty$. If $s_1 \leq s_2 \leq \cdots \leq s_c$ then $\widetilde{\mathrm{Err}}_{\mathrm{wc}}^{\mathrm{ma}}(\delta^\star, \rho) \leq \widetilde{\mathrm{Err}}_{\mathrm{wc}}^{\mathrm{ma}}(\delta^\star, \infty)$, with equality when $s_1 = s_2 = \cdots = s_c$. When $s_1 > s_2 > \cdots > s_c$ then either $\widetilde{\mathrm{Err}}_{\mathrm{wc}}^{\mathrm{ma}}(\delta^\star, \rho) > \widetilde{\mathrm{Err}}_{\mathrm{wc}}^{\mathrm{ma}}(\delta^\star, \infty)$ or $\widetilde{\mathrm{Err}}_{\mathrm{wc}}^{\mathrm{ma}}(\delta^\star, \rho) < \widetilde{\mathrm{Err}}_{\mathrm{wc}}^{\mathrm{ma}}(\delta^\star, \infty)$ is possible.*

*Proof.* We prove Proposition 4.2 by showing that the relevant equality and inequality hold for any pair $k \neq y$ which implies that they hold for the worst class error as well.

Recall $\widetilde{\mathrm{Err}}_{\mathrm{wc}}^{\mathrm{ma}}(\delta^\star, \infty)$ and $\widetilde{\mathrm{Err}}_{\mathrm{wc}}^{\mathrm{ma}}(\delta^\star, \rho)$ for $\rho < \infty$:

$$\widetilde{\mathrm{Err}}_{\mathrm{wc}}^{\mathrm{ma}}(\delta^\star, \infty) = \max_{k \neq y} Q\left( \frac{1}{\sqrt{\frac{4}{(s_y + s_k)^2}\left(N_y^{-1} + N_k^{-1}\right)}} \right) \quad \text{and} \quad \widetilde{\mathrm{Err}}_{\mathrm{wc}}^{\mathrm{ma}}(\delta^\star, \rho) = \max_{k \neq y} Q\left( \frac{1}{\sqrt{\frac{1}{N_y s_y^2} + \frac{1}{N_k s_k^2}}} \right)$$

The expressions are clearly equal when $s_y = s_k$, so we focus on the other two cases, assuming without loss of generality that $N_y < N_k$. Note that

$$Q\left( \frac{1}{\sqrt{\frac{4}{(s_y + s_k)^2}\left(N_y^{-1} + N_k^{-1}\right)}} \right) < Q\left( \frac{1}{\sqrt{\frac{1}{N_y s_y^2} + \frac{1}{N_k s_k^2}}} \right)$$

if and only if

$$\frac{4}{(s_y + s_k)^2}\left(N_y^{-1} + N_k^{-1}\right) < \left(s_y^2 N_y\right)^{-1} + \left(s_k^2 N_k\right)^{-1}.$$

Writing $q = \frac{N_y}{N_k} < 1$ and $r = \frac{s_k}{s_y}$, the inequality above is equivalent to

$$f(r) := (1 + r)^2 + \left(\frac{1}{r} + 1\right)^2 q - 4(1 + q) > 0.$$

Note that

$$f'(r) = 2(1 + r)\left(1 - \frac{q}{r^3}\right),$$

and therefore, we have $f'(r) > 0$ for $r > 1$. Since $f(1) = 0$, this implies that $f(r) > 0$ for all $r = \frac{s_k}{s_y} > 1$ and $q \leq 1$. Therefore, when signal strengths are aligned with class sizes (i.e., $s_y < s_k$) we have

$$Q\left( \frac{1}{\sqrt{\frac{4}{(s_y + s_k)^2}\left(N_y^{-1} + N_k^{-1}\right)}} \right) < Q\left( \frac{1}{\sqrt{\frac{1}{N_y s_y^2} + \frac{1}{N_k s_k^2}}} \right) \quad \text{and hence lower error for } \rho < \infty \text{ as claimed.}$$

When the signals are not aligned and so $r < 1$, the sign of $f'(r)$ depends on the value of $q$, and hence both relative orders are possible, concluding the proof. $\qquad\square$

## F.2    Logit adjustment

**Proposition 4.4.** *If $s_1 = s_2 = \cdots = s_c$ then $\widetilde{\mathrm{Err}}_{\mathrm{wc}}^{\mathrm{la}}(\iota^*, \rho)$ is monotonic decreasing in $\rho$ and, for all values of $\rho$, we have $\widetilde{\mathrm{Err}}_{\mathrm{wc}}^{\mathrm{la}}(\iota^*, \rho) \leq \widetilde{\mathrm{Err}}_{\mathrm{wc}}^{\mathrm{ma}}(\delta^\star, \rho)$.*

*Proof.* First notice that by the assumption in the lemma it holds that $\|\mu_i\|^2 = \frac{s}{\sqrt{d}}$ for all $i \in [c]$ and for some $s > 0$.

Recall the expressions for $\widetilde{\mathrm{Err}}_{\mathrm{wc}}^{\mathrm{la}}(\iota^*, \rho)$ eq. (75) and $\widetilde{\mathrm{Err}}_{\mathrm{wc}}^{\mathrm{ma}}(\delta^*, \rho)$ eq. (71):

$$\widetilde{\mathrm{Err}}_{\mathrm{wc}}^{\mathrm{la}}(\iota^*, \rho) = \max_{k \neq y} Q\left(\frac{sN_yT_{(k\backslash y)} + sN_kT_{(y\backslash k)}}{2\sqrt{(N_k - N_y)^2 N^{\backslash y,k} + N_yT^2_{(k\backslash y)} + N_kT^2_{(y\backslash k)}}}\right)$$

and

$$\widetilde{\mathrm{Err}}_{\mathrm{wc}}^{\mathrm{ma}}(\delta^*, \rho) = \max_{y,k \neq y} Q\left(\frac{s}{\sqrt{N_y^{-1} + N_k^{-1}}}\right).$$

where $N^{\backslash y,k} = \sum_{i \neq y,k}^c N_i$ and $T_{(i\backslash j)} = 2N_i + N^{\backslash i,j} + \rho$.

We show that the inequality $Q\left(\frac{sN_yT_{(k\backslash y)} + sN_kT_{(y\backslash k)}}{2\sqrt{(N_k - N_y)^2 N^{\backslash y,k} + N_yT^2_{(k\backslash y)} + N_kT^2_{(y\backslash k)}}}\right) \leq Q\left(\frac{s}{\sqrt{N_y^{-1} + N_k^{-1}}}\right)$ holds for any pair $k, y$ and therefore holds for the maximum over pairs. By the monotonicity of the $Q$ function it is enough to show that for any pair $y \neq k$,

$$\frac{N_y(2N_k + N^{\backslash yk} + \rho) + N_k(2N_y + N^{\backslash yk} + \rho)}{\sqrt{(N_k - N_y)^2 N^{\backslash yk} + (2N_k + N^{\backslash yk} + \rho)^2 N_y + (2N_y + N^{\backslash yk} + \rho)^2 N_k}} \geq \frac{2}{\sqrt{N_y^{-1} + N_k^{-1}}}. \tag{76}$$

We start by showing that the LHS of eq. (76) is minimized when $\rho = 0$. Let $f_1(\rho) = \frac{g_1(\rho)}{h_1(\rho)}$ where

$$g_1(\rho) := N_y\left(2N_k + N^{\backslash yk} + \rho\right) + N_k\left(2N_y + N^{\backslash yk} + \rho\right)$$

and

$$h_1(\rho) := \sqrt{(N_k - N_y)^2 N^{\backslash yk} + (2N_k + N^{\backslash yk} + \rho)^2 N_y + (2N_y + N^{\backslash yk} + \rho)^2 N_k}.$$

Then,

$$g_1'(\rho) = N_y + N_k$$

and

$$h_1'(\rho) = \frac{1}{2}\left((N_k - N_y)^2 N^{\backslash yk} + (2N_k + N^{\backslash yk} + \rho)^2 N_y + (2N_y + N^{\backslash yk} + \rho)^2 N_k\right)^{-1/2} \cdot$$
$$\left(8N_yN_k + 2N^{\backslash yk}(N_y + N_k) + 2\rho(N_y + N_k)\right).$$

Therefore,

$$f_1'(\rho) = \frac{(N_y + N_k)\left((N_k - N_y)^2 N^{\backslash yk} + (2N_k + N^{\backslash yk} + \rho)^2 N_y + (2N_y + N^{\backslash yk} + \rho)^2 N_k\right)}{\left((N_k - N_y)^2 N^{\backslash yk} + (2N_k + N^{\backslash yk} + \rho)^2 N_y + (2N_y + N^{\backslash yk} + \rho)^2 N_k\right)^{3/2}}$$
$$- \frac{\left(4N_yN_k + N^{\backslash yk}(N_y + N_k) + \rho(N_y + N_k)\right)\left(4N_yN_k + N^{\backslash yk}(N_y + N_k) + \rho(N_y + N_k)\right)}{\left((N_k - N_y)^2 N^{\backslash yk} + (2N_k + N^{\backslash yk} + \rho)^2 N_y + (2N_y + N^{\backslash yk} + \rho)^2 N_k\right)^{3/2}}$$
$$= \frac{(N_y + N_k)(N_k - N_y)^2 N^{\backslash yk} + N_yN_k\left((2N_k + N^{\backslash yk} + \rho) + (2N_y + N^{\backslash yk} + \rho)\right)^2}{\left((N_k - N_y)^2 N^{\backslash yk} + (2N_k + N^{\backslash yk} + \rho)^2 N_y + (2N_y + N^{\backslash yk} + \rho)^2 N_k\right)^{3/2}} > 0,$$

which means that $f_1$ is monotonicity increasing in $\rho$ (and the composition of the $Q$-function on it is monotonicity decreasing) and

$$f_1(\rho) \geq f_1(0) = \frac{N_y\big(2N_k + N^{\backslash yk}\big) + N_k\big(2N_y + N^{\backslash yk}\big)}{\sqrt{(N_k - N_y)^2 N^{\backslash yk} + \big(2N_k + N^{\backslash yk}\big)^2 N_y + \big(2N_y + N^{\backslash yk}\big)^2 N_k}}.$$

Next we define $f_2$ as a function of $N^{\backslash yk}$ as follows:

$$f_2\big(N^{\backslash yk}\big) = \frac{N_y\big(2N_k + N^{\backslash yk}\big) + N_k\big(2N_y + N^{\backslash yk}\big)}{\sqrt{(N_k - N_y)^2 N^{\backslash yk} + \big(2N_k + N^{\backslash yk}\big)^2 N_y + \big(2N_y + N^{\backslash yk}\big)^2 N_k}}$$

Similarly to the former step, we look at the derivative of $f_2(N^{\backslash yk})$:

$$
\begin{aligned}
f_2'(N^{\backslash yk}) &= \frac{(N_y + N_k)\Big((N_k - N_y)^2 N^{\backslash yk} + \big(2N_k + N^{\backslash yk}\big)^2 N_y + \big(2N_y + N^{\backslash yk}\big)^2 N_k\Big)}{\Big((N_k - N_y)^2 N^{\backslash yk} + \big(2N_k + N^{\backslash yk}\big)^2 N_y + \big(2N_y + N^{\backslash yk}\big)^2 N_k\Big)^{3/2}} \\
&\quad - \frac{\Big(\frac{(N_k - N_y)^2}{2} + N_y\big(2N_k + N^{\backslash yk}\big) + N_k\big(2N_y + N^{\backslash yk}\big)\Big)\Big(N_y\big(2N_k + N^{\backslash yk}\big) + N_k\big(2N_y + N^{\backslash yk}\big)\Big)}{\Big((N_k - N_y)^2 N^{\backslash yk} + \big(2N_k + N^{\backslash yk}\big)^2 N_y + \big(2N_y + N^{\backslash yk}\big)^2 N_k\Big)^{3/2}} \\
&= \frac{(N_k - N_y)^2 \frac{N^{\backslash yk}}{2}(N_y + N_k) + 4 N_y N_k \Big(\big(N_k + N_y + N^{\backslash yk}\big)^2 - \frac{1}{2}\Big)}{\Big((N_k - N_y)^2 N^{\backslash yk} + \big(2N_k + N^{\backslash yk}\big)^2 N_y + \big(2N_y + N^{\backslash yk}\big)^2 N_k\Big)^{3/2}} > 0
\end{aligned}
$$

Where the last inequality holds since $N_k + N_y \geq 2$. Therefore it holds that:

$$f_2(N^{\backslash yk}) \geq f_2(0) = \frac{2}{\sqrt{N_k^{-1} + N_y^{-1}}}$$

and we get

$$\frac{N_y\big(2N_k + N^{\backslash yk} + \rho\big) + N_k\big(2N_y + N^{\backslash yk} + \rho\big)}{\sqrt{(N_k - N_y)^2 N^{\backslash yk} + \big(2N_k + N^{\backslash yk} + \rho\big)^2 N_y + \big(2N_y + N^{\backslash yk} + \rho\big)^2 N_k}} \geq \frac{2}{\sqrt{N_y^{-1} + N_k^{-1}}}$$

as required. $\qquad\square$

### F.3 An intuition why CDT attains non-zero training error

Let us consider the optimization problem associated with the CDT predictor:

$$
\begin{aligned}
W^{\mathrm{cdt}}, b^{\mathrm{cdt}} := \;& \underset{W \in \mathbb{R}^{d \times c}, b \in \mathbb{R}^c}{\arg\min} \; \|W\|_{\mathrm{F}}^2 + \rho \|b\|^2 \\
& \text{subject to } \; \frac{1}{\delta_{y_i}}\big(w_{y_i}^\top x_i + b_{[y_i]}\big) - \frac{1}{\delta_k}\big(w_k^\top x_i + b_{[k]}\big) \geq 1 \;\text{ for all } \; i \leq N \text{ and } k \neq y_i.
\end{aligned}
\qquad (77)
$$

Based on the definition of the CDT predictor optimization, eq. (77), it holds that for any $(x_i, y_i)$ in the training set, the CDT predictor satisfies:

$$\frac{1}{\delta_{y_i}}\big(w_{y_i}^\top x_i + b_{[y_i]}\big) - \frac{1}{\delta_k}\big(w_k^\top x_i + b_{[k]}\big) \geq 1.$$

It is worth noting that this condition can be satisfied even in scenarios when the predictor attain non-zero training error. For example, assume that $\delta_{y_i} = 1$ and $\delta_k = 4$ when $w_{y_i}^\top x_i + b_{[y_i]} = 2$ and $w_k^\top x_i + b_{[k]} = 3$, as shown below:

$$\frac{1}{\delta_{y_i}}\left(w_{y_i}^\top x_i + b_{[y_i]}\right) - \frac{1}{\delta_k}\left(w_k^\top x_i + b_{[k]}\right) = 2 - \frac{3}{4} \geq 1,$$

the condition of eq. (77) holds. However, during inference, the learned predictor does not use the $\delta$ terms to compensate for incorrect predictions. This leads to the following scenario:

$$\left(w_{y_i}^\top x_i + b_{[y_i]}\right) - \left(w_k^\top x_i + b_{[k]}\right) = 2 - 3 < 0,$$

which implies that:

$$\arg\max_{j\in[c]}\left(w_j^\top x_i + b_{[j]}\right) \neq y_i.$$

Thus the training error of the predictor is greater than zero. We verify this intuition using a simple setting of synthetic Gaussian data in $\mathbb{R}^2$, see in Appendix G.7.

### F.4 The limitation of CDT in multiclass classification

**Proposition 4.6.** *For any $\epsilon > 0$ and $c \geq 3$ classes, there exists an instance $\{s_i, N_i\}_{i=1}^c$ for which*

$$\lim_{d\to\infty} \mathrm{Err}_{\mathrm{wc}}\left(\tilde{W}^{\mathrm{cdt}}, 0\right) \geq \frac{1}{2} - \epsilon \quad \text{while} \quad \lim_{d\to\infty} \mathrm{Err}_{\mathrm{wc}}\left(\tilde{W}^{\mathrm{ma}}(\delta^\star, \infty), 0\right) \leq \epsilon.$$

*Proof.* Corollary D.9 gives

$$\lim_{d\to\infty} \mathrm{Err}_{\mathrm{wc}}\left(\tilde{W}^{\mathrm{cdt}}, 0\right) \geq \max_{k\neq y} Q\left(\frac{s_y\sqrt{N_y}}{\sqrt{1 + \left(\frac{\tilde{\delta}_k}{\tilde{\delta}_y}\right)^2\left(\frac{\tilde{\Delta}+\tilde{\delta}_{y-k}^{(k)}}{\tilde{\Delta}+\tilde{\delta}_{k-y}^{(y)}}\right)^2\frac{N_k}{N_y} + \left(\frac{1}{\tilde{\Delta}+\tilde{\delta}_{k-y}^{(y)}}\right)^2\frac{\tilde{\zeta}_{k,k}^{(y)}}{\tilde{\delta}_y^2 N_y}}}\right)$$

$$\overset{(1)}{\geq} \max_{k\neq y} Q\left(\frac{s_y\sqrt{N_y}}{\sqrt{1 + \left(\frac{\tilde{\delta}_k}{\tilde{\delta}_y}\right)^2\left(\frac{\tilde{\Delta}+\tilde{\delta}_{y-k}^{(k)}}{\tilde{\Delta}+\tilde{\delta}_{k-y}^{(y)}}\right)^2\frac{N_k}{N_y}}}\right). \tag{78}$$

where (1) holds by monotonicity of the Q-function and $\frac{\tilde{\zeta}_{k,k}^{(y)}}{\tilde{\delta}_y^2 N_y} > 0$.

Let $\tilde{\Delta}_{k,y} := \sum_{i\neq k,y} \tilde{\delta}_i^2$ and denote by $\psi_{y,k}$

$$\psi_{y,k} = \left(\frac{\tilde{\delta}_y}{\tilde{\delta}_k}\right)\left(\frac{\tilde{\Delta}+\tilde{\delta}_{k-y}^{(y)}}{\tilde{\Delta}+\tilde{\delta}_{y-y}^{(k)}}\right) = \left(\frac{\tilde{\delta}_y}{\tilde{\delta}_k}\right)\left(\frac{\tilde{\Delta}_{k,y}+\tilde{\delta}_k^2+\tilde{\delta}_k\tilde{\delta}_y}{\tilde{\Delta}_{k,y}+\tilde{\delta}_y^2+\tilde{\delta}_k\tilde{\delta}_y}\right)$$

$$= \left(\frac{\tilde{\Delta}_{k,y}/\tilde{\delta}_k + \tilde{\delta}_k + \tilde{\delta}_y}{\tilde{\Delta}_{k,y}/\tilde{\delta}_y + \tilde{\delta}_y + \tilde{\delta}_k}\right) = 1 + \tilde{\Delta}_{k,y}\frac{1/\tilde{\delta}_k - 1/\tilde{\delta}_y}{\tilde{\Delta}_{k,y}/\tilde{\delta}_y + \tilde{\delta}_y + \tilde{\delta}_k} \tag{79}$$

To establish our result, we define the signal strengths $s_1 > s_2 > \cdots > s_c$ and class sizes $N_1 < N_2 < \cdots < N_c$ as follows. For every $y \in [c]$, we set

$$s_y = \frac{2Q^{-1}(\epsilon/c)}{\sqrt{N_y}}, \tag{80}$$

and we take $N_1 = 1$, and, for every $i > 1$, set

$$N_i = \left\lceil \max\left\{1, \frac{c^2}{4}\right\} N_{i-1} \left( \frac{2Q^{-1}(\epsilon/c)}{Q^{-1}(\frac{1}{2} - \epsilon)} \right)^2 + 1 \right\rceil. \tag{81}$$

We now consider two cases.

**Case 1: There exists $y < k$ such that $\tilde{\delta}_y < \tilde{\delta}_k$.** Therefore, by eq. (79) we have $\psi_{y,k} < 1$ and, by eq. (78),

$$\lim_{d \to \infty} \mathrm{Err}_{\mathrm{wc}}\left(\tilde{W}^{\mathrm{cdt}}, 0\right) \geq Q\left( \frac{s_y \sqrt{N_y}}{\sqrt{1 + \frac{N_k}{N_y}}} \right).$$

By our definitions of $N_i$ and $s_i$ in eqs. (80) and (81) above, we have Now notice that by our definition for $N_i$, eq. (81), it holds that

$$\frac{s_y \sqrt{N_y}}{\sqrt{1 + \frac{N_k}{N_y}}} < Q^{-1}\left(\frac{1}{2} - \epsilon\right),$$

giving the claimed bound.

**Case 2: For all $y < k$ it holds that $\tilde{\delta}_k < \tilde{\delta}_y$.** In this case we have

$$\psi_{1,2} = 1 + \tilde{\Delta}_{1,2} \frac{1/\tilde{\delta}_2 - 1/\tilde{\delta}_1}{\tilde{\Delta}_{1,2}/\tilde{\delta}_1 + \tilde{\delta}_1 + \tilde{\delta}_2} < 1 + \tilde{\Delta}_{1,2} \frac{1}{\tilde{\delta}_2 \tilde{\delta}_1 + \tilde{\delta}_2^2}$$

$$< 1 + \frac{[c-2]\tilde{\delta}_2^2}{\tilde{\delta}_2 \tilde{\delta}_1 + \tilde{\delta}_2^2} \leq 1 + \frac{c-2}{2} = \frac{c}{2}.$$

Which means that

$$\lim_{d \to \infty} \mathrm{Err}_{\mathrm{wc}}\left(\tilde{W}^{\mathrm{cdt}}, 0\right) \geq Q\left( \frac{s_1 \sqrt{N_1}}{\sqrt{1 + \frac{4}{c^2} \frac{N_2}{N_1}}} \right).$$

By the same explanation of case 1 it holds that

$$\frac{s_y \sqrt{N_y}}{\sqrt{1 + \frac{4}{c^2} \frac{N_k}{N_y}}} < Q^{-1}\left(\frac{1}{2} - \epsilon\right),$$

Therefore for any $\delta$:

$$\lim_{d \to \infty} \mathrm{Err}_{\mathrm{wc}}\left(\tilde{W}^{\mathrm{cdt}}, 0\right) \geq \frac{1}{2} - \epsilon$$

as claimed.

To conclude the proof, we show that the worst class error of MA is less than equal to $\epsilon$. Recall that

$$\lim_{d \to \infty} \mathrm{Err}_{\mathrm{wc}}\left(\tilde{W}^{\mathrm{ma}}(\delta^\star, \infty), 0\right) \leq c \cdot \max_{k \neq y} \widetilde{\mathrm{Err}}_{\mathrm{wc}}^{\mathrm{ma}}(\delta^\star, \infty),$$

and note that eq. (19) and our choices of $s_i$ and $N_i$ give

$$\widetilde{\mathrm{Err}}_{\mathrm{wc}}^{\mathrm{ma}}(\delta^\star, \infty) = \max_{k \neq y} Q\left( \frac{s_y \sqrt{N_y}}{1 + \frac{s_y^2 N_y}{s_k^2 N_k}} \right) = Q\left( \frac{2Q^{-1}(\epsilon/c)}{2} \right) = \frac{\epsilon}{c},$$

completing the proof. $\qquad\square$

### F.5 The limitation of CDT in binary classification

We show that when using CDT with $c = 2$, the direction of the classifiers are independent of $\delta_1$ and $\delta_2$, and therefore CDT is no better than the MM. We note that while Kini et al. (2021) claim to analyze CDT in the binary case (and show improved performance on minority classes), the method they analyze is really a binary version of MA.

**Lemma F.1.** *Let $\mathcal{D} = \{(x_1, y_1, ..., x_n, y_n)\}$ be a separable training set such that for any $i \in [n]$ $y_i \in \{1, 2\}, x_i \in \mathbb{R}^d$. Assume $w_1, w_2 \in \mathbb{R}^d$ comprise the CDT predictor with parameters $\delta_1, \delta_2 > 0$. In addition let $\hat{w}_1, \hat{w}_2$ be the maximum margin predictor over $\mathcal{D}$ then for any $i \in \{1, 2\}$,*

$$\frac{w_i}{\|w_i\|} = \frac{\hat{w}_i}{\|\hat{w}_i\|}$$

*Proof.* From the dual problem it holds that the learned predictor is of the form:

$$w_1 = \frac{1}{\delta_1} \sum_{i=1}^{n} \lambda_i y_i x_i \quad \text{and} \quad w_2 = -\frac{1}{\delta_2} \sum_{i=1}^{n} \lambda_i y_i x_i,$$

where $\lambda_i$ is the dual variable that correspond the $(x_i, y_i)$ sample. In addition, notice that since $\delta_1, \delta_2 > 0$ there exists $0 < c \in \mathbb{R}$ such that $\delta_1 = c \cdot \delta_2$, which means that:

$$w_1 = \frac{1}{c\delta_2} \sum_{i=1}^{n} \lambda_i y_i x_i = \frac{1}{c\delta_2} w' \quad \text{and} \quad w_2 = -\frac{1}{\delta_2} \sum_{i=1}^{n} \lambda_i y_i x_i = -\frac{1}{\delta_2} w'. \tag{82}$$

Recall that the CDT predictor solve the following optimization problem:

$$\underset{w_1, w_2}{\text{minimize}} \|w_1\|^2 + \|w_2\|^2 : \forall i \in [N] \quad y_i \left( \frac{w_1}{\delta_1} - \frac{w_2}{\delta_2} \right)^\top x_i \geq 1 \tag{83}$$

plug in eq. (82) in eq. (83):

$$\underset{w'}{\text{minimize}} \frac{1}{c^2 \delta_2^2} \|w'\|^2 + \frac{1}{\delta_2^2} \|w'\|^2 : \forall i \in [N] \quad y_i \left( \frac{1}{c^2 \delta_2^2} w' + \frac{1}{\delta_2^2} w' \right)^\top x_i \geq 1$$

which is equivalent to:

$$\underset{w'}{\text{minimize}} \|w'\|^2 : \forall i \in [N] \quad y_i \left( \frac{1}{c^2 \delta_2^2} + \frac{1}{\delta_2^2} \right) w'^\top x_i \geq 1. \tag{84}$$

Recall that the maximum margin predictor satisfies $\hat{w}_1 = -\hat{w}_2$ (which essentially results in learning a single predictor $\hat{w} = 2\hat{w}_1$). Hence eq. (84) implies that

$$2\hat{w}_1 = \left( \frac{1}{c^2 \delta_2^2} + \frac{1}{\delta_2^2} \right) w' \implies 2 \frac{c^2 \delta_2}{1 + c^2} \hat{w}_1 = w'$$

we can plug this back in $w_1$ and $w_2$ and get that the CDT predictor is just a scaling of the maximum margin predictor:

$$w_1 = \frac{2c}{1 + c^2} \hat{w}_1 \quad \text{and} \quad w_2 = -\frac{2c^2}{1 + c^2} \hat{w}_1 = \frac{2c^2}{1 + c^2} \hat{w}_2$$

$\square$

See empirical validation in Appendix G.7.

## G    Experiments

In this section, we provide a detailed presentation of our experiments. We showcase experiments conduct on multiple seeds and settings, including additional experiments omitted from the main paper due to space constraints.

**How we compute predictors.**  Unless stated otherwise, we find the MM, MA, CDT and LA predictors we consider by solving the corresponding margin maximization problems (defined in Section 2.2) using CVXPY (Diamond & Boyd, 2016) with the MOSEK solver (MOSEK ApS, 2023).

**How we compute error approximation.**  To compute the error approximation, we use the expressions for $\tilde{\alpha}$ and $\tilde{b}$ (refer to details in Appendix C) to construct the statistics score parameters $\hat{\nu}^{(y)}$ and $\hat{\Sigma}^{(y)}$ as described in Appendix D. Then, for calculating the error of the $y$'th class, we employ Monte Carlo simulation. Specifically, we sample 10,000 random samples from $\mathcal{N}\left(\hat{\nu}^{(y)}, \hat{\Sigma}^{(y)}\right)$ using numpy.random.multivariate_normal and record the number of positive vectors as $N_+$. The error approximation of the $y$'th class is subsequently set to $1 - \pi_+$, where $\pi_+ = \frac{N_+}{10,000}$.

**Gaussian mixture model datasets.**  To generate a Gaussian mixture dataset that aligns with our data assumptions, we follow the steps outlined below. For a dataset with $c$ classes, we define the first $c$ standard basis vectors as the directions of $\mu_1, \ldots, \mu_c$. Specifically, each $\mu_i$ is

$$\mu_i = e_i \cdot \sqrt{\frac{s_i}{\sqrt{d}}} \quad \text{such that} \quad \|\mu_i\|^2 = \frac{s_i}{\sqrt{d}}.$$

Additionally, for each data point, we sample a noise vector from a multivariate Gaussian distribution with $\sigma^2 = \frac{1}{d}$, resulting in $n_i \in \mathbb{R}^d$. Then, each data point is constructed as the sum of its signal and noise vectors as follows:

$$x_i = \mu_{y_i} + n_i.$$

**Exponential long-tail profile.**  Whenever we mention the use of an exponential long-tail profile, we are referring to the exponential profile proposed by Cui et al. (2019). Specifically, for a desired imbalance ratio $R = N_{\max}/N_{\min}$ and a maximum sample size $N_{\max}$, the number of samples for each class is defined as:

$$N_i = N_{\max} \times \left(\frac{1}{R}\right)^{\frac{c-i+1}{c}}.$$

### G.1    Evaluating the impact of dimension on the validity of our approximation (Figure 1)

The approximations employed in this study rely on dimension dependent concentration bounds. To evaluate their effectiveness, we generate three distinct instances of our problem across varying dimensions, aiming to measure how accurately our approximation predicts the actual error obtained by the exact predictors (i.e., predictors that are obtained using the true feature kernel rather than the expected kernel). Each sub-figure in Figure 1 displays the empirical and theoretical worst-class errors eq. (2) for the maximum margin (MM), class-dependent temperature (CDT), margin adjustment (MA), and logit adjustment (LA) predictors. Specifically, for MA and LA, we present the worst class error achieved at the near-optimal parameters discussed in Section 4. For CDT, we employed $\delta_i = \left(\frac{1}{N_i}\right)^{\gamma^\star}$, where $\gamma^\star$ represents the $\gamma$ value that results in the lowest worst class error based on our theoretical predictions.

Our main goal in conducting experiments with diverse problem instances is to showcase situations where MA surpasses LA and vice versa, as well as instances where all methods demonstrate comparable performance. We repeat each experiment with 10 seeds; in the figures, the shaded area corresponds to two standard deviations from the empirical mean (one standard deviation above and below), providing a measure of variability. The solid line with markers represents our theoretical approximation for the worst-class error, enabling a comparison between the empirical and theoretical outcomes. Lastly, in all the experiments we use 4 classes and a balanced test set with 500 samples per class.

**Common classes have weaker signals (left panel).** We used $N_i$ values of 5, 50, 100 and 200, and corresponding $s_i$ values of of 0.5, 0.3, 0.2 and 0.1. MA performs better than LA, and CDT performs nearly as badly as MM, consistent with Proposition 4.4 and Proposition 4.6.

**Common classes have stronger signals (middle panel).** We used the same $N_i$ values as in the previous panel, but changed the corresponding $s_i$ values to 0.5, 0.7, 0.9 and 1.1. LA performs better than MA, and CDT performs nearly as badly as MM, consistent with Proposition 4.4 and Proposition 4.6.

**Less class imbalance (right panel).** We used $N_i$ values of 20, 35, 80, 100, and corresponding $s_i$ values of of 0.4, 0.3, 0.25 and 0.2. In this less imbalanced instance, CDT performs similarly to MA and LA, while MM still performs noticeably worse.

### G.2 Performance of the MM predictor and norm deviation

In this section, we provide a detailed empirical analysis of the failure of the maximum margin (MM) predictor under class imbalance.

Recall the approximate error of the MM predictor from Appendix D:

$$\lim_{d \to \infty} \mathrm{Err}_{y \to k}(\tilde{W}^{\mathrm{mm}}, \tilde{b}^{\mathrm{mm}}) = Q\left( \frac{s_y N_y + \phi_k^{\mathrm{mm}}(\rho)}{\sqrt{N_y + N_k + \psi_{k,k}^{\mathrm{mm}}(\rho)}} \right),$$

where,

$$\phi_k^{\mathrm{mm}}(\rho) = \lim_{d \to \infty} -\sqrt{d}\frac{\frac{1}{\xi_k} - \frac{1}{\xi_y}}{(M + \rho)} \quad \text{and} \quad \psi_{k,k}^{\mathrm{mm}}(\rho) = \lim_{d \to \infty} -\frac{M + 2\rho}{(M + \rho)^2}\left( \frac{1}{\xi_k} - \frac{1}{\xi_y} \right)^2.$$

By examining these equations, we observe that when $\rho < \infty$, i.e., learning with bias, and $N_k > N_y$, it holds that:

$$\phi_k^{\mathrm{mm}}(\rho) = -\infty.$$

This results in a worst-class error lower bound of 1. On the other hand, when $\rho = \infty$ i.e., learning without bias, we get that:

$$\lim_{d \to \infty} \mathrm{Err}_{y \to k}(\tilde{W}^{\mathrm{mm}}, \tilde{b}^{\mathrm{mm}}) = Q\left( \frac{s_y N_y}{\sqrt{N_y + N_k}} \right) = Q\left( \frac{s_y \sqrt{N_y}}{\sqrt{1 + \frac{N_k}{N_y}}} \right). \tag{85}$$

This observation indicates that as the imbalance ratio $\frac{N_k}{N_y}$ increases, the lower bound for the worst-class error becomes $\frac{1}{2}$, which leads to a high test error. In other words, when there is a significant class imbalance, the predictor is more likely to misclassify instances from minority classes, resulting in a higher overall error rate. We validate this observation in the following experiment.

**Experiment details.** We use four classes, with $N_{\max} = 200$, and vary the exponential long tail profiles from an imbalance ratio of 1 to 100 in $\mathbb{R}^d$ where $d = 10^5$. The signal strengths are set inversely proportional to the number of samples in each class, i.e., $s_i = \frac{1.2}{\sqrt{N_i}}$ (which allows us to fix the signal effect in the numerator of eq. (85) to 1.2). For testing the performance of the learned predictor, we use a balanced test set with 500 samples per class. The results are reported over 5 different seeds, where the shaded area represents two standard deviations from the empirical mean (one above and one below), and the solid line with markers represents our theoretical model prediction for the class error.

**Discussion.** Figure 5 demonstrates that our approximation is tight even at moderate dimension. Specifically, we observe a catastrophic failure of the MM predictor with bias even in scenarios with extremely small class imbalance. Furthermore, the error of the predictor without bias on minority classes increases as the imbalance ratio becomes larger.

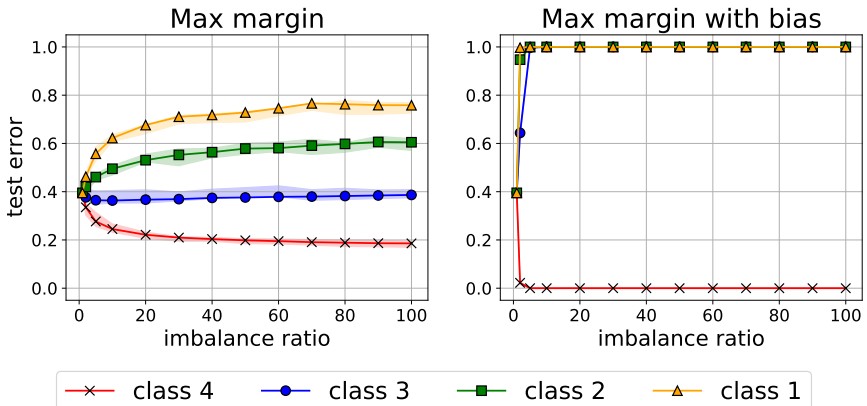

**Figure 5.** Per-class error vs. imbalance ratio for MM predictors. The $x$-axis shows the imbalance ratio, i.e., the ratio between the largest and smallest per-class sample sizes. The $y$-axis represents the test error, and we plot the test error of each class. The left side shows the test error of the maximum margin (MM) predictor without bias, while the right side shows the test error of the MM predictor with bias. The shaded regions show empirical results (two standard deviations over 5 random seeds) and the solid lines show our analytical approximation.

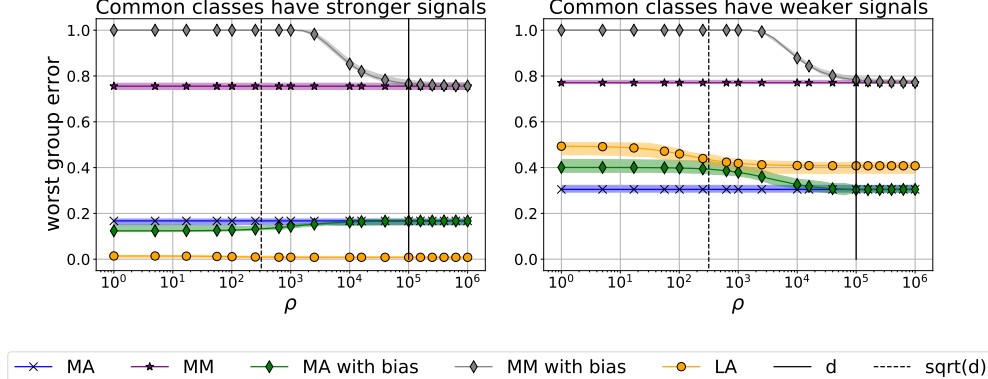

**Figure 6.** Worst class error vs. $\rho$ for MM, MA and LA in **linear classification** on synthetic Gaussian data at the near-optimal hyperparameters presented in Section 4. The shaded regions show empirical results (two standard deviations over 5 random seeds) and the solid lines show our analytical approximation.

## G.3 Testing discontinuity in $\rho$

Our analysis of the MA and MM predictors in Section 4 reveals a discontinuity in the lower bound of the worst class error with respect to $\rho$ as the dimension $d$ approaches infinity. Specifically, for $\rho < \infty$, we observe a certain value in the lower bound of the worst class error at the near-optimal margins of MA, while for $\rho = \infty$, we encounter a different lower bound that is independent of $\rho$ eq. (86).

$$\widetilde{\mathrm{Err}}_{\mathrm{wc}}^{\mathrm{ma}}(\delta^\star, \rho) = \begin{cases} \max_{k \neq y} Q\left(\frac{1}{\sqrt{(s_y^2 N_y)^{-1} + (s_k^2 N_k)^{-1}}}\right) & \rho = \infty \\ \max_{k \neq y} Q\left(\frac{s_y + s_k}{2\sqrt{N_y^{-1} + N_k^{-1}}}\right) & \rho < \infty. \end{cases} \tag{86}$$

This raises an interesting question regarding how $\rho$ affects the transition between learning with bias and without bias in finite dimension. To investigate this, we conduct experiments where we vary the value of $\rho$ in a finite-dimensional setting. By sweeping over different $\rho$ values, we observe how the errors of the MM, MA, and LA predictors change as a function of $\rho$ (at the near-optimal parameters).

**Experiment details.** We generate two Gaussian datasets, each containing four classes in $\mathbb{R}^d$ with $d = 10^5$. Additionally, both datasets contain 5, 50, 100, and 200 samples per class. In the first dataset, the signal strengths are aligned with the order of the sample sizes, with values of 0.5, 0.7, 0.9, and 1.1. In the second dataset, the signal strengths are in reversed order, with values of 0.5, 0.3, 0.2, and 0.1. We utilize this setup to illustrate several observations. When the signals align with the class frequencies, the LA predictor outperforms MA, and MA with bias is superior to MA without bias. Conversely, when the signal strengths are reversed, MA without bias becomes the most effective choice. Moreover, we anticipate that MM with bias will yield worse performance in both cases. As $\rho$ increases, we expect MA and MM (with bias) to converge to their corresponding predictors without bias. Furthermore, we employ a balanced test set consisting of 500 samples per class.

**Discussion.** Our results in Figure 6 demonstrate a smooth convergence between predictors with and without bias. Specifically, we observe that there is no midpoint of $\rho$ that surpasses $\rho = 1$ or $\rho = \infty$. Moreover, as our theory suggests we observe a catastrophic failure of the MM predictor with bias even in scenarios with extremely small class imbalance. Additionally, we see the expected behavior of MA when the signals are aligned and reversed. Moreover, as our approximation eq. (67) suggests, we see that the transition between the $\rho < \infty$ regime to the $\rho = \infty$ regime starts when $\rho$ is of the order of $\sqrt{d}$.

### G.4  Classification with RBF kernel

To test the validity of our predictions beyond the scope of our model, we conduct classification experiments using the RBF kernel on imbalanced CIFAR10, MNIST and FashionMNIST datasets that were featurized using ViT-B/32 CLIP model (Radford et al., 2021). Specifically we solve the kernel form of the optimization problems in Section 2.1 (see Appendix A) when $K$ is the RBF kernel (defined below). We consider two different class size profiles. The first profile follows a standard "long-tailed" exponential distribution (Cui et al., 2019; Cao et al., 2019), with 500 examples for the majority class and 5 examples for the minority class. The second profile maintains the same majority and minority sizes but increases the sample sizes of the other classes to worsen the theoretical predictions for CDT.

**Data generation.** For each dataset use in our tests (CIFAR10, MNIST and FashionMNIST), we sample 5, 8, 13, 23, 38, 64, 107, 179, 299, 500 and 5, 100, 120, 140, 160, 180, 200, 220, 300, 500 samples per class for the exponential and modified profiles, respectively. We use the image encoder of pre-trained ViT-B/32 CLIP model (Radford et al., 2021) to extract features in $\mathbb{R}^{512}$ for each image. In addition, we featurize the standard test sets of each dataset and use to test the learned predictors.

**Hyperparameter sweep.** Taking a random feature perspective, using the RBF kernel essentially corresponds to taking $d \to \infty$. Therefore, we use our theoretical tuning that matches this specific setting. For the CDT and MA predictors, we use $\delta_i = \left(\frac{1}{N_i}\right)^\gamma$, while for the LA predictor, we use $\iota_i = \tau\left(\frac{N_i}{N}\right)$ (assuming the signals have equal strengths).

**RBF Kernel.** The RBF kernel is

$$K^{\mathrm{RBF}}_{[i,j]} = \exp\left(-\frac{\|x_i - x_j\|^2}{2\zeta^2}\right),$$

where $K^{\mathrm{RBF}}$ is an $N \times N$ matrix and $\zeta$ is the kernel width. It is important to note that as $\zeta$ becomes smaller, the kernel becomes closer to a diagonal matrix, agreeing with our expected kernel approximation with the scaling $\|\mu_i\|^2 = \frac{s_i}{\sqrt{d}}$ and in the limit in the limit $d \to \infty$. Conversely, as $\zeta$ becomes larger, the kernel widens and becomes less diagonal. We illustrate this in Figures 7a to 7c.

**Distance from theory.** To quantify the similarity between $K^{\mathrm{RBF}}$ and the expected $K$ kernel (see eq. (9)) used in our theoretical analysis, we define the "distance from theory" as follows:

$$\mathrm{DT}\left(K^{\mathrm{RBF}}\right) = \min_{\alpha_1, \alpha_2 \in \mathbb{R}} \left\|K^{\mathrm{RBF}} - \alpha_1 B - \alpha_2 I\right\|_{\mathrm{F}},$$

where $I$ is the $N \times N$ identity matrix and $B \in \mathbb{R}^{N \times N}$ has elements $B[i,j] = \mathbb{1}_{\{y_i = y_j\}}$. Thus, the distance from theory measures the Euclidean (Frobenius) distance between $K^{\mathrm{RBF}}$ and the block matrix $K' = \alpha_1^\star B + \alpha_2^\star I$

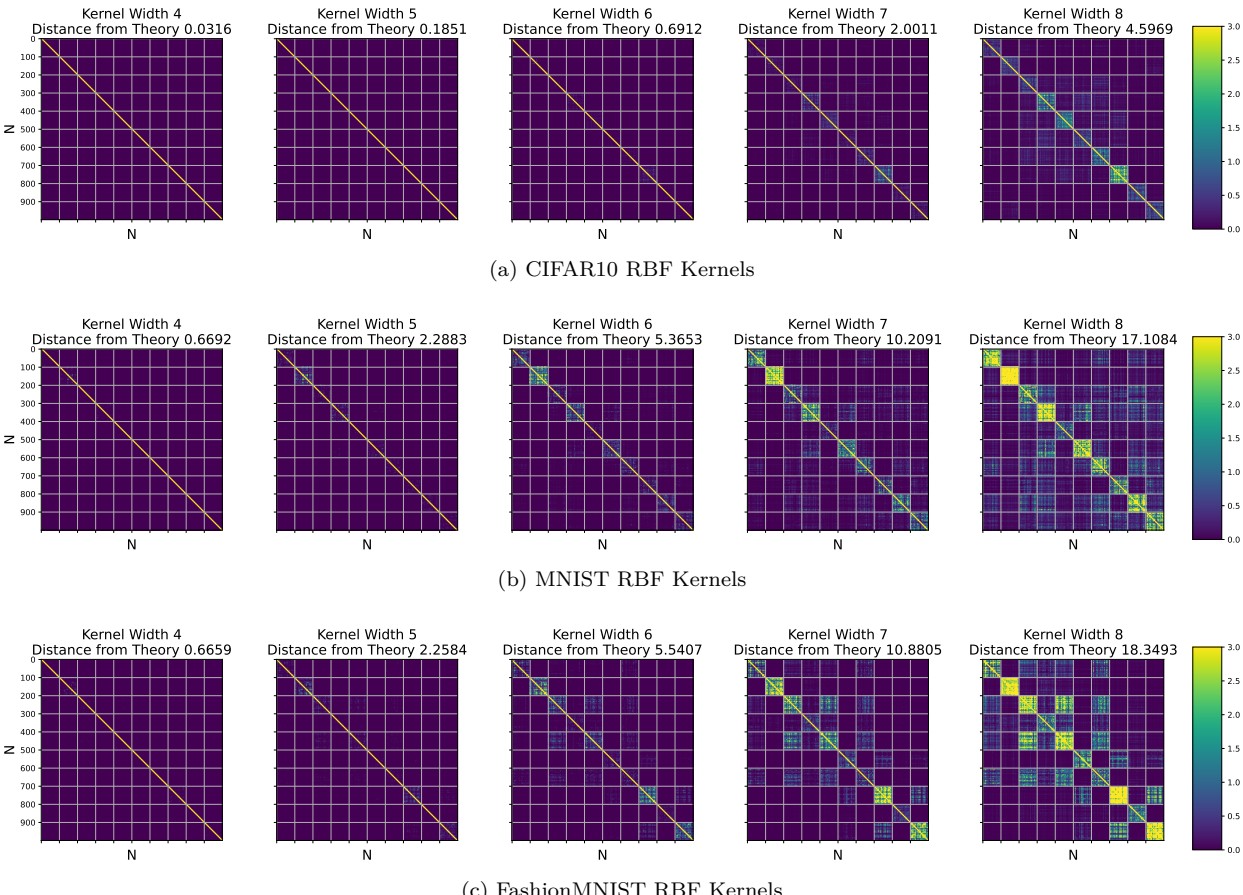

(a) CIFAR10 RBF Kernels

(b) MNIST RBF Kernels

(c) FashionMNIST RBF Kernels

**Figure 7.** Illustration of the RBF kernel on (a) CIFAR10 (b) MNIST and (c) FashionMNIST with varying widths using 100 samples per class. The "distance from theory" at the widest kernel width indicates that CIFAR10 has properties that fits the best to our theoretical model.

that best approximates $K^{\mathrm{RBF}}$ and is consistent with our data model (see sections 2.3 and 3.1) when the signals have equal strengths.

**Discussion.** In Figure 8, we extend Figure 3 (which corresponds to kernel width 6.0) to show results with additional kernel widths. The figure illustrates that many of our predictions continue to hold in this setting, particularly the high errors of CDT and MM (especially with bias), which persist across all tested kernel widths. Additionally, we observe that a kernel width of 6.0 appears to agree most closely with our theory. As seen in Figure 8, as we approach this kernel width, the optimal tuning parameters tend to be around $\gamma, \tau = 1$. Conversely, as we move away from it (e.g., kernel widths of 7.0 and 8.0), the optimal tuning parameters deviate from $\gamma, \tau = 1$.

Moreover, Figures 9 and 10 show similar observations for MNIST and FashionMNIST datasets. Specifically, in both datasets, we observe that the "modified class profile" leads CDT to failure in all kernel widths. Margin adjustment with bias, on the other hand, has optimal hyperparameters around $\gamma = 1$.

For margin adjustment without bias and logit adjustment, our theoretical predictions hold better for CIFAR10 than they do for MNIST and FashionMNIST. This is consistent with the higher "distance from theory" computed for the latter two datasets Figures 7a to 7c. Two likely causes for the higher distance are: (1) the class signals are non-orthogonal, and (2) the signal strengths are not of equal magnitude. For FashionMNIST (Figure 7c), both causes seem to be at play, since there are significant kernel magnitudes across classes (indicating class signals are not orthogonal), and also the classes have unequal signal strengths. For MNIST (Figure 7b), we see that in a high kernel width, $K^{\mathrm{RBF}}$ looks like the signal vectors of different classes

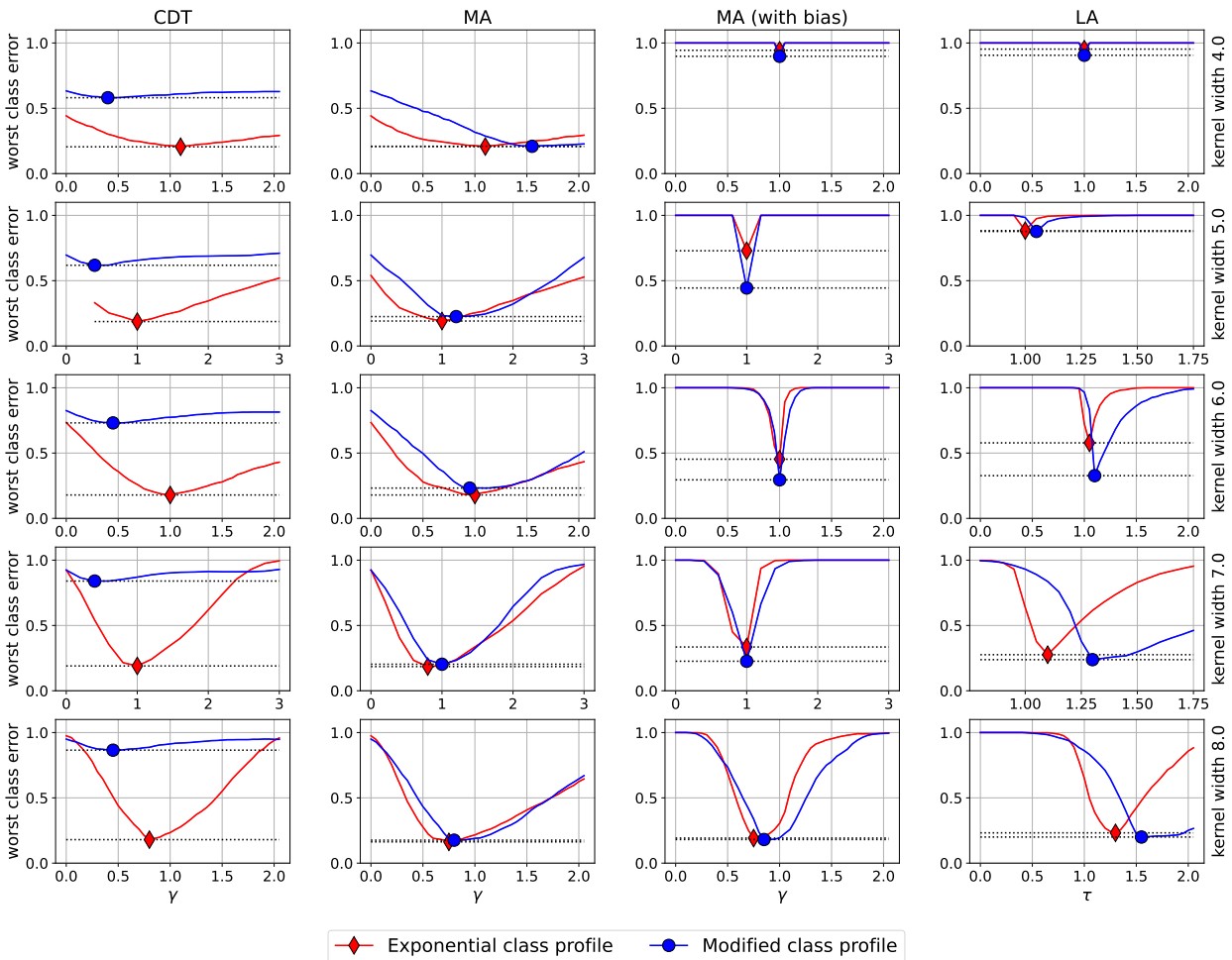

**Figure 8.** Empirical worst class test error vs. loss tuning hyperparameter for the different methods we consider and **kernel classification** on two imbalanced versions of CIFAR10 with varying kernel widths.

are nearly orthogonal (as there are no bright blocks or cells that are not centered around the diagonal), but still the classes have unequal strengths, as some class blocks are much brighter than other class blocks.

## G.5 CLIP fine-tuning

We conduct experiments to test our theoretical observations beyond the realm of linear classifiers and Gaussian data by fine-tuning zero-shot CLIP (ViT-B/32) (Radford et al., 2021) on three imbalanced CIFAR10 datasets, each varying in the number of classes used.

In the **three-class dataset**, we include images of airplanes, cats, and horses from CIFAR10, with sample sizes per class of 1000, 500 and 5, respectively.

The **four-class dataset** consists of images of airplanes, cats, deer, and horses from CIFAR10, with sample sizes per class of 1000, 500, 300 and 5, respectively.

The **ten-class dataset** comprises images from all classes of CIFAR10, with 500, 450, 350, 300, 280, 250, 230, 200, 150, and 5 sample sizes per class, respectively.

We consider a reduced number of classes in an attempt to make it more challenging for CDT to succeed, as our theory suggests that a significant imbalance between consecutive classes hinders CDT's performance. For each model, we use the official CIFAR10 test set and keep only the relevant classes for each experiment.

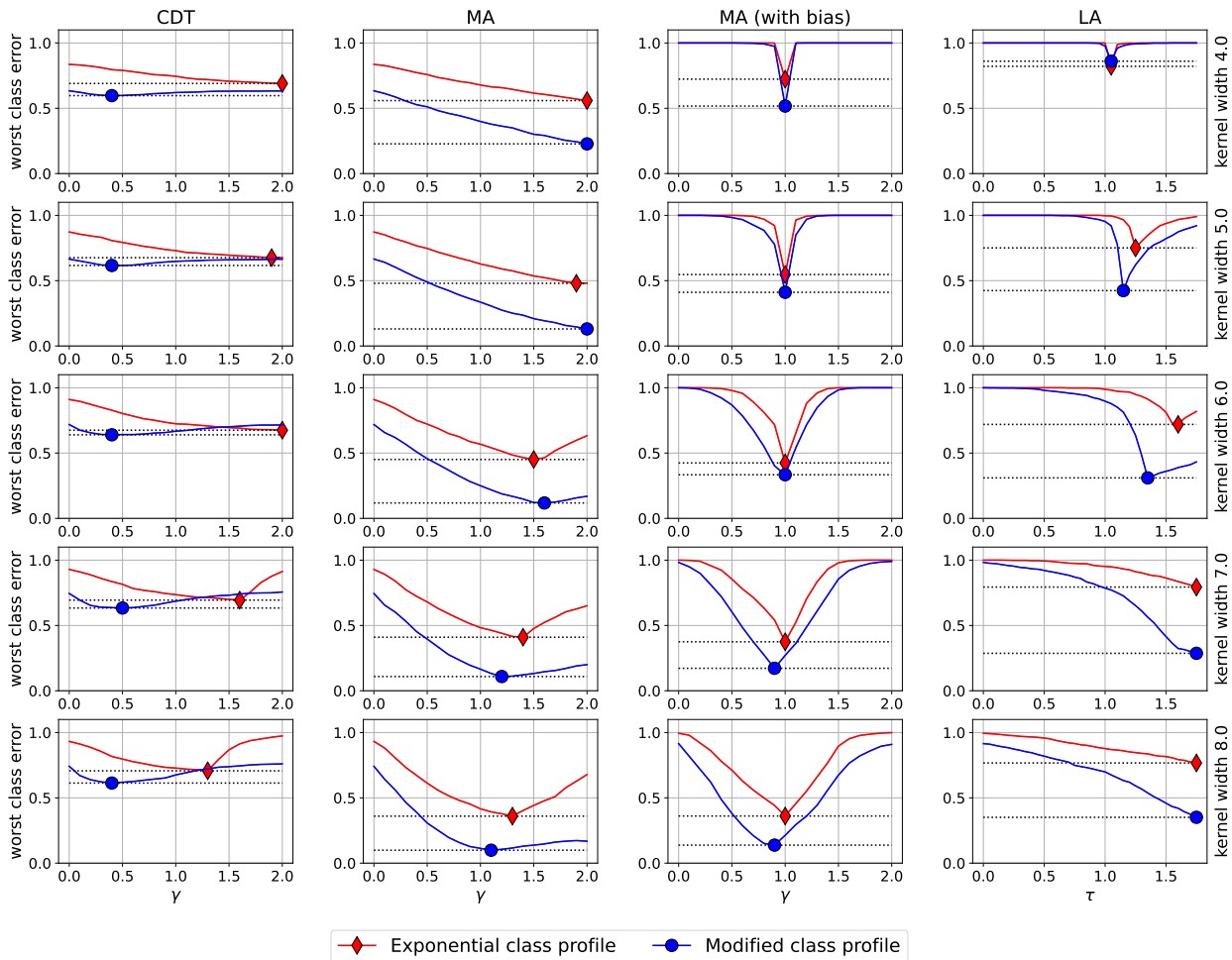

**Figure 9.** Empirical worst class test error vs. loss tuning hyperparameter for the different methods we consider and **kernel classification** on two imbalanced versions of MNIST with varying kernel widths.

For each instance, we use PyTorch (Paszke et al., 2019) to fine-tune CLIP by minimizing the CDT and MA losses using the standard tuning parameter $\delta_i = \left(\frac{1}{N_i}\right)^{\gamma}$ over the training set. In addition we test the post-hoc bias adjustment of the MM predictor using both standard $\tau \log\left(\frac{N_i}{N}\right)$ and theoretical tuning $\iota_i = \tau \frac{N_i}{N}$.

**Zero-shot initialization.** To initialize the zero-shot model, we utilize templates from the official GitHub repository of CLIP (Radford et al., 2021) for CIFAR10.

**Training procedure.** Fine-tuning is performed using PyTorch (Paszke et al., 2019) while training is conducted for 1000 epochs, employing the SGD optimizer with a batch size of 128, no momentum, no weight decay, and gradient clipping with global norm threshold of 1. The learning rate is set to 1e-4 with cosine learning rate scheduler, and the training is distributed across 4 GPUs.

**Discussion.** Figure 11 demonstrates partial agreement with our theory. CDT performs comparably to MA, and our theoretical setting of LA is slightly superior in the ten-class dataset but inferior in the other two cases. We note that CDT exhibits greater sensitivity to the tuning of $\gamma$ than MA, which contradicts the predictions of Behnia et al. (2023). Additionally, we observe that CDT does not converge to zero training error (Figure 11b), supporting Remark 2.1.

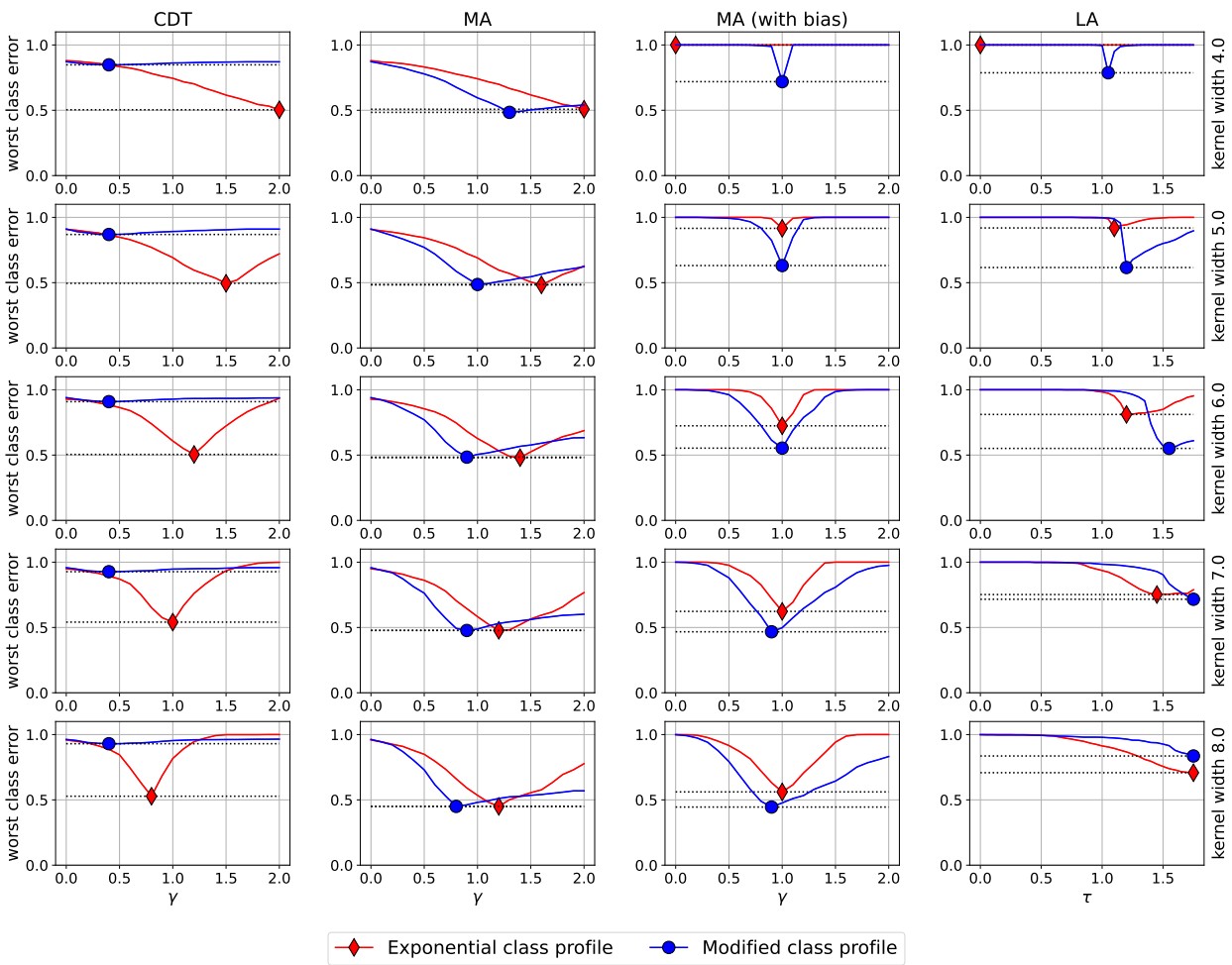

**Figure 10.** Empirical worst class test error vs. loss tuning hyperparameter for the different methods we consider and **kernel classification** on two imbalanced versions of FashionMNIST with varying kernel widths.

## G.6 Linear classification with Gaussian model

To validate the theoretical observations presented in Section 4, we conduct experiments to test the performance of the predictors we analyze in this work on various instances of our model. The empirical validation aims to demonstrate the following key points:

1. The theoretical predictions for the error of all predictors are tight even in a finite dimension.

2. There are instances of data where CDT performs very poorly, while MA and LA can perform extremely well.

3. When the signals are of equal strength, MA with and without bias achieve the same optimum for the worst class error. Additionally, LA surpasses MA.

4. There are instances where MA without bias outperforms MA with bias and LA.

5. At least in a one-dimensional hyperparameter sweep, the near-optimal value predicted from our theory is indeed optimal.

6. In certain scenarios, our theoretical tuning outperforms standard tuning.

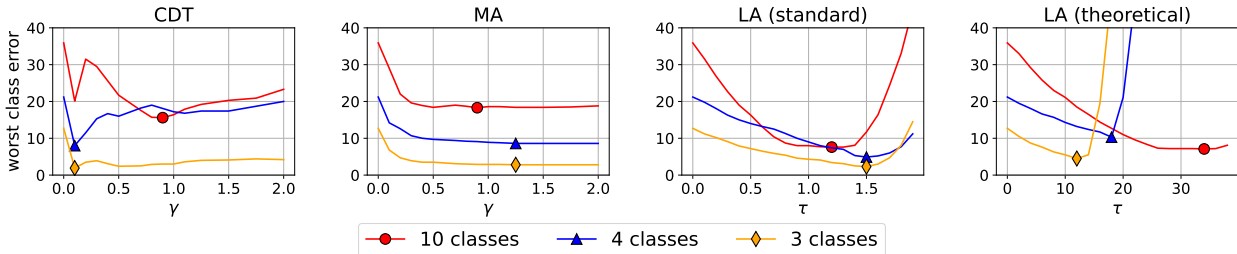

(a) CLIP fine-tuning and post-hoc bias adjustment worst class error (repeating Figure 3)

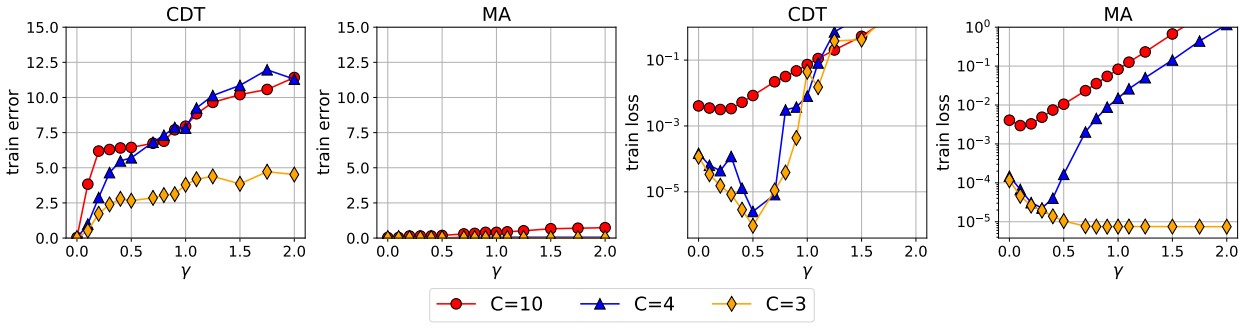

(b) Train error and train loss of CDT and MA

**Figure 11.** (a) Empirical worst class test error vs. loss tuning hyperparameter and (b) training loss and error vs. loss tuning hyperparameter for CDT and MA in **neural network fine-tuning** on class-imbalanced subsets of CIFAR10.

**Data setting.** We conduct experiments using five different instances of Gaussian mixture datasets that conform to our data model described in Section 2.3. Each dataset consists of four classes in $\mathbb{R}^d$, where $d = 10^5$. The test set for each instance is balanced, with 500 samples per class. The instances differ in terms of the following characteristics:

**Instance 1 (equal signals)**: has 5, 50, 100 and 200 samples per class and signal strengths 0.5 for all classes.

**Instance 2 (reversed signals)**: has 5, 50, 100 and 200 samples per class and signal strengths 0.5, 0.3 0.2 and 0.1, respectively.

**Instance 3 (aligned signals)**: has 5, 6, 100 and 200 samples per class and signal strengths 0.5, 1.0, 1.5 and 2.0.

**Instance 4 (weak aligned signals)**: has 5, 50, 100 and 200 samples per class and signal strengths 0.5, 0.7,0.9 and 1.1.

**Instance 5 (strong aligned signals)**: has 5, 50, 100 and 200 samples per class and signal strengths 0.5, 1.0, 1.5 and 2.0., respectively.

**Experiment details.** We tune each method by sweeping over a scalar hyperparameter. Specifically we tune MA and LA using our theoretical expressions from Section 4, setting $\delta_i = (\delta_i^\star)^\gamma$ and $\iota_i = \tau \iota_i^*$, respectively, and sweeping over $\gamma$ and $\tau$ such that value 1 recovers the theoretical tuning and value 0 reduces to standard maximum margin. For CDT we use standard tuning and set $\delta_i = N_i^{-\gamma}$. Additionally, whenever we mention that we use standard tuning for MA and LA we use $\delta_i = N_i^{-\gamma}$ and $\iota_i = \tau \log N_i$ respectively.

**Balanced error.** The balanced error is defined by

$$\mathrm{Err}_{\mathrm{bal}}\left(W, b\right) := \frac{1}{c} \sum_{y=1}^{c} \mathrm{Err}_y\left(W, b\right)$$

Where $\mathrm{Err}_y\left(W, b\right)$ is defined in eq. (1).

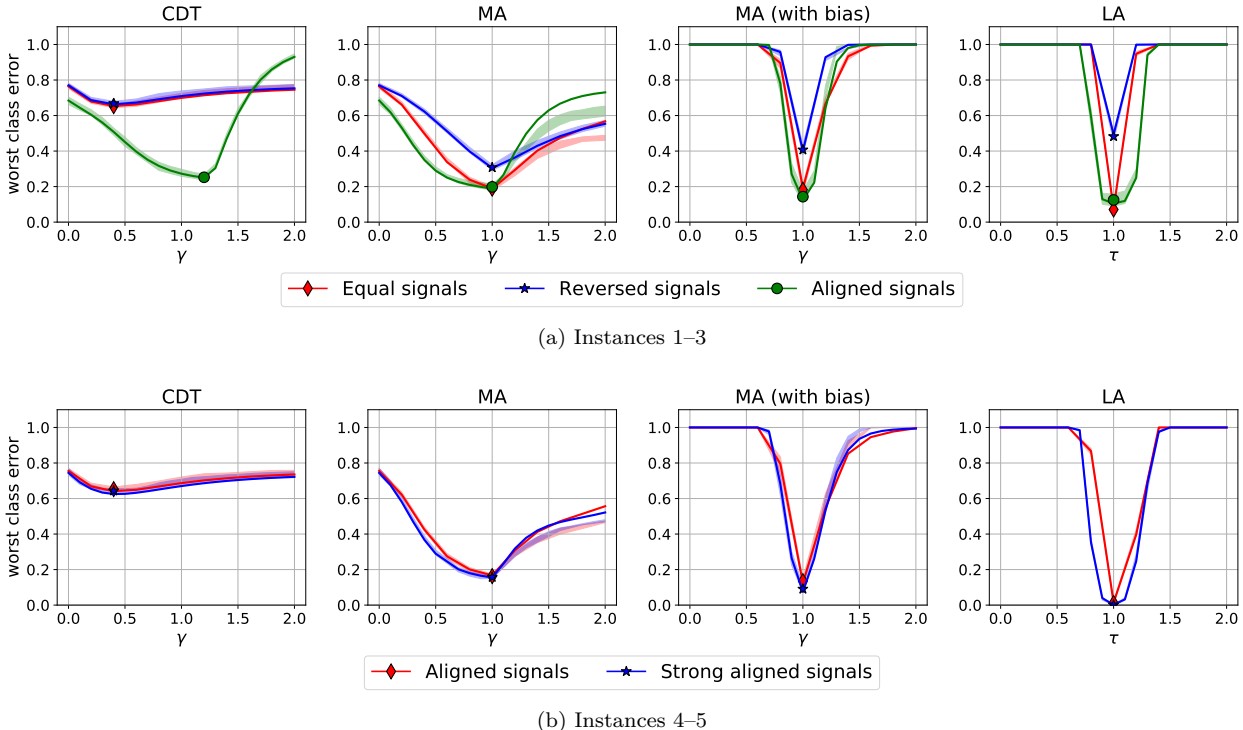

(a) Instances 1–3

(b) Instances 4–5

**Figure 12.** Worst class error vs. loss tuning hyperparameter for the different methods we consider in **linear classification** on synthetic data from our model. The shaded region show empirical results (two standard deviations over 5 random seeds) and the solid lines show our analytical approximation. Figure 12a is exactly the same as the top row (showing worst class error) in Figure 2.

**Macro $F_1$ score.** The macro $F_1$ score is calculated by averaging the $F_1$ scores for each class. The $F_1$ score for each class $y$ is given by:

$$(F_1)_y = \frac{2 \cdot \text{precision}_y \cdot \text{recall}_y}{\text{precision}_y + \text{recall}_y}$$

where

$$\text{precision}_y = \frac{\text{TP}_y}{\text{TP}_y + \text{FP}_y} \quad \text{and} \quad \text{recall}_y = \frac{\text{TP}_y}{\text{TP}_y + \text{FN}_y},$$

and $\text{TP}_y$, $\text{FP}_y$, $\text{FN}_y$ are the true positive, false positive and false negative rates for class $y$, respectively, assuming the data comes from the (class imbalanced) training distribution.

The macro $F_1$ score is then computed by averaging the $F_1$ scores across all $C$ classes:

$$\text{macro } F_1 = \frac{1}{C} \sum_{i=1}^{C} (F_1)_i.$$

**Discussion.** Our experimental findings, as illustrated in Figures 12 and 13, provide strong support for the validity of our theory. Specifically, in Figure 12a, we observe that when the signal strengths are equal, both MA with and without bias exhibit equivalent performance when $\gamma = 1$, while LA outperforms MA when $\tau = 1$. Furthermore, across all test cases, we find no value of $\gamma$ that improves CDT's performance on the worst class. Additionally, Figure 12b demonstrates that CDT can achieve good performance in certain cases. In Figure 13, we compare the worst class error of standard tuning to theoretical tuning for all methods. The results indicate that, in all tested scenarios except for instance 5, theoretical tuning consistently outperforms standard tuning, particularly in the case of LA. Finally, Figures 14 illustrate the tightness of our analytical expression per class in each tested instance

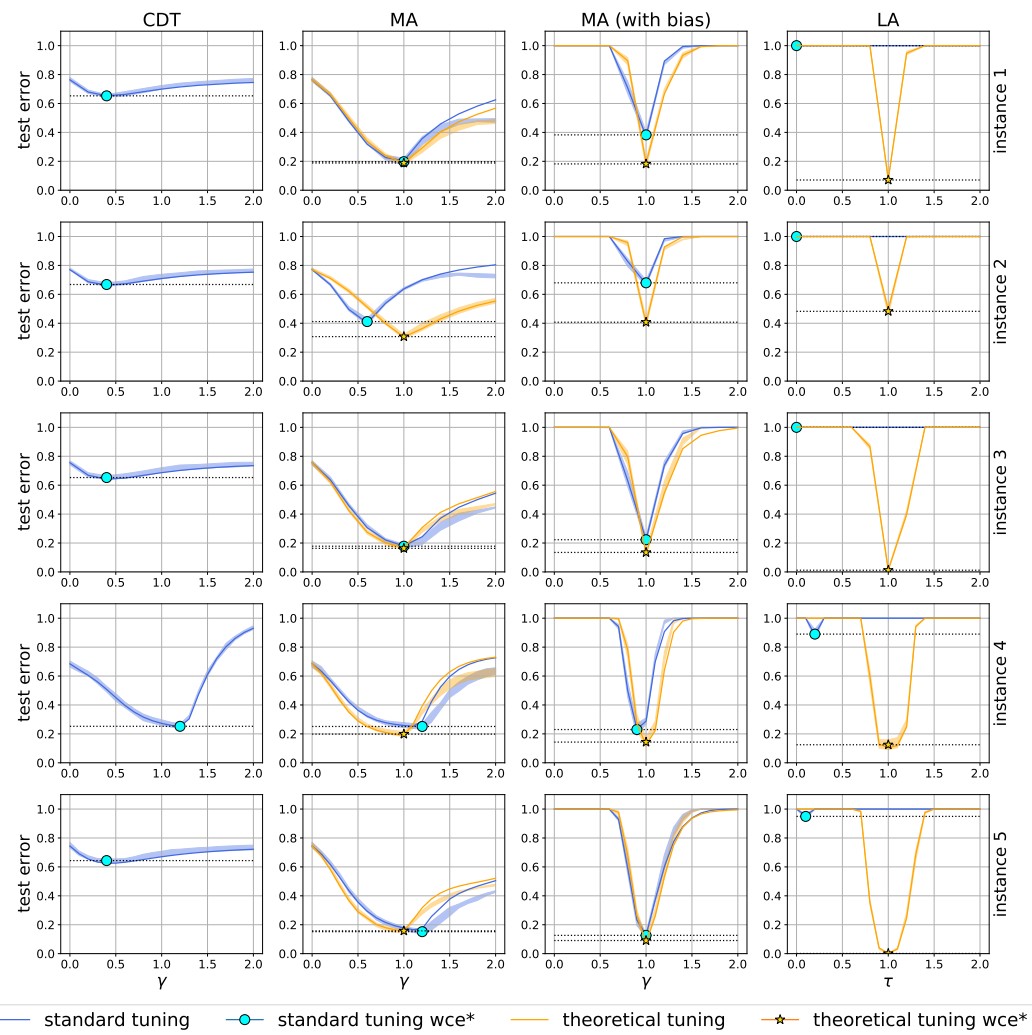

**Figure 13.** Worst class error vs. loss tuning hyperparameter (standard blue and theoretical orange) for the different methods we consider in **linear classification** on synthetic data from our model. The shaded regions show empirical results (two standard deviations over 5 random seeds) and the solid lines show our analytical approximation.

## G.7    Empirical demonstration of the limitations of CDT

We conduct two additional experiments to examine the effect of the $\delta$ hyperparameter on the decision boundary of the CDT and MA predictors (without learning a bias parameter).

**Data generation.** We generate two synthetic datasets (binary and multiclass) in $\mathbb{R}^2$ by following these steps for each class:

1. Sample $N_i$ random samples from $\mathcal{N}(0, I_2)$.

2. Generate a random projection matrix $Q_i$ from $\mathbb{R}^2$ to $\mathbb{R}^2$

3. Perform random projection using $Q_i$ and add the signal vector $\mu_i$ to the projected points.

Where for the binary experiment we use:

$$\mu_1 = \begin{bmatrix} 2 \\ 2 \end{bmatrix} \ \text{ and } \ \mu_2 = \begin{bmatrix} -1 \\ 2 \end{bmatrix}$$

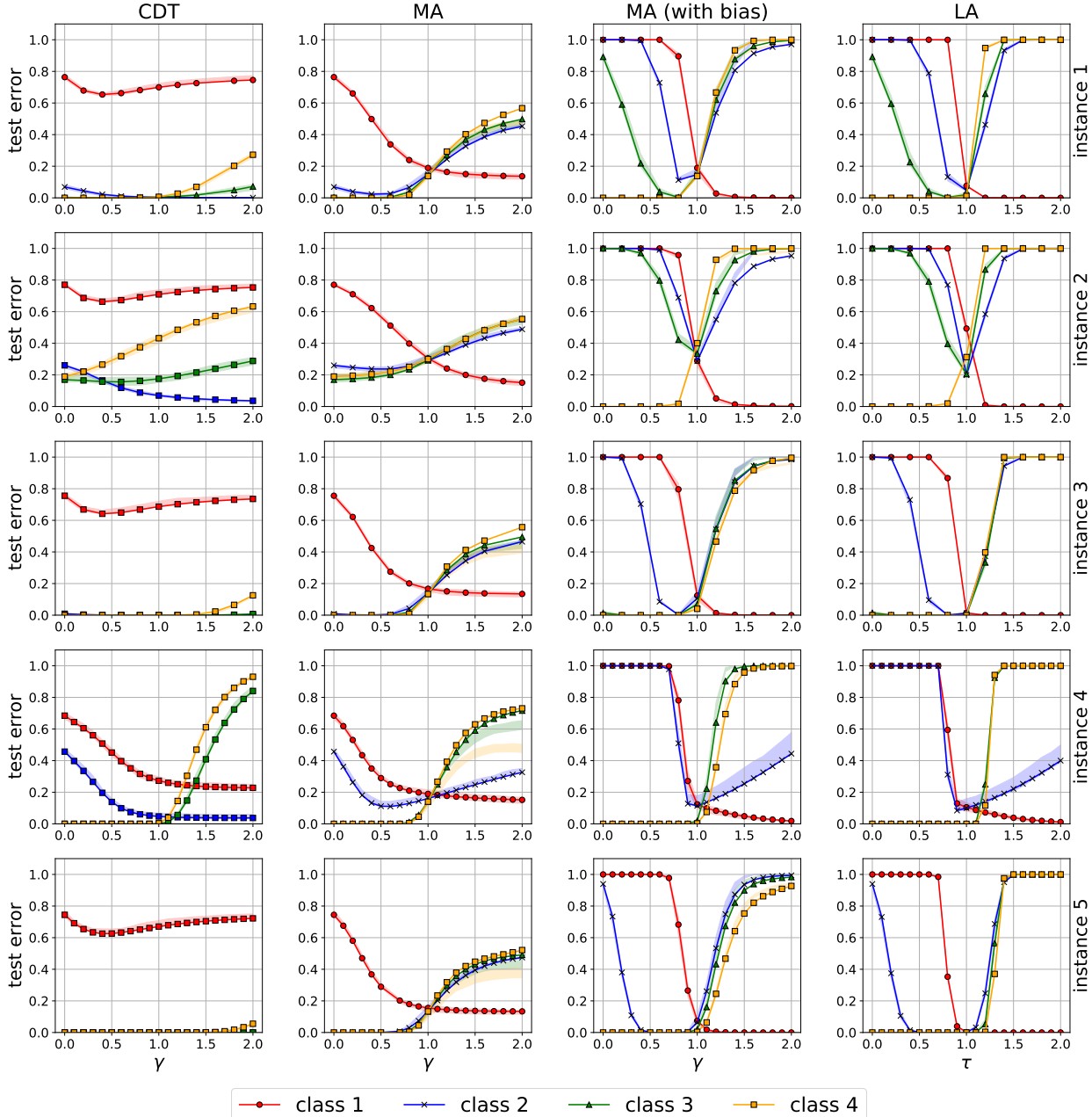

**Figure 14.** Per class error vs. loss **theoretical tuning** hyperparameter for the different methods we consider in **linear classification** on synthetic data from our model. The shaded regions show empirical results (two standard deviations over 5 random seeds) and the solid lines show our analytical approximation.

and for the multiclass experiment we use: Where for the binary experiment we use:

$$\mu_1 = \begin{bmatrix} 3 \\ 0 \end{bmatrix} \quad , \quad \mu_2 = \begin{bmatrix} 0 \\ 3 \end{bmatrix} \quad \text{and} \quad \mu_2 = \begin{bmatrix} 3 \\ 3 \end{bmatrix}$$

**Discussion.** In Figure 15, we observe that in the binary case, the hyperparameter $\delta$ has no effect on the decision boundary of the CDT predictor (see Figure 15a). However, when used in the MA predictor, it allows for shifting the decision boundary towards the majority class, which leads to potential improvement of generalization on minority classes (see Figure 15b). Similarly, in Figure 16, we see that the hyperparameter

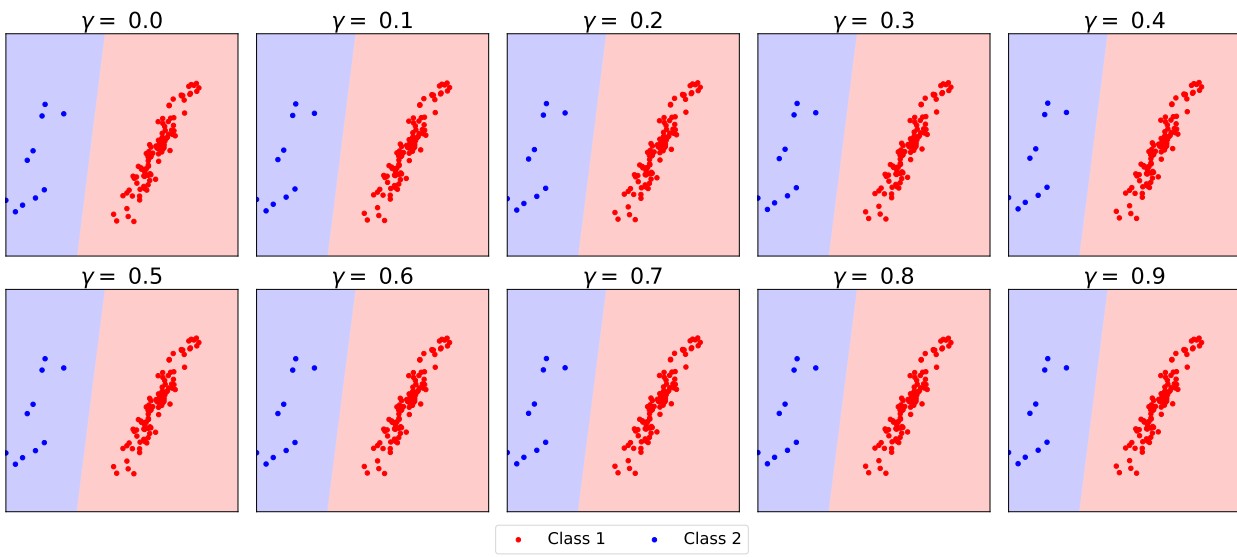

(a) The CDT predictor decision boundary using different choice of hyperparameters

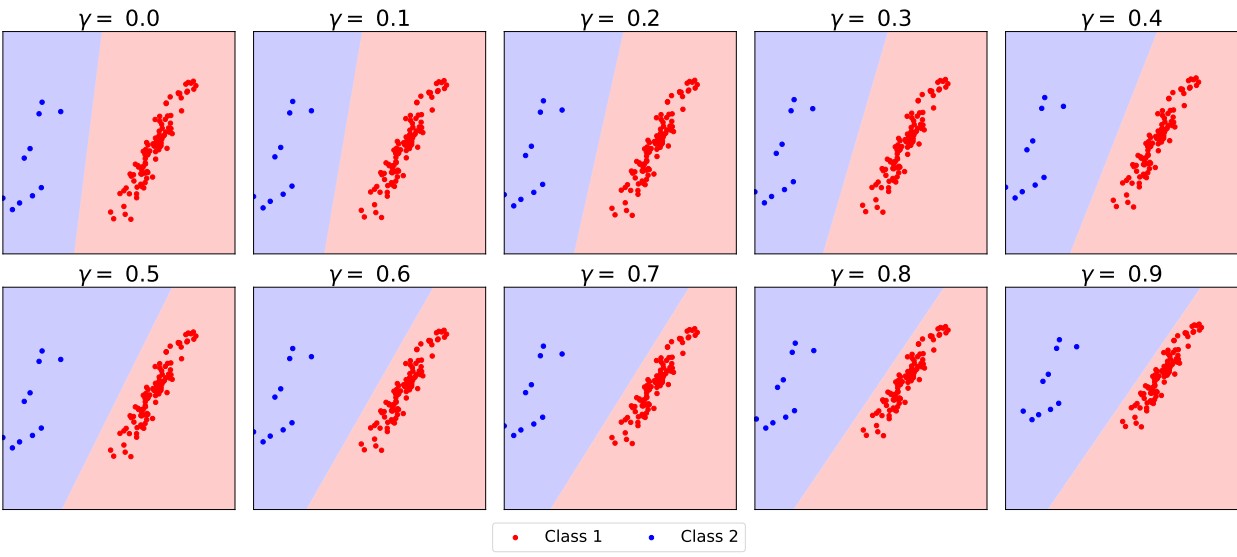

(b) The MA predictor decision boundary using different choice of hyperparameters

**Figure 15.** The effect of $\gamma$ on the decision boundary of (a) CDT and (b) MA predictors in **binary classification**. As it can be seen, in binary classification the hyperparameter $\delta$ has no affect on the decision boundary of the CDT predictor while in MA it changes the angle of the margin towards majority.

$\delta$ can lead to non-trivial training errors for CDT (see Figure 16a), while for MA, it changes the angle of the learned predictor towards the majority class (see Figure 16b).

## G.8   Implicit bias of gradient descent

To validate the implicit bias of gradient descent on the MA and CDT losses described in Appendix B, we compare the test error of the classifiers obtained by gradient descent (Figure 17a) with their corresponding predictors (Figure 17b) on a synthetic Gaussian dataset.

**Dataset.** We use a Gaussian dataset in $\mathbb{R}^d$ for $d = 10^4$ with four classes. Each class contains $5, 50, 80$, and $100$ samples, respectively. The signal strengths for all classes are set to $s_i = 0.5$. Additionally, we use a balanced test set containing 500 samples per class.

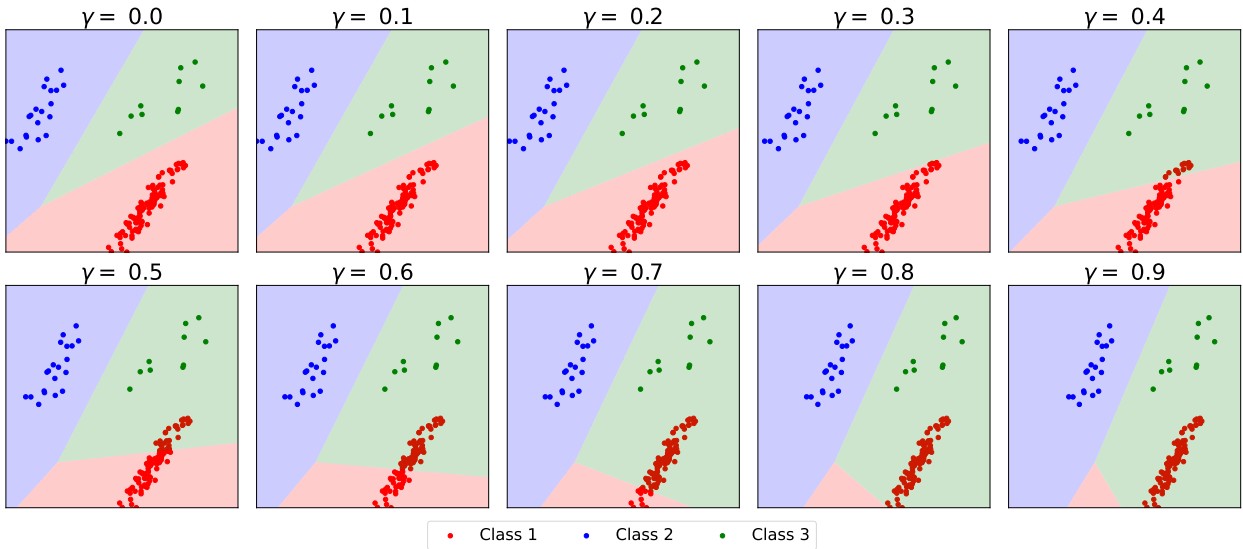

(a) The CDT predictor decision boundary using different choice of hyperparameters

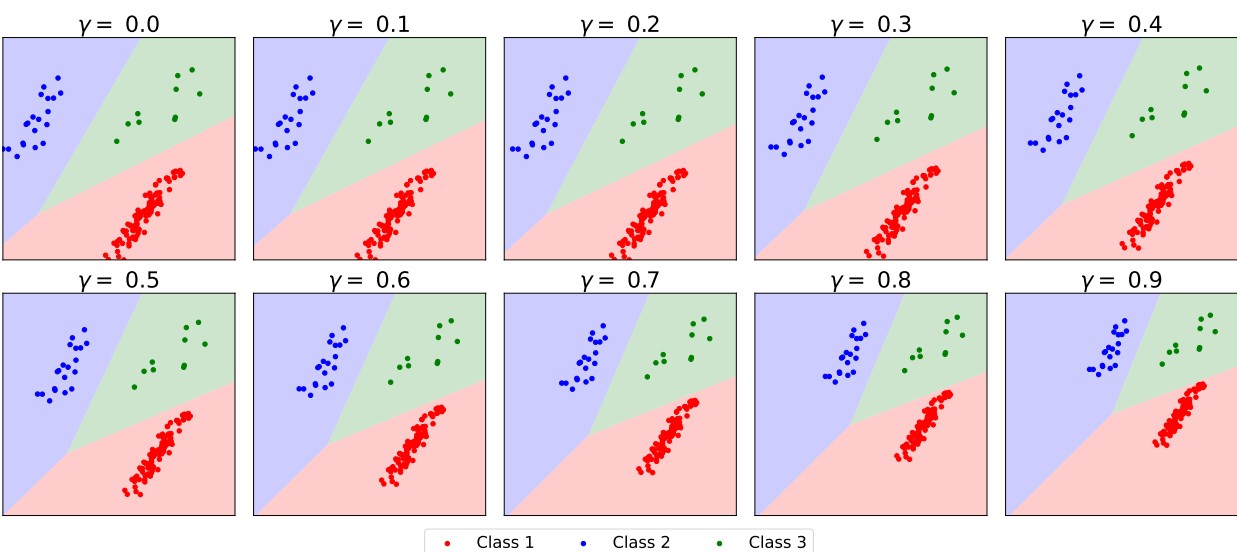

(b) The MA predictor decision boundary using different choice of hyperparameters

**Figure 16.** The effect of $\gamma$ on the decision boundary of the (a) CDT and (b) MA predictors in **multiclass classification**. As it can be seen, in multiclass classification the hyperparameter $\delta$ can lead CDT to have non-trivial train error, while in MA it changes the angle of the margin towards majority.

**Gradient descent training.** We use PyTorch (Paszke et al., 2019) to run gradient descent on the MA and CDT losses (without learning the bias parameter) bias for 20,000 epochs. To avoid learning rate tuning and accelerate convergence we employed a variant of the Polyak step size for functions whose minimal value is 0:

$$\eta_t := 0.05 \cdot \frac{\mathcal{L}(W_t)}{\|\nabla \mathcal{L}(W_t)\|^2} \tag{87}$$

In all experiments we initialized $W_0 = 0$ and did not use weight decay.

**Hyperparameter sweep.** For CDT we use the standard tuning i.e., $\delta_i = \left(\frac{1}{N_y}\right)^{\gamma}$ and for MA we use our theoretical tuning from Section 4.

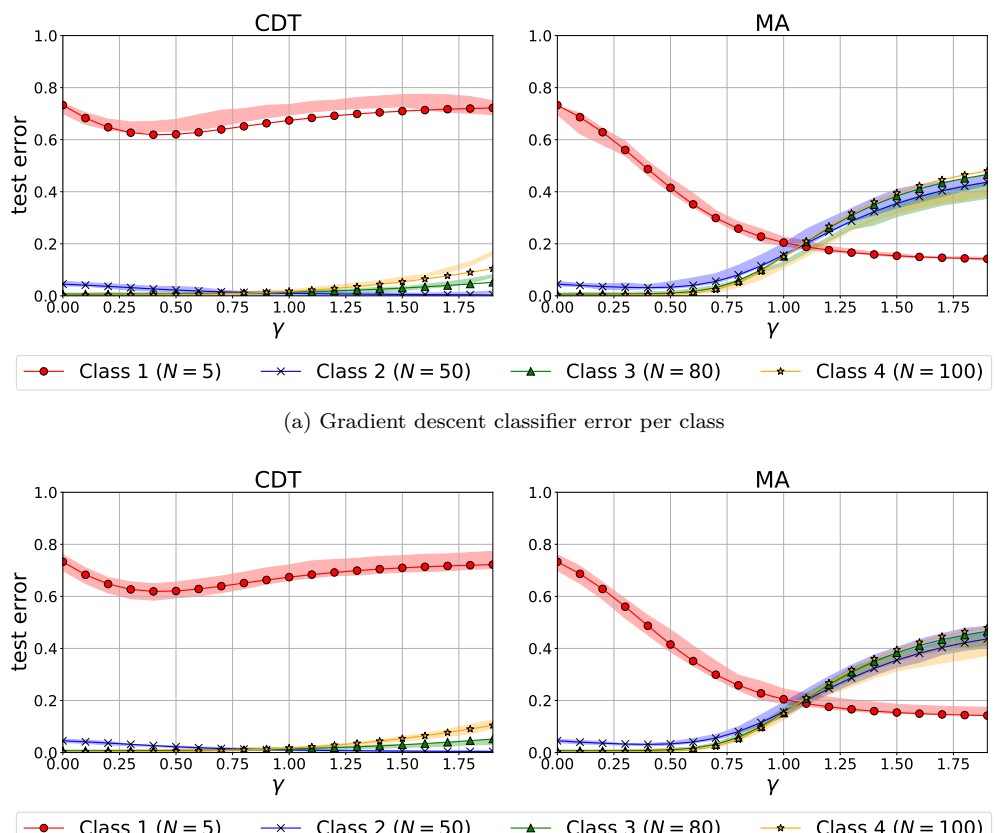

(a) Gradient descent classifier error per class

(b) Exact predictors error per class

**Figure 17.** An illustration of the **implicit bias of gradient descent** on linear classifiers over Gaussian data. (a) Per class error vs. loss hyperparameter using gradient descent (b) Per class error vs. loss hyperparameter using CVXPY (Diamond & Boyd, 2016) to solve the corresponding margin problem. The shaded regions show empirical results ($\pm$ one standard deviation over 3 random seeds) and the dashed lines show our analytical approximation for the error.

**Discussion.** Figure 17 shows that, as our theory suggests, gradient descent on different losses converges in the direction of their corresponding predictors, as described in Section 2.1.

### G.9 Extended comparison between LA and MA predictors

To validate whether the LA predictor, tuned for near-optimal parameters with equal strength signals, outperforms the MA predictor under reversed or aligned signal strengths, we compare their worst class error lower bounds at the near-optimal parameters of MA (with and without bias) eq. (19) and LA eq. (22), and calculate the histogram of bound differences for these cases.

**Use case parameter generation.** For each case study, we conduct 500 experiments where, in each experiment, we sample the number of classes uniformly from 3 to 10. Each class size is sampled uniformly from 1 to 200, the dimension $d$ is sampled uniformly from $10^5$ to $10^7$, the signal strengths are sampled uniformly from 0.01 to 1, and $\rho$ is sampled uniformly from 1 to 1000.

**Discussion.** Our findings in Figure 18 show that for aligned signals, LA (tuned for equal signals) is consistently better than MA without bias, though in rare cases MA with bias slightly outperforms it. For reversed signals, LA is generally worse than MA with bias, but there are occasional exceptions. These findings strongly suggest that LA is consistently better than MA without bias for aligned signals, aligning with our additional experiments in Section 5. Additionally, the converse does not hold: for reversed signals, MA is not always superior to LA, even when LA is not optimally tuned.

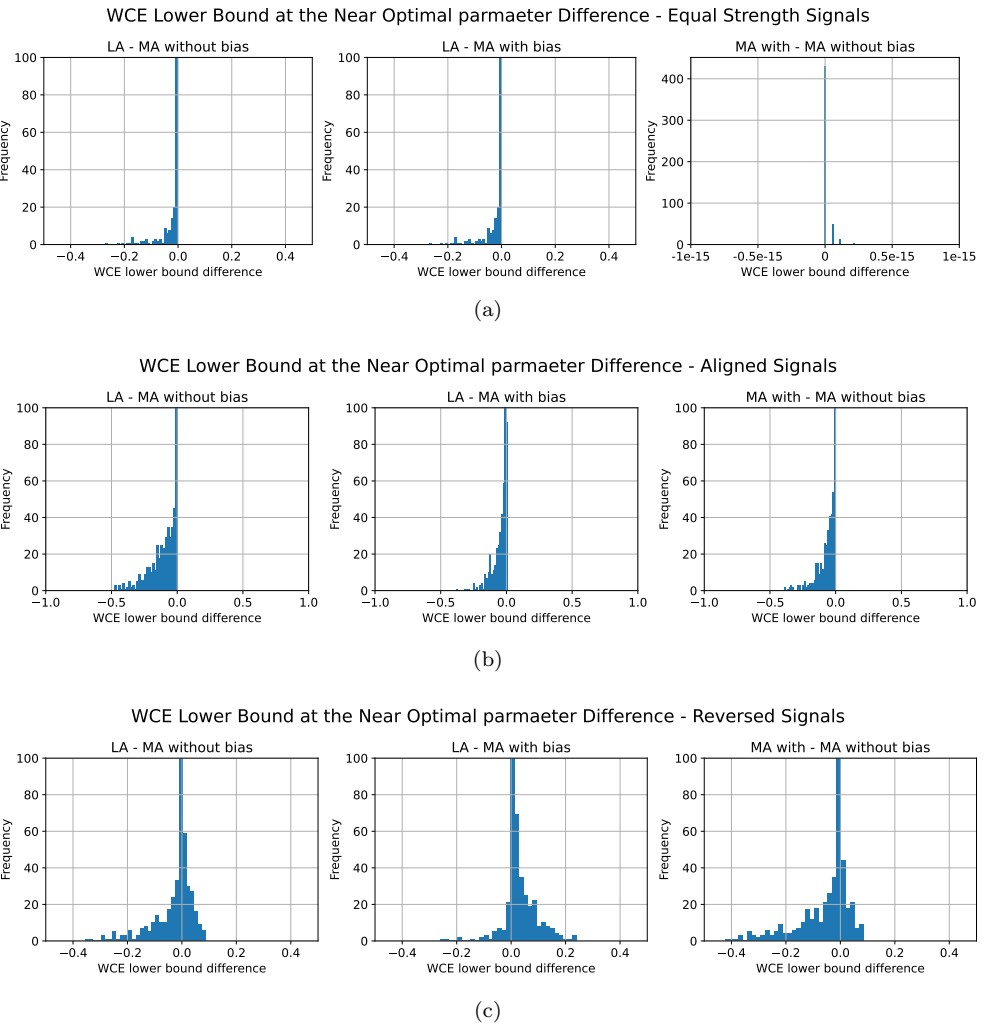

**Figure 18.** Histogram of the worst-class error lower bound differences between LA at its near-optimal parameter $\iota^*$ eq. (22) and MA with and without bias at the near-optimal parameter $\delta^\star$ eq. (19). Shown for cases with (a) equal signal strengths, (b) aligned signal strengths, and (c) reversed signal strengths. Each case confirms the validity of proposition 4.2. Row (a) also confirms proposition 4.4. Row (b) demonstrates that, even with near-optimal parameters for the equal-strength signals case, LA is consistently superior to MA. Row (c) shows that, in most cases, MA outperforms LA for reversed signals, though there are instances where LA, tuned for equal signal strengths surpasses MA.

