# OpenReview forum: "An Analytical Model for Overparameterized Learning Under Class Imbalance"
_TMLR — Accepted by TMLR_

### Review · Reviewer_kJEg · 2024-12-20

**Summary Of Contributions:**

This paper studies the behavior of various methods to mitigate the class imbalance problem in multi-class regression such as logit adjustment (LA), class-dependent temperature (CDT) and margin adjustment (MA). In a high-dimensional Gaussian mixture setting, the authors are able to derive key properties of the classifiers learnt by these methods, exhibiting their advantages and pitfalls, their hyper-parameter scalings and confirming these theoretical insights in numerical experiments.

**Audience:**

Yes

**Claims And Evidence:**

Yes

**Requested Changes:**

Nothing important. Small remark about the first footnote on the overparameterized regime: it is not clear what is meant here (typically, "overparameterized" means that the number of parameters is higher than the number of input points isn't it?)

**Strengths And Weaknesses:**

Strengths:
- this paper provides a useful theoretical analysis, that enables to gain insights on the behavior of MM, LA, CDT and MA which are not easy to intuit just by looking at the definition of these classifiers
- the theoretical framework is clearly stated, and the limitations of the hypothesis made are discussed
- there is a clear consistency between the theoretical predictions and the numerical results.

Weakness:
- the supplementary material is very long and I did not read it

---

> ### Author Response · Authors · 2025-01-11
>
> ### Strengths and Weaknesses
> Thank you for your thoughtful review. We are pleased that you found our theoretical framework clearly stated and our analysis insightful.
> We acknowledge that the extensive theoretical analysis required a lengthy appendix. To make it easier for interested readers to navigate, we have organized the appendix into smaller sections and referenced the relevant parts within the main paper. To further improve accessibility, we will include a Table of Contents in the appendix in the final version to assist future readers.
> Thank you again for your valuable feedback.
>
> ### Requested changes
> You are correct – our analysis specifically targets this scenario. We will revise the text to clarify that our setting focuses on the regime where the number of training points, $N$, is significantly smaller than the number of parameters, $d$, enabling the model to interpolate the training set.

---

### Review · Reviewer_BKTX · 2024-12-22

**Summary Of Contributions:**

The paper leverages a high-dimensional Gaussian mixture model to study class-imbalanced learning in overparameterized settings. It introduces an analytical framework to estimate test errors for various methods, including logit adjustment (LA), class-dependent temperature (CDT), and margin adjustment (MA). The authors derive closed-form approximations for test errors across different configurations and identify failure modes of cross-entropy minimization in imbalanced scenarios. Additionally, the paper offers practical insights into the optimal tuning of hyperparameters for MA and LA, with experimental results that support the theoretical analysis.

**Audience:**

Yes

**Claims And Evidence:**

Yes

**Requested Changes:**

1. What does $N$ represent?
2. Please write a more clear explanation about Figure 1. In Section G.1, the authors claim that Figure 1 demonstrates the empirical and theoretical worst-class errors eq (2). It's unclear to me which subfigure is for empirical and which is for theoretical.
3. Comparing different methods in Figures 2, 3, and 4 is difficult. If the authors want to compare different methods for various settings, it's better to include all methods (CDT, MA, LA,...) in the same figure and separate settings independently. Or give a table for a more visual comparison.

**Strengths And Weaknesses:**

**Strengths:**
1. The paper provides a theoretical framework for understanding class imbalance in overparameterized settings.
2. It systematically compares MA, LA, and CDT, and identifies their strengths and weaknesses.
3. The paper offers concrete guidelines for hyperparameter tuning, making the results actionable for practitioners.
4. Experimental results support the theoretical analysis.

**Weaknesses:**
1. The paper assumes a simple setting, Gaussian mixture data and linear classifiers, which may limit the applicability of findings to complex neural networks.
2. The description of experiments is not very clear. Especially how the empirical results validate theoretical analysis is unclear.

---

> ### Author Response · Authors · 2025-01-11
>
> Thank you for your thoughtful and constructive review.
>
> ### Strengths and Weaknesses
> We are glad that you found our experimental results align with our theoretical analysis. To keep the main paper focused, detailed experiments are provided in the appendix: Linear classification with the Gaussian model (G.6), RBF kernel classification (G.4), and CLIP fine-tuning (G.5). As mentioned in Section 5, the linear classification experiments validate both our propositions and near-optimal hyperparameter tuning. For the RBF and CLIP experiments, we demonstrate that some predictions, such as MM and CDT catastrophic failure and the optimal worst-class error, hold even outside our theoretical setting. We will make sure to better highlight the connection between our empirical observations and theoretical analysis in the next revision, as detailed below
>
> ### Requested changes
> 1. $N$ refers to the number of samples in the training set. To clarify this, we’ll move the notation section from Appendix A.1 to the main body in the next revision.
> 2. Thank you for your feedback. In the next revision, we will modify the subplot legend to improve clarity for the reader. To clarify the figure, each subplot represents a different data setting, as indicated by the subplot titles, and compares the empirical worst-class error (shaded areas) of each predictor with our theoretical prediction (solid lines). We also emphasize in the figure description that 'The shaded areas indicate empirical measurements, while the solid lines represent our analytical approximation.
> 3. Thank you for your feedback. We decided to represent our results in this way (by keeping the different data settings within the same subplot) to emphasize the validity of our theoretical observations for each predictor. For example, in Figure 2, we clearly show that in various settings, CDT and MM fail, while MA and LA are optimal in terms of worst-class error at the near-optimal hyperparameters. We also find it useful to highlight the effect of hyperparameter tuning on the micro F1-score and balanced test error in this case. Additionally, in Figures 3 and 4, this approach emphasizes that when going beyond our theoretical setting, some observations still hold – such as CDT occasionally failing, MM failing and the optimal worst-class error being achieved near our optimal hyperparameter tuning.

---

### Review · Reviewer_JvdP · 2025-01-06

**Summary Of Contributions:**

This paper proposes an analytical approximation of classification error for imbalanced data, under the setting of linear classification with data sampled from a Gaussian mixture model. The authors analyze their approximation for various loss functions used in the literature under settings of data imbalance, like margin adjustment, class-dependent temperature, and logit adjustment. The approximation relies on viewing the classification problem as an equivalent margin problem and approximating the kernel via its expectation. The authors validate their approximation across 3 experiments: a synthetic GMM setting, RBF kernel on imbalanced CIFAR10 data, and CLIP model.

**Audience:**

Yes

**Claims And Evidence:**

Yes

**Requested Changes:**

In Sec. 3.1 around Eq. (9), the paper asserts that the solution to Eqs. (5) and (6) are dependent only on the kernel matrix. I understand intuitively how this is true and it is sort of demonstrated in Eq. 14, if I understand correctly. Nevertheless, this seems like a bit of a jump in the paper that is not immediately clear to me. Perhaps there could be some justification here or in the Appendix, or perhaps it could be reworded. Especially as this seems to be a central point in the paper.

Some notation with subscripts and bracket indexing was confusing, and took me a bit to understand. E.g. Eq. (10) where the $[i]$ in $\alpha_{y[i]}$ is not the i'th index of $y$ but the i'th index of $\alpha_y$.

**Strengths And Weaknesses:**

Clear writing and notation. Paper is easy to follow.

Approximations seem to follow empirical findings, especially for synthetic data.

---

> ### Author Response · Authors · 2025-01-11
>
> We appreciate your thoughtful feedback. We are pleased to hear that you found our paper clear and that the synthetic data experiments reinforced our theoretical results.
> ### Requested changes
> We will rephrase the beginning of section 3.1 (near eq. 9) such that it emphasizes that when substituting $w_y = \sum_{i\in[N]} \beta_{y[i]} x_{i}$ in equations 5 or 6, we get that the optimization problem is only dependent on the kernel matrix $K$, and we will move eq. 22 in Appendix A.2 into the paper body to ease the reader.
>
> In the notation paragraph (section 2), we explain that the $i$th column of a matrix $A$ is denoted by $a_i$, and that we use brackets to refer to specific matrix or vector entries, e.g., $A_{[i, j]} = a_{i[j]}$. We will add a reminder about this notation around eq. (10).

---

> > ### Comment · Reviewer_JvdP · 2025-01-14
> > **Response**
> >
> > Thanks, I'm happy with the proposed changes on clarity.

---

### Decision · Action_Editor_ujJ1 · 2025-02-13

**Recommendation:** Accept as is

**Comment:**

This paper presents an analytical approximation of classification error for imbalanced data and studies the solutions found by various methods in terms of this novel approximation. This is clearly of interest to some in the community. The reviews were all positive to begin with and what minor concerns there were were addressed fully in rebuttal. This is a clear accept.

**Audience:**

The reviewers agreed that TMLR's audience would be interested in the findings of this paper. This is indicated in comments from reviewers, including one that says that this paper provides "concrete guidelines for hyperparameter tuning" and another that says that this paper provides "a useful theoretical analysis".

**Claims And Evidence:**

The reviewers unanimously agreed that the submission is supported by accurate, convincing and clear evidence. In particular, reviewers noted that the approximations agree closely with the empirical findings, especially for synthetic data. Most of the reviewers felt that the paper was clear and easy to follow, with one reviewer asking for additional clarification on the experiments that was addressed in rebuttal.